# Adult stem cell activity in naked mole rats for long-term tissue maintenance

Shamir Montazid [1,16], Sheila Bandyopadhyay[2,16], Daniel W. Hart[3], Nan Gao[2], Brian Johnson [4], Sri G. Thrumurthy[5], Dustin J. Penn[6], Bettina Wernisch[6], Mukesh Bansal[7], Philipp M. Altrock [8], Fabian Rost [9], Patrycja Gazinska[10], Piotr Ziolkowski[11], Bu'Hussain Hayee[5], Yue Liu[2], Jiangmeng Han[2], Annamaria Tessitore[12], Jana Koth [13], Walter F. Bodmer[13,14], James E. East [15], Nigel C. Bennett[3], Ian Tomlinson[14] ✉ & Shazia Irshad [1] ✉

The naked mole rat (NMR), *Heterocephalus glaber*, the longest-living rodent, provides a unique opportunity to explore how evolution has shaped adult stem cell (ASC) activity and tissue function with increasing lifespan. Using cumulative BrdU labelling and a quantitative imaging approach to track intestinal ASCs (*Lgr5*[+]) in their native in vivo state, we find an expanded pool of *Lgr5*[+] cells in NMRs, and these cells specifically at the crypt base (*Lgr5*[+CBC]) exhibit slower division rates compared to those in short-lived mice but have a similar turnover as human *LGR5*[+CBC] cells. Instead of entering quiescence (G0), NMR *Lgr5*[+CBC] cells reduce their division rates by prolonging arrest in the G1 and/or G2 phases of the cell cycle. Moreover, we also observe a higher proportion of differentiated cells in NMRs that confer enhanced protection and function to the intestinal mucosa which is able to detect any chemical imbalance in the luminal environment efficiently, triggering a robust pro-apoptotic, anti-proliferative response within the stem/progenitor cell zone.

In multicellular organisms, long-term maintenance of tissue homeostasis requires a tight regulation of adult stem cell (ASC) activity to ensure efficient repair and regeneration[1]. In high-turnover mammalian tissues, like the intestine, population equilibrium is primarily controlled through the continual division and differentiation of ASCs, and subsequent apoptosis of mature cells[2]. The longer survival of ASCs puts them at an increased risk of accumulating mutations and reduces their fitness which is seen during ageing[3–5] and in diseases such as cancers[6,7].

The drive to proliferate and the need to maintain genomic integrity are two of the most powerful forces acting on ASCs. Mammalian ASCs in many tissues are predominantly found in a quiescent

[1]Nuffield Department of Clinical Laboratory Sciences, Radcliffe Department of Medicine, University of Oxford, Oxford OX3 9DU, UK. [2]Department of Biological Sciences, Rutgers University, Newark 07102 NJ, USA. [3]Mammal Research Institute, Department of Zoology and Entomology, University of Pretoria, Pretoria 0028, Republic of South Africa. [4]Division of Biomedical Informatics, Department of Medicine, University of California San Diego, 9500 Gilman Dr, La Jolla 92093 CA, USA. [5]Endoscopy, King's College Hospital NHS Foundation Trust, London SE5 9RS, UK. [6]Konrad Lorenz Institute of Ethology, Department of Interdisciplinary Life Sciences, University of Veterinary Medicine, Vienna 1160, Austria. [7]Bristol Myers Squibb, San Diego 92121 CA, USA. [8]Department for Theoretical Biology, Max Planck Institute for Evolutionary Biology, 24306 Ploen, Germany. [9]DRESDEN-concept Genome Center, Center for Molecular and Cellular Bioengineering, Technische Universität Dresden, 01307 Dresden, Germany. [10]Biobank Research Group, Lukasiewicz Research Network, PORT Polish Center for Technology Development, Wroclaw, Poland. [11]Department of Clinical and Experimental Pathology, Wroclaw Medical University, 50-368 Wroclaw, Poland. [12]Novo Nordisk Research Centre Oxford, Oxford OX37FZ, UK. [13]Weatherall Institute of Molecular Medicine, University of Oxford, Oxford OX3 9DS, UK. [14]Department of Oncology, University of Oxford, Oxford OX3 7DQ, UK. [15]Translational Gastroenterology Unit, Experimental Medicine Division, Nuffield Department of Clinical Medicine, John Radcliffe Hospital, Headington, Oxford OX3 9DU, UK. [16]These authors contributed equally: Shamir Montazid, Sheila Bandyopadhyay. ✉e-mail: ian.tomlinson@oncology.ox.ac.uk; shazia.irshad@ndcls.ox.ac.uk

state, allowing them to persist as non-dividing cells over extended periods of time[8–10]. In addition to avoiding DNA replication-induced mutations, quiescent cells also rely on anaerobic glycolysis for ATP production which minimises damage from cellular respiration[11–13]. However, DNA repair is attenuated during quiescence as error-prone nonhomologous end joining repair is utilised[14,15]. A switch from quiescence to proliferation is associated with an increase in the levels of reactive oxygen species[11,16] and accrual of replication errors[17]. The increasing damage that accumulates in actively cycling cells is balanced by the activation of more precise DNA repair by homologous recombination in late S and G2 phases of the cell cycle[15,18–20].

Activation of repair mechanisms temporally delays cell cycle progression until lesions are repaired[21–23]. As DNA replication and cell division are tightly coordinated, correlations between mutations and cell division rates from diverse fields[24–26] have often been interpreted as merely reflecting the replicative origin of mutations, but accrual of mutations should track cell divisions from both replicative and non-replicative sources when repair is efficient[27]. A comprehensive analysis of how long specific cell types, especially ASCs, spend in different phases of the cell cycle has the potential to help understand the contributions of various sources to somatic mutagenesis and reconcile findings of similar mutation rates in human ASCs with different turnover rates[28] and their differentiated progeny in mitotically active and inactive tissues[29].

The need to understand ASC function within their niche has relied on mammalian model systems like mice[30] and species-specific differences in ASC kinetics have largely been overlooked. Recently, xenotransplantation techniques enabling more refined analyses of human ASC dynamics in vivo have shown that the crypt-based columnar (CBC) cells (*LGR5*[+CBC]) in the intestinal epithelium, long believed to be continuously cycling, unlike the +4 or other ASC populations[9], enter a quiescent (G0) state in contrast to murine *Lgr5*[+CBC] cells[31]. These differences in the cell cycle states of ASCs from different species will inevitably affect the types of damage accrued, repair mechanisms employed and, ultimately, somatic mutation rates. Interestingly, a cross-species study that sequenced colonic crypts from 16 mammals has shown an inverse scaling of somatic mutation rates with lifespan suggesting that somatic mutation rates are evolutionarily constrained[32].

The evolutionary adaptations of ASCs, which represent the apex of the cellular hierarchy in tissues, possibly have one of the greatest impacts on organismal health and lifespan. However, there is a critical paucity of in vivo data on the basic parameters of cell division in ASCs and progenitor cells in long-lived species. Accurate estimates of ASC division rates can be achieved with cumulative labelling assays[33,34]. Similarly, determining tissue-specific ASC division rates is important and, in relation to the intestine, although *Lgr5*-expressing cells are the bona fide stem cells in both the small intestine and colon[35,36], local niche factors driving differential ASC division rates need to be considered. Investigating ASC dynamics will enable a better understanding of the underlying mechanisms that contribute to different lifespans and mutation rates across species with diverse evolutionary histories and longevity.

## Results

### *Lgr5* is expressed in crypt base columnar cells in naked mole rats

The intestinal crypt is an anatomically distinct structure that makes up the stem cell niche in many species, including mice and humans, and adult stem cells (ASCs) within the niche are located at the crypt base and marked by *Lgr*5 expression[35,36]. We specifically assessed the presence, abundance, and distribution of *Lgr5*-expressing cells (*Lgr5*[+]) within the NMR intestine. Using an in situ hybridization (ISH) probe specific to NMR *Lgr5* mRNA, we found that the NMR crypts harboured 150% more *Lgr5*[+] cells in the small intestine (duodenum) compared to mouse (Fig. 1a, Supplementary Fig. 1). These

*Lgr5*[+] cells in the NMR small intestine were interspersed by larger cells which most likely represent Paneth cells. In the distal colon, NMR crypts accommodated 57% and 51% more *Lgr5*[+] cells than corresponding mouse and human crypts, respectively (Fig. 1b, Supplementary Figs. 2 and 3).

Spatial analysis of *Lgr5*[+] cell distribution along the vertical crypt axis showed a consistently higher frequency (87%) of *Lgr5*[+] cells at the crypt base at cell positions 0 to 3 in the mouse small intestinal crypts, in agreement with previous findings[35] (Fig. 1c, left) while 76% of *Lgr5*[+] cells occupied positions 0 to 4 in the wider NMR duodenal crypts (Fig. 1c, right). Similarly, single-cell analysis in the colonic crypts revealed that 62% of all murine *Lgr5*[+] cells resided at positions 0 to 4 (Fig. 1d, left), 68% at positions 0 to 4 in humans (Fig. 1d, middle) and 80% resided at positions 0 to 4 in the NMRs (Fig. 1d, right). Therefore, in all three species, the majority of *Lgr5*[+] or *LGR5*[+] cells reside at the basal positions of the crypts.

### NMR *Lgr5*[+CBC] cells are proliferative and do not enter quiescence

To determine any proliferative differences between the NMR and mouse *Lgr5*[+] cells, and human *LGR5*[+] cells, specifically at the crypt base (*Lgr5*[+CBC] or *LGR5*[+CBC]), we co-labelled intestinal tissue from all three species with the proliferation marker, Ki67, and *Lgr5/LGR5* ISH probes specific to each species (Fig. 2a, b, left). Analysis of the small intestine revealed that the majority (82%) of mouse *Lgr5*[+CBC] scored positive for Ki67 (Fig 2a, right), which is in line with previous observations[37], while in the NMR, we detected 65% of *Lgr5*[+CBC]Ki67[+] cells (Fig. 2a, right). In the colon, there was no difference in the Ki67 status of *Lgr5*[+CBC] cells between the two rodent species, with approximately 48% of *Lgr5*[+CBC] cells scoring positive for Ki67 (Fig. 2b, right). However, in human colonic crypts, only 7% of *LGR5*[+CBC] were positive for Ki67 (Fig. 2b, right) which is similar to previous studies showing slower cycling status of crypt base cells in humans[36,38,39]. It is noteworthy that we observed the Ki67 antigen in NMR cells as highly concentrated foci in the nucleus whereas in the mouse, Ki67 staining was mostly present throughout the nucleoplasm (Fig. 2a, b).

In order to assess if *Lgr5*[+CBC] cells in NMRs entered quiescence (G0), we used p27, a cyclin-dependent kinase inhibitor, that modulates the probability that cells will exit the active cell cycle after a mitotic division[40,41]. Firstly, we found no *Lgr5*[+CBC] cells that were also positive for p27 in the mouse small intestine (duodenum), in agreement with the actively cycling status of mouse ASCs[31,35,42] (Fig. 2c). Similarly, we detected no p27 positivity in the NMR *Lgr5*[+CBC] cells of the duodenum (Fig. 2c). We did observe a few cells in the upper region of the small intestinal crypts that expressed p27 in both rodent species (Supplementary Fig. 4a).

Analysis of the colonic tissue revealed that only 8.80% (±1.76%) of murine *Lgr5*[+CBC] cells expressed p27 (Fig. 2d), similar to that reported recently[31], which most likely represents cells arrested in G1. In contrast, we found 86.34% (± 3.05%) of human *LGR5*[+CBC] cells in individuals of 28 to 33 years of age were positive for P27 (Fig. 2d), which is also in agreement with previous findings that used an older cohort (age 50-60 years) and showed P27[+] cells to be predominantly in G0[31]. We found no evidence of p27 expression in the NMR *Lgr5*[+CBC] cells in the colon (Fig. 2d). In all three species, cells further up the colonic crypts showed p27/P27 expression (Supplementary Fig. 4b). Anterior prostate tissues from both mouse and NMR were also used as positive controls to check the cross-reactivity of p27 antibody (Cell Signalling Technology, #3686) used in our study (Supplementary Fig. 4c).

In summary, our data shows that, unlike human *LGR5*[+CBC] cells, a much higher proportion of mouse and NMR *Lgr5*[+CBC] cells exhibit Ki67 positivity that suggests the presence of these cells in the S, G2, and/or M phases of the cell cycle when Ki67 protein is continuously produced[43]. While non-proliferating *LGR5*[+CBC] cells in humans appear to be P27 positive and hence in G0 (quiescence), *Lgr5*[+CBC] cells in mice and NMRs that were unaccounted for by Ki67 staining did not show p27

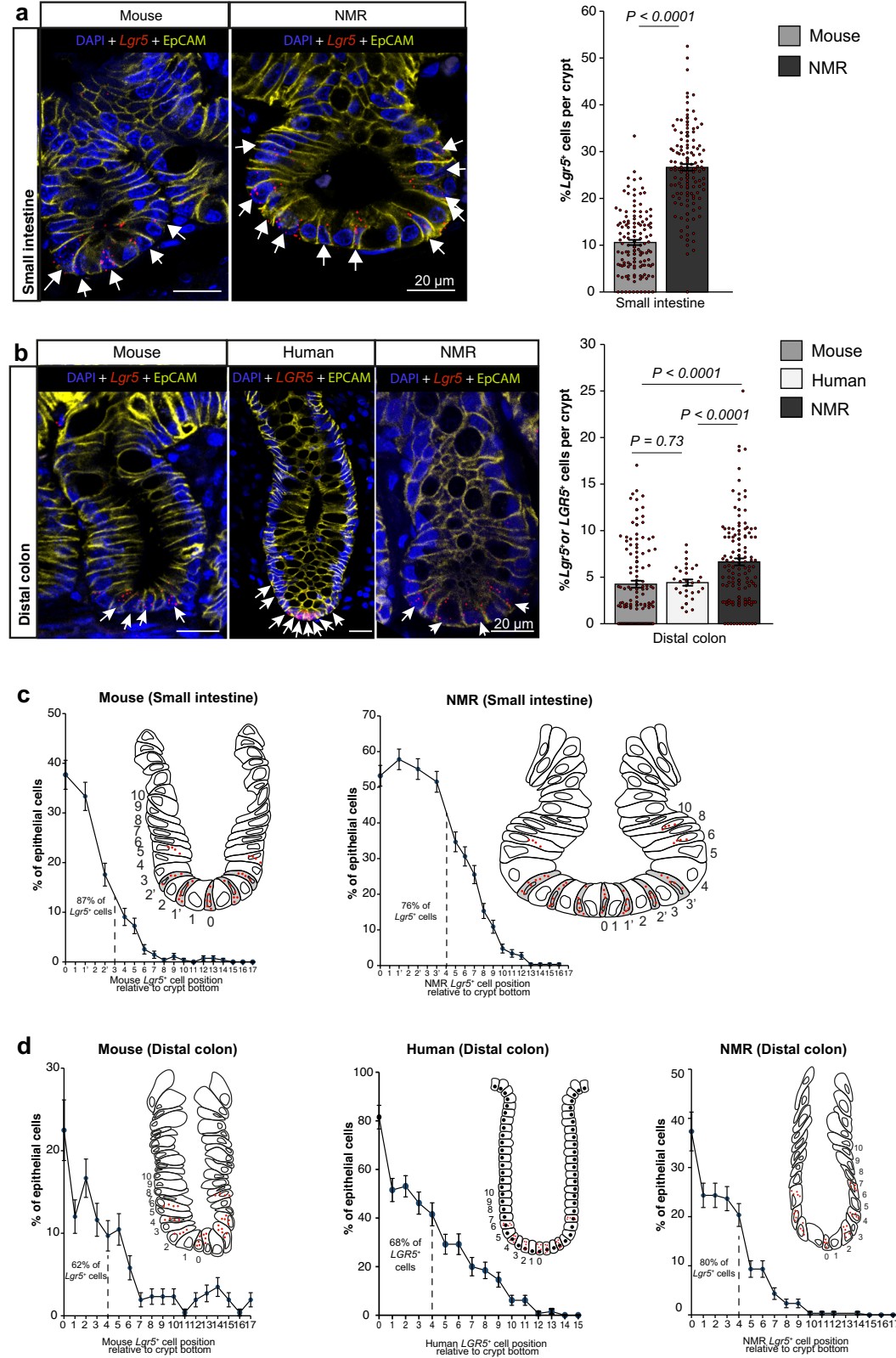

positivity. The lack of Ki67 expression in these rodent cells may be reflective of the undetectable levels of this protein in G1 when Ki67 is continuously degraded[43].

## Cell cycle kinetics of ASCs of NMR small intestine

Ki67 expression provides a snapshot information of a cell's proliferation status. In order to precisely determine the cellular kinetics of ASCs

in the small intestine of wild-caught NMRs (6 to 24 months old), we used a cumulative labelling protocol with 5-Bromo-2'-deoxyuridine (BrdU) to temporally track the uptake of this nucleoside analogue during the DNA-synthesis phase (S) in $Lgr5^{+CBC}$ cells (Fig. 3a). Details of how we first determined the appropriate dose of BrdU to be administered in NMRs (100 mg/kg bodyweight) and the frequency of injections needed (every 8 h) to ensure maximal bioavailability of BrdU

**Fig. 1 | Abundance and location of *Lgr5*+ cells in the NMR intestine.**
**a** Representative confocal images of mouse and NMR small intestinal (duodenum) crypts co-stained with species-specific *Lgr5* RNAscope probe (red), anti-EpCAM antibody (yellow) and DAPI (blue). White arrows indicate *Lgr5*-expressing (*Lgr5*+) cells. Bar graph showing the mean percentage (±SEM) of *Lgr5*+ cells in total epithelial cells per crypt in wild-caught mice (12-months-old) and wild-caught NMRs (12-months-old) (*n* = 138 crypts counted from 3 animals per species, *P* < 0.0001, *t*-test). **b** Co-stained confocal images of the mouse, human and NMR distal colonic crypts. White arrows indicate *Lgr5*+ or *LGR5*+ cells. Bar graph showing the mean percentage (±SEM) of *Lgr5*+ or *LGR5*+ cells per colonic crypt in mice (12-months-old, *n* = 129 crypts counted from 3 animals), humans (28–33 year-old, *n* = 30 crypts from 4 individuals) and NMRs (12-months-old, *n* = 150 crypts from 3 animals). Exact *P*-values from *t*-tests are shown on the graph. **c** Scatter plot showing the mean percentage (±SEM) of *Lgr5*+ cells at specific positions relative to the crypt base in the small intestine. 87% of *Lgr5*+ cells reside at positions 0 to 3 in mice and 76% are found at positions 0 to 4 in NMRs (*n* = 276 cells counted at each position using 3 animals per species). The inset represents the numbering schema used for each cell along the crypt axis. **d** Distribution of rodent *Lgr5*+ or human *LGR5*+ cells (mean percentage ± SEM) at specific positions in the colonic crypts. 62%, 68% and 80% of *Lgr5*+ or *LGR5*+ cells reside at positions 0 to 4 in mouse, human and NMR crypts, respectively (*n* = 258 cells counted at each position from 3 animals per rodent group; *n* = 130 cells counted at each position from 4 human samples). In all cases, statistical significance was determined by Student's *t*-tests using two-tailed, unpaired and unequal variance. Scale bars are indicated on the images (20 μm).

in all proliferating cells of NMR crypts are given in the Supplementary Note 1 and Supplementary Fig. 5.

Previous studies in the mouse small intestine have shown that within 4 h of BrdU administration, a third of crypt-based columnar (CBC) cells labelled positive for BrdU and, within 24 h, all CBC cells per crypt section were BrdU+ or EdU+, implying an average cycling time of these cells in the order of one day[35,44]. By contrast, our experiments in NMRs (6 to 24 months old) showed that after 8 h (0.33 days), only 28% (±1.92) of CBC in the small intestine had taken up the label (Fig. 3b, c). We, therefore, extended the administering of BrdU for a longer period, pulsing NMRs every 8 h for 5 days and analysing intestinal tissue at regular intervals. By day 4, 96.5% (±1.5) of NMR CBCs in the small intestine labelled positive for BrdU (Fig. 3b, c).

To estimate the length of the total cell cycle ($T_T$) and DNA-synthesis phase ($T_s$) specifically in *Lgr5*+CBC cells, we co-stained NMR duodenal tissue with an anti-BrdU antibody and *Lgr5*-ISH probe (Fig. 3d). Based on the assumption that *Lgr5*+CBC cells are asynchronously distributed in the cell cycle and divide asymmetrically[33,45] and do not enter quiescence (G0) (Fig. 2c), we measured the labelling index which is the proportion of cells labelled with BrdU at a specific time in the total *Lgr5*+CBC population (*Lgr5*+CBCBrdU+/*Lgr5*+CBC). The labelling index increased in a linear relationship with time until saturation was reached at day 3.5 (Fig. 3e). We derived the $T_T$ of NMR *Lgr5*+CBC cells from the slope of the regression line ($T_T$ = 1/slope, Eq. 1 in Methods), which was 4.00 days (±0.21) (Fig. 3e). The $T_s$ of these cells was estimated from the y-intercept of the regression line using the labelling index at the instant of the first BrdU treatment at time 0 ($t_0$) ($T_s$ X slope, Eq. 1 in Methods). This was found to be 0.47 day (±0.10) (Fig. 3e). For estimation of $T_T$ and $T_s$ in the mouse duodenal *Lgr5*+CBC cells, we injected BrdU in C57BL/6J mice (2-3 months-old) every 6 h for 54 h and harvested small intestinal tissues hourly after each injection (Fig. 3f, top). BrdU uptake in mice also showed a linear increase with time until saturation was reached after 25 h of BrdU treatment (Fig. 3f, bottom). The $T_T$ of mouse *Lgr5*+CBC cells was estimated as 1.45 days (±0.07) and the $T_S$ was 0.392 day (±0.03) (Fig. 3f, bottom). This division time matched closely with that previously found in mouse small intestinal CBCs where no *Lgr5* staining was performed[46].

We proceeded to determine the length of the M phase ($T_M$) in the mouse and NMR *Lgr5*+CBC cells of the duodenum by quantifying the proportion of *Lgr5*+CBC cells that were positive for phospho-histone H3 (Ser28) (Supplementary Fig. 6a). Using the formula $T_M$ = $T_T$ X *Lgr5*+CBCPHH3+(Ser28)/*Lgr5*+CBC (Eq. 3, Methods), this was estimated to be 0.07 days (±0.01) in mice and 0.06 days (± 0.01) in NMRs (Fig. 3g, left). Having derived the $T_T$, $T_s$ and $T_M$ of *Lgr5*+CBC cells in both species, we next found the length of G1 and G2 phases (Eq. 7, Methods) to be 1.00 days (±0.08) in mice and 3.47 days (±0.23) in NMRs (Fig. 3g, middle). The differences in the length of time spent by duodenal *Lgr5*+CBC cells in specific phases of the cell cycle and comparisons between the two species are shown in Fig. 3g. In summary, we observed no significant difference in the duration of S and M phases between mouse and NMR *Lgr5*+CBC cells. However, these small intestinal

ASCs in NMRs arrest roughly 3 times longer in the gap phases (G1, G2) of the cell cycle compared to mice.

We also used the BrdU labelling index (LI) in vivo at a single time point ($t$) (30 min in mice and 1 day in NMRs) to estimate the $T_T$ of *Lgr5*+ cells above the crypt base which most likely represent early progenitor cells (Fig. 3h). These cells occupy positions 5 to 12 in NMRs and 4 to 10 in mice from the crypt base (Fig. 1c). Using the equation $T_T$ = ($t$ + $T_S$)/LI, for $t \le T_T - T_S$[33] (Eq. 1, Methods) where $T_s$ is assumed to be similar in *Lgr5*+ cells located at different positions in the crypts[31] (0.392 days in mice and 0.47 days in NMRs, Fig. 3e, f), we found the $T_T$ of *Lgr5*+above the crypt base cells was 1.30 days (±0.06) in mice and 2.34 days (±0.17) in NMRs (Fig. 3i, left). In comparison to ASC cell division rates at the crypt base, *Lgr5*+ cells above the crypt base were 12% faster in mice and 71% faster cycling in NMRs (Fig. 3i).

**Cross-species comparison of ASC division rates in the colon**
Similar to the small intestine, *Lgr5*+ cells at the crypt base also function as ASCs in the mouse[35] and human colon[36]. However, differences in the cell cycle kinetics due to differences in the local niche factors driving proliferation between the two tissue types need to be explored. Analysing colonic tissue of wild-caught NMRs (6 to 24-months-old) after successive pulsing with BrdU (Fig. 4a, b), we observed that only 7% (±0.92) of colonic CBCs had taken up the label after 8 h (Fig. 4c). By day 5, 71% (±3.02) of CBCs labelled positive for BrdU (Fig. 4c). To estimate the length of the total cell cycle ($T_T$) and S-phase ($T_s$) specifically in *Lgr5*+CBC cells, we co-stained NMR distal colon with an anti-BrdU antibody and *Lgr5*-ISH probe (Fig. 4d). The proportion of *Lgr5*+CBC cells labelled with BrdU increased linearly with time in the colon (Fig. 4e). We derived the $T_T$ of colonic *Lgr5*+CBC cells in NMRs from the slope of the regression line ($T_T$ = 1/slope, Eq. 1 in Methods), which was 7.52 days (±0.52) (Fig. 4e). Interestingly, these estimates of NMR *Lgr5*+CBC cell division time are very similar to that recently reported for human colonic *LGR5*+CBC cells (7.3 days)[31]. Using the rate of BrdU incorporation and the proportion of *Lgr5*+CBC cells labelled at the instant of the first BrdU injection at $t$ = 0 in the colon, we estimated the $T_s$ to be 0.42 days (±0.08) (Fig. 4e, Eq. 1 Methods). For determining division rates of *Lgr5*+CBC cells in the mouse colon, we administered BrdU in vivo in mice according to the schema in Fig. 4f (top). Using the slope of the linear regression, $T_T$ was estimated to be 2.02 days (±0.08) (Fig. 4f, bottom; Eq. 1 Methods) while the $T_s$ was found to be 0.392 days (±0.04) (Fig. 4f, bottom; Eq. 1 Methods).

We used phospho-histone H3 (ser28) staining in both mouse and NMR colonic tissues to detect cells in the M-phase of the cell cycle (Supplementary Fig. 6b). The length of the M phase ($T_M$) was calculated by quantifying the fraction of *Lgr5*+CBC cells that were positive for phospho-histone H3 using Eq. 3 (Methods). The $T_M$ was similar in mouse and NMR *Lgr5*+CBC cells, 0.08 day (±0.01) and 0.13 day (±0.03), respectively (Fig. 4g, left). Having estimated the $T_T$, $T_s$ and $T_M$, we next calculated the time colonic *Lgr5*+CBC cells spent in the gap phases (G1 and G2) of the cell cycle using Eq. 7 (Methods), which was 1.55 days (±0.09) in mice and 6.97 days (±0.53) in NMRs (Fig. 4g, middle). In summary, no significant differences in the duration of S and M phases

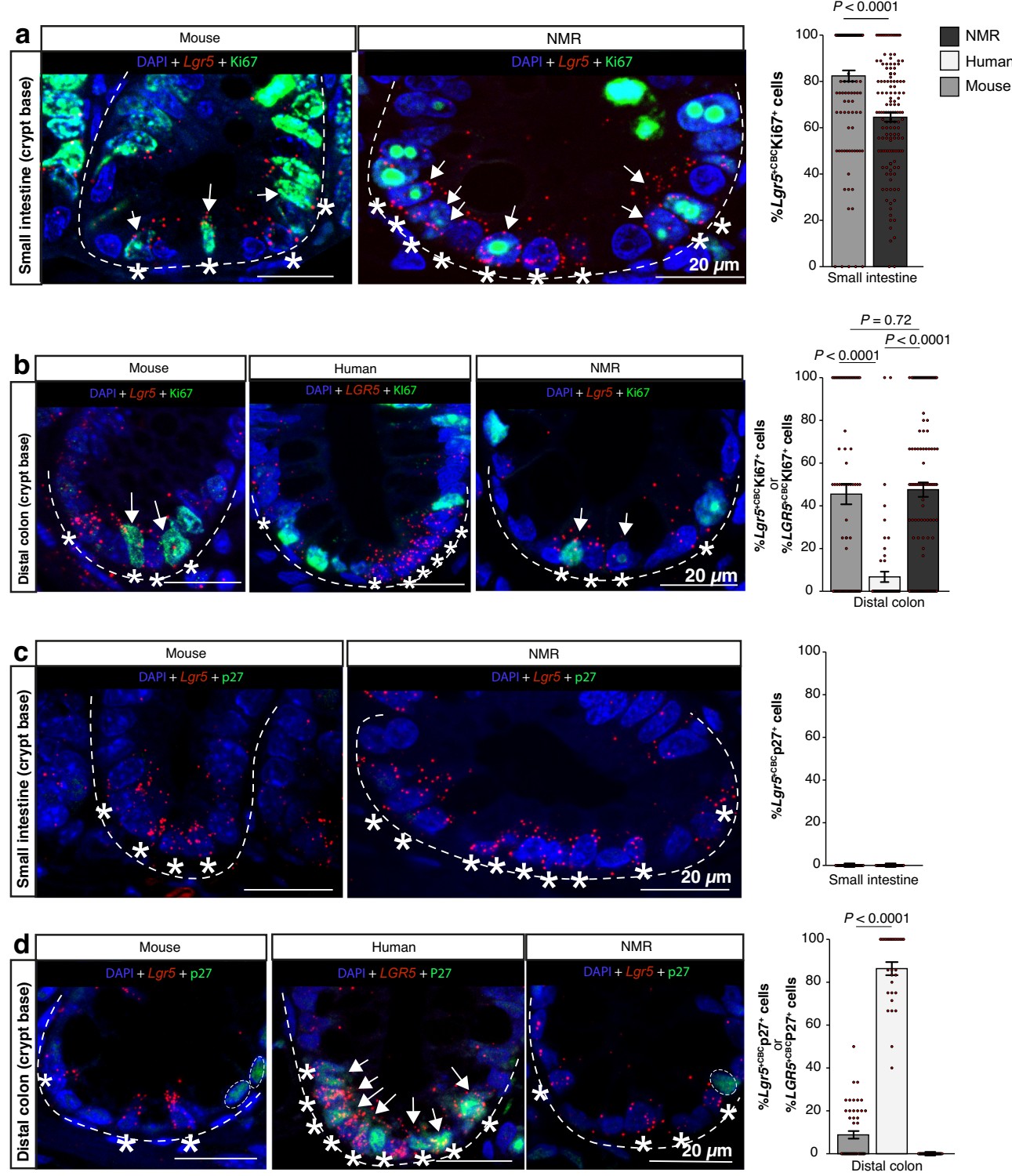

of the cell cycle were observed in mouse and NMR $Lgr5^{+CBC}$ cells residing in the colon, but the gap phases (G1, G2) were 3.5 times longer in NMRs compared to mice (Fig. 4g). It is also noteworthy that $Lgr5^{+CBC}$ cells in the colon of rodent species divided significantly more slowly than their counterparts in the small intestine (88% slower in NMRs and 39% in mice).

Analysis of BrdU incorporation in vivo at time $t$ (30 min in mouse and 1 day in NMR) provided the estimates in the average length of $T_T$ of $Lgr5^{+above\ crypt\ base}$ cells (early progenitors) in the colon (Fig. 4h). These $Lgr5^+$ cells are located at positions 5 to 10 from the crypt base in both

NMRs and mice (Fig. 1d). Using Eq. 1 (Methods) where $T_s$ is assumed to be similar in $Lgr5^+$ cells located at different positions in the crypts[31] (0.392 days in mice and 0.42 days in NMRs, Fig. 4e, f), we found the $T_T$ of $Lgr5^{+above\ the\ crypt\ base}$ cells to be 1.38 days (±0.03) in mice and 2.74 days (±0.18) in NMRs (Fig. 4i, left). Similar to our findings in the small intestine, $Lgr5^+$ cells above the crypt base in the colon divide much faster than colonic ASCs at the crypt base ($Lgr5^{+CBC}$) (46% faster in mice and 174% faster in NMRs, Fig. 4i).

We expanded our analysis to include human colonic $LGR5^{+CBC}$ cells for side-by-side comparisons of cell cycle kinetics with rodent species.

**Fig. 2 | Cycling status versus quiescence of *Lgr5*[+CBC] cells. a** Confocal images showing the crypt-base region of the wild-caught mouse (12-months-old) and wild-caught (12-months-old) NMR small intestine co-stained with *Lgr5* RNAscope probe (red), anti-Ki67 antibody (green) and DAPI (blue). Asterisk (*) indicates *Lgr5*-expressing cells at the crypt base (*Lgr5*[+CBC]) and white arrows mark *Lgr5*[+CBC]Ki67[+] cells. Bar graph showing the mean percentage (±SEM) of Ki67[+] cells in the total population of *Lgr5*[+CBC] cells per crypt in mouse and NMR small bowel (*n* = 126 crypts counted from 3 animals/group, *P* < 0.0001, *t*-test). **b** Confocal images showing the crypt base of mouse, human and NMR distal colons co-stained with species-specific *Lgr5* or *LGR5* RNAscope probe (red), anti-Ki67 or KI67 antibody (green) and DAPI (blue). Asterisks (*) indicate *Lgr5*[+CBC] and white arrows mark *Lgr5*[+CBC]Ki67[+] cells. Bar graph showing the mean percentage (±SEM) of Ki67[+] cells in total *Lgr5*[+CBC] cells per crypt of NMR and mouse colon (*n* = 80 crypts counted from 3 animals/group, *P* = 0.72, *t*-test) and the mean percentage (±SEM) of Ki67[+]*LGR5*[+CBC] cells per crypt in human colon (*n* = 65 crypts from 4 normal human samples, *P* < 0.0001, *t*-test).

**c** Confocal images of mouse and NMR small intestinal crypt base co-stained with species-specific *Lgr5* RNAscope probe (red), anti-p27 antibody (green) and DAPI (blue). Asterisks (*) indicate *Lgr5*[+CBC]. All *Lgr5*[+CBC] cells in the mouse and NMR small intestinal crypts were negative for p27. Bar graphs showing the mean percentage (±SEM) of *Lgr5*[+CBC]p27[+] cells per crypt (*n* = 51 crypts counted from 3 animals/group). **d** Confocal images of the crypt base in mouse, human and NMR colon co-stained with species-specific *Lgr5* or *LGR5* RNAscope probe (red), anti-p27 or P27 antibody (green) and DAPI (blue). Asterisks (*) indicate *Lgr5*[+CBC] or *LGR5*[+CBC] cells. White arrows show *LGR5*[+CBC]P27[+] cells in the human crypt. White circles mark p27 positive cells in mouse and NMR crypts. Bar graphs showing the mean percentage (±SEM) of p27[+] or P27[+] cells in the total population of *Lgr5*[+CBC] or *LGR5*[+CBC] cells per crypt (*n* = 50 crypts counted from 3 animals/rodent group and *n* = 30 crypts from 4 huma*n* samples, *P* < 0.0001, *t*-test). In all cases, Student's *t*-tests using two-tailed, unpaired and unequal variance was used. Scale bars are indicated on the images (20 μm).

---

We first attempted to delineate the time spent by human *LGR5*[+CBC] cells in the active cell cycle (G1 to M) from that in the quiescent state (G0) by using the expression of KI67, phospho-H3 (S28) and P27 to mark the proportion of *LGR5*[+CBC] cells in specific cell cycle phases. In slow-cycling cells, Ki67 has been shown to be in the detectable range in immunofluorescence assays mostly from the S to M phases of the cell cycle[43]. We first detected 11.23% (±3.16) *LGR5*[+CBC] cells that scored positive for KI67[+] in normal human colonoscopy samples (65 to 70 years, *n* = 5) (Supplementary Fig. 7a). Using this proportion of *LGR5*[+CBC]KI67[+] cells and $T_T$ of 7.3 days reported recently[31], we found the total length of time spent by *LGR5*[+CBC] cells in S, G2 and M phases of the cell cycle was 0.82 days (Eq. 2, Methods). By quantifying the proportion *LGR5*[+CBC] cells that were Phospho-Histone H3 (S28) positive (1.20% ±0.30), we found the length of the M phase ($T_M$) to be 0.09 days (Supplementary Fig. 7b, Eq. 3 Methods). Using our estimate of $T_M$ and the duration of the S phase of *LGR5*[+] cells provided previously[31] (0.25 day), we found that human colonic *LGR5*[+CBC] cells spent 0.48 days in the G2 phase of the cell cycle. (Supplementary Fig. 7b, Eq. 4 Methods).

We next used P27 expression to quantify the proportion of *LGR5*[+CBC] cells in the G1 and G0 phases of the cell cycle and found that 84.8% (±1.91) were P27 positive (Supplementary Fig. 7a). Using Eq. 5 ($T_{(G1, G0)}$[P27+] = $T_T$ (ref. 31) X *LGR5*[+CBC]P27[+]/ *LGR5*[+CBC], Methods), we estimated that *LGR5*[+CBC] cells spent 6.19 days in G1 and G0 (Supplementary Fig. 7b). To discriminate *LGR5*[CBC] cells in G0 from those in G1, we used data published by Ishikawa et al. where double staining of Pyronin Y and Hoechst 33342 was used to identify LGR5[+]P27[+] cells with DNA[LOW]/RNA[LOW] content in G0 and DNA[LOW]/RNA[HIGH] in G1[31]. Approximately 83% of LGR5[+]P27[+]cells were found to be in the G0 phase[31], referred to as the quiescence fraction (QF). Using Eq. 6 ($T_{G0}$[P27+] = QF (ref. 31) × $T_{(G1, G0)}$[P27+], Methods), we estimated the duration of G0 to be roughly 5.14 days (Supplementary Fig. 7b). Finally, we found the duration of $T_{G1}$ (Eq. 7 Methods) was 1.34 days (Supplementary Fig. 7b). In summary, our cross-species comparison shows that human and NMR colonic ASCs divide slower than mouse ASCs (*Lgr5*[+CBC]) and while human *LGR5*[+CBC] cells slow down their division rates by entering quiescence (G0), NMR *Lgr5*[+CBC] cells extended their cell cycle by spending the majority of their time in the gap phases (G1/G2) (Fig. 4j).

## ASC division rates scale with lifespan and correlate with somatic mutation rates

Using our estimates of *Lgr5*[+CBC] cell division rates in mice and NMRs, we sought to look for a relationship between the rate of ASC division and lifespan. First, we assessed the congruence between Ki67 positivity and cell division rates determined by BrdU labelling (Fig. 3e–f, Fig. 4e–f). We found that there was a positive association between Ki67 positivity and cell division rates per year of *Lgr5*[+CBC] cells in the NMR and mouse intestinal tissues (Fig. 5a). We, therefore, used Ki67 or KI67 as a surrogate marker to assess any changes in the rodent *Lgr5*[+CBC] or human *LGR5*[+CBC] cell division rates at different ages in all three species. The

proportion of *Lgr5*[+CBC]Ki67[+] cells did not change in the small intestine between 6 months, 1-year and 3-year-old NMRs (Fig. 5b, left). Similarly, there were also no proliferative differences in the small intestinal *Lgr5*[+CBC] cells between 2-months, 12-months and 18-months-old mice (Fig. 5b, middle). We also found no change in the KI67 positivity in human colonic *LGR5*[+CBC] cells with increasing age (28 to 74 years; Fig. 5b, right).

Having established that the cell division rates of *Lgr5*[+CBC] or *LGR5*[+CBC] cells did not alter with age, we next investigated the relationship between ASC division rates and lifespan. Using log-log allometric regression analysis, we found an inverse correlation of ASC division rates in the colon with lifespan (*P* = 0.002, 95% confidence interval, Fig. 5c, left). A similar qualitative relationship was observed with the *Lgr5*[+CBC] cell division rates in NMR and mouse small intestine (Fig. 5c, left). Interestingly, the cell division rates of rodent *Lgr5* or human *LGR5* cells above the crypt base, which proliferate faster than *Lgr5*[+CBC] or *LGR5*[+CBC] cells in all three species (Figs. 3i, 4i and ref. 31), did not correlate with lifespan (*P* = 0.33, Fig. 5c, right). We predict that by the end of their life, *Lgr5*[+CBC] cells in mice with a lifespan of 3.7 years (age at which 80% of individuals reaching adulthood die)[32] would have undergone approximately 931 divisions in the small intestine and 669 in the colon. In NMRs, *Lgr5*[+CBC] cells will have divided 2281 times in the small intestine and 1213 in the colon by the end of life[32]. Based on the cell division estimates of human colonic *LGR5*[+CBC] cells[31], the total divisions by the end of life (83.6 years)[32] in humans will be approximately 4180.

Somatic mutation rates in the colonic crypts across 16 mammalian species have recently been shown to scale with lifespan[32], and the same study used previous estimates of ASC division rates in mice[47], rats[48] and humans[39,49] to suggest that differences in ASC kinetics cannot fully explain differences in somatic mutation rates and burden across species[32]. We used our estimates of *Lgr5*[+CBC] division rates in mouse and NMR colons and those previously described for human colonic *LGR5*[+CBC] cells[31] to understand to what extent the observed somatic mutation rates[32] can be attributed to differences in the ASC division rates. Using log-log allometric regression analysis, we observed a positive correlation between the somatic mutation rates and ASC division rates in the colon (Fig. 5d). Under a simpler model where mutation accumulation in the crypt is a result of mutations which occur at cell division in ASCs, we found that in mice every cell division would lead to about 4.4 substitution mutations, compared to 2 in NMRs and 1 in humans. In mice, substitution mutation accumulation in the crypt occurred at a rate roughly 17 times faster than in humans and 9 times faster than NMRs[32]. Normalising substitution mutation rate per species by *Lgr5*[+CBC] or *LGR5*[+CBC] cell division rates (181 divisions/year in mice and 50 divisions/year in humans) reduced the fold difference between mouse and human mutation rates to 5, thus roughly 72% of the excess mutations in mice per year can be accounted for by ASC division rates. The equivalent comparison between mouse and NMR

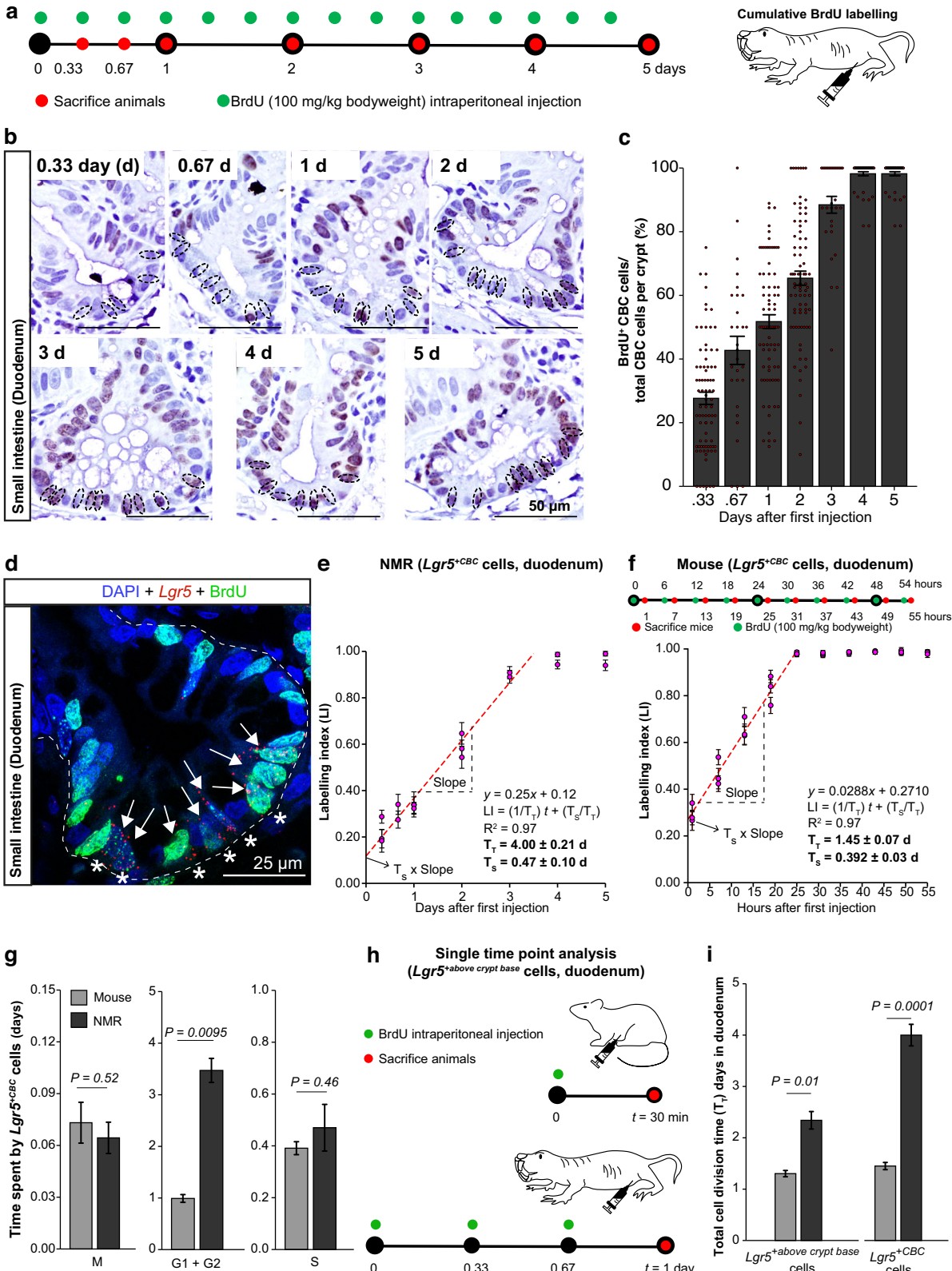

**Cumulative BrdU labelling**

**a**

Sacrifice animals ● BrdU (100 mg/kg bodyweight) intraperitoneal injection ●

**b** Small intestine (Duodenum): 0.33 day (d), 0.67 d, 1 d, 2 d, 3 d, 4 d, 5 d; 50 µm

**c** BrdU⁺ CBC cells/ total CBC cells per crypt (%) vs Days after first injection

**d** Small intestine (Duodenum): DAPI + *Lgr5* + BrdU; 25 µm

**e** NMR (*Lgr5*⁺ᶜᴮᶜ cells, duodenum)

Labelling index (LI) vs Days after first injection

$y = 0.25x + 0.12$
$LI = (1/T_T)\,t + (T_S/T_T)$
$R^2 = 0.97$
$T_T = 4.00 \pm 0.21$ d
$T_S = 0.47 \pm 0.10$ d

**f** Mouse (*Lgr5*⁺ᶜᴮᶜ cells, duodenum)

Sacrifice mice ● BrdU (100 mg/kg bodyweight) ●

Labelling index (LI) vs Hours after first injection

$y = 0.0288x + 0.2710$
$LI = (1/T_T)\,t + (T_S/T_T)$
$R^2 = 0.97$
$T_T = 1.45 \pm 0.07$ d
$T_S = 0.392 \pm 0.03$ d

**g** Time spent by *Lgr5*⁺ᶜᴮᶜ cells (days): Mouse, NMR

M: $P = 0.52$; G1 + G2: $P = 0.0095$; S: $P = 0.46$

**h** Single time point analysis (*Lgr5*⁺ᵃᵇᵒᵛᵉ ᶜʳʸᵖᵗ ᵇᵃˢᵉ cells, duodenum)

● BrdU intraperitoneal injection
● Sacrifice animals

**i** Total cell division time (T_T) days in duodenum

$P = 0.01$; $P = 0.0001$

*Lgr5*⁺ᵃᵇᵒᵛᵉ ᶜʳʸᵖᵗ ᵇᵃˢᵉ cells, *Lgr5*⁺ᶜᴮᶜ cells

substitution rates per ASC division rate showed 73% higher mutation rates in mice resulting from having a faster ASC turnover. In NMRs and humans, ASC division rates were similar but there was a 1.96-fold difference in mutation rates which is possibly due to differences in either the repair efficiency or differential damage accrued by NMR *Lgr5*⁺ᶜᴮᶜ and human *LGR5*⁺ᶜᴮᶜ cells. The longer arrest of NMR *Lgr5*⁺ᶜᴮᶜ cells in G1 and/or G2 compared to the extended period of G0 of human *LGR5*⁺ᶜᴮᶜ

cells (Fig. 4j) suggests that NMR cells may experience higher damage due to increased metabolic rates in the active cell cycle. In summary, we show that ASC division rates play an important role in the accrual of somatic mutations, with fast-dividing mouse lineages accumulating much higher mutation rates while longer-lived species like NMRs and humans reduce their mutation rates by slowing down their ASC turnover.

**Fig. 3 | Cytokinetics of *Lgr5*[+CBC] in the NMR small intestine. a** Schema showing BrdU injection schedule in NMRs (green) and time of tissue collection (red). **b** NMR crypts co-stained with anti-BrdU antibody (brown) and haematoxylin (blue). Dotted circles indicate crypt-based columnar (CBC) cells. **c** Bar graphs showing the mean percentage (±SEM) of labelled CBC cells (BrdU[+]CBC) at different time points in NMRs, derived by counting *n* = 20 crypts per animal (3 animals sacrificed at 0.33, 1, 2 days and 2 at 0.67, 3, 4, 5 days). **d** Immunofluorescent image showing *Lgr5* mRNA (red), BrdU (green), and DAPI (blue) in NMR crypt injected with BrdU for 3 days. Asterisks (*) mark *Lgr5*[+] cells at the crypt base (*Lgr5*[+CBC]) and white arrows indicate *Lgr5*[+CBC]BrdU[+] cells. **e** Scatter plot showing a linear relationship between the ratio of *Lgr5*[+CBC]BrdU[+] cells to the total number of *Lgr5*[+CBC] cells (labelling index) with time in NMRs until BrdU labelling becomes saturated. Each data point represents the mean (±SEM) of 20 crypts counted per animal (*n* = 3 animals at 0.33, 1 and 2 days; *n* = 2 at 0.67, 3, 4 and 5 days). **f** Top, BrdU injection times in mice (green) and time points when small intestinal tissues were collected (red). Bottom, scatter plot showing a linear increase in BrdU labelled murine *Lgr5*[+CBC] cells with time. Each dot denotes the mean (±SEM) of 30 crypts counted per animal (*n* = 3 animals/time point). **g** Bar graphs (mean ± SEM) showing the length of specific cell cycle phases of mouse and NMR *Lgr5*[+CBC] cells. Exact *P*-values have been indicated on the graphs. **h** Schema showing the experimental design for estimating the total cell cycle time, $T_T$, of *Lgr5*[+] cells located above the crypt base by injecting BrdU (green) in mice and NMRs and time points when tissues were collected (red). **i** Bar graphs showing the mean (±SEM) $T_T$ of *Lgr5*[+] cells above the crypt base (left) and *Lgr5*[+CBC] (right) in mice and NMRs. Exact *P*-values have been indicated on the graphs. Statistical significance was determined by performing Student's *t*-tests using two-tailed, unpaired and unequal variance. Scale bars are indicated on the images (25 µm or 50 µm).

## NMR intestine is resistant to dextran sodium sulphate (DSS) treatment

To assess differences above the stem cell zone in the NMR intestine, we also characterised the differentiated cells in these animals which showed a higher proportion of enteroendocrine, enterocytes and goblet cells compared to mice (Supplementary Note 2, Supplementary Fig. 8, Supplementary Table 1). We observed a thicker mucus layer and elevated expression of mucins in the NMR intestine (Supplementary Note 2, Supplementary Fig. 8d), which suggested an enhanced protection of the underlying epithelium from invading pathogens, as well as to toxins and other environmental irritants. To test this we treated wild-caught NMRs with dextran sodium sulphate (DSS), which is a commonly used chemical that penetrates the mucosal membrane and induces acute intestinal injury and inflammation in mice[50,51]. In mice, a typical protocol involves administering 2.5% DSS in drinking water for 7 days with disease induction occurring within 3-7 days[51]. This method of delivering DSS could not be used in NMRs as they do not drink water and get all their water requirements from their plant-based diet. For direct comparisons, we, therefore, attempted to administer DSS in both mice and NMRs by oral gavaging that ensured delivery into the stomachs of the animals (Fig. 6a). Oral gavaging can induce significant stress in animals and, therefore, our strategy involved gavaging every 4 h during the day for 3 consecutive days in total (Fig. 6a). There were significant differences in the extent of intestinal damage caused by the two methods of delivering DSS (via drinking water or oral gavaging) in mice which are discussed in detail in Supplementary Note 3 (Supplementary Fig. 9). Differences in weight loss experienced by mice using the two methods is also discussed in Supplementary Note 4 (Supplementary Fig. 10).

Oral gavaging in mice (C57BL/6J, 2 to 4 months-old) for 3 days resulted in mild damage in roughly 30% of the distal colon (Fig. 6b). In these damaged regions, we observed beginnings of surface epithelial erosion and crypt atypia, changes in the crypt size, crypt branching and expansion into the muscularis externa (Fig. 6b). There was also some modest infiltration of immune cells into the lamina propria and submucosa region (Fig. 6b, Supplementary Fig. 9c, e). Normal crypts were seen in undamaged regions of the colon in DSS-treated mice (Fig. 6b). There were no signs of ulcerations, submucosal oedema or complete loss of surface epithelium in the distal colon which were commonly observed in the longer 7-day treatment with 2.5% DSS (Supplementary Fig. 9d, e). Our intestinal injury model of 3 days by oral gavaging in mice resembled a chronic model of colitis with a low degree of dysplasia-like atypia where 1% DSS was administered in water over a period of 42 days[52].

Interestingly, in NMRs (6 to 24 months old), 2.5% DSS treatment via oral gavage did not even induce mild epithelial damage (Fig. 6c) and no inflammation was detected in any part of the colon (Supplementary Fig. 11a). We further attempted to induce damage in NMRs by increasing the concentration of administered DSS to 5.0% and 8.75%. In 5.0% DSS-treated NMRs, the stool consistency changed from normal hard black (score 0) to soft (score 1) while in the 8.75% treated NMRs, the stool consistency became very soft and slightly watery (score 2) (Fig. 6d). However, histological analysis of the intestinal tissue of NMRs treated with these higher concentrations of DSS showed an intact epithelial lining (Fig. 6e) and there were no signs of inflammation (Supplementary Fig. 11a). Pathological scoring by two independent pathologists comparing DSS-treated mice and NMRs is summarized in Fig. 6f and Supplementary Fig. 11a. It is possible that if the experiment had been extended for longer than 3 days, we may have detected some damage in the NMR intestine, but in our side-by-side comparison with mice, we concluded that NMRs are more resistant to tissue damage by DSS.

## Dextran sodium sulphate (DSS) triggers a robust pro-apoptotic and anti-proliferative response within NMR crypts

We expanded our analysis to assess any perturbations in the intestinal crypt cells after 2.5% DSS treatment for 3 days in mice and NMRs. In mice, the number of *Lgr5*[+] cells at the crypt base (*Lgr5*[+CBC]) in regions of mild damage (crypt atypia) was similar to unaffected regions of the distal colonic crypts in DSS-treated and control animals (3 to 4 *Lgr5*[+CBC] cells per crypt) (Fig. 7a, b, left). Similarly, in the most proximal regions of the intestine (duodenum), DSS-treated mice showed no change in the *Lgr5*[+CBC] cell numbers compared to controls (Fig. 7b, right). This observation is in agreement with a previous report that showed that the magnitude of loss of *Lgr5*[+] cells is dependent on the administered dose of DSS, with complete ablation only observed in affected regions after treatment with 3% DSS for 6 days[53]. Analysis of DNA fragmentation using terminal-deoxynucleotidyl-transferase-dUTP-nick-end-labelling (TUNEL) assay enabled in situ visualization of the apoptotic process at a single cell level[54]. We observed a roughly 3-fold increase in TUNEL[+] cells at the crypt base in the distal colon as well as in the duodenum of DSS-treated mice compared to controls (Fig. 7c, left). In cells residing above the crypt base at positions 5 to 13 (colon) and 4 to 13 (small intestine), we found a modest increase in the apoptotic index in the distal colon and no change in the duodenum of treated mice (Fig. 7c, right).

Surprisingly, analysis of DSS-treated NMR intestinal tissues by ISH showed a complete absence of *Lgr5* expression in cells at the base and above the crypts in distal colons (Fig. 7d, e). Additionally, DSS treatment also affected the *Lgr5* expression in the NMR small intestine. Quantification of *Lgr5*-mRNA fluorescent puncta showed no signal in 86% of cells at positions 0 to 6 along the crypt axis in the duodenum (Fig. 7e). To confirm if the loss of *Lgr5* expression in DSS-treated NMRs was due to mRNA degradation triggered early in apoptosis[55], we used TUNEL staining on NMR tissues. In 2.5% DSS-treated NMRs, we observed highly elevated levels (18-fold increase) of TUNEL[+] cells in the crypt base cells of the distal colon and a 4-fold increase in the duodenal crypt base compared to controls (Fig. 7f, left). In cells above the crypt base located at positions 5 to 13, we also found an increase in TUNEL[+] cells in the distal colon and duodenum of treated NMRs (Fig. 7f, right).

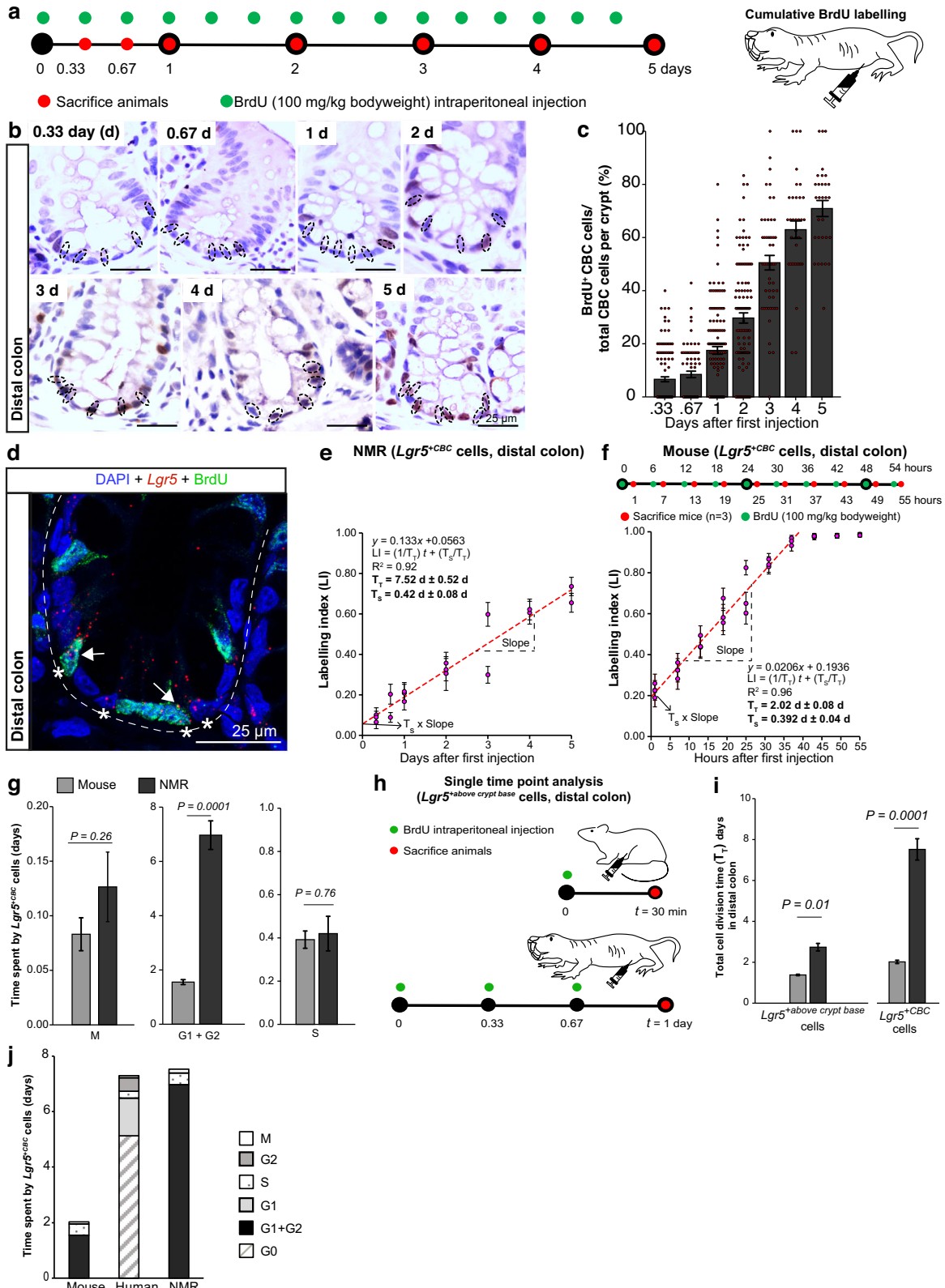

The precise fold change differences in the apoptotic index following 2.5% DSS treatment for 3 days between the two rodent species and with a previous report of mice treated for 5 days[56] is summarised in Supplementary Table 2. In summary, in response to DSS-treatment, crypt-based cells representing ASCs are more extensively eliminated by apoptosis in NMRs compared to mice. Additionally, unlike in mouse, apoptosis of cells within the progenitor zone is not

confined to the distal colon but observed also in the proximal regions of the NMR small intestine.

Alterations in the cell proliferation of the distal colon have been reported previously in mice treated for 5 days with 2.5% DSS[56,57]. We also sought to assess any proliferative changes following 3 days of oral gavaging with 2.5% DSS in mice and NMRs. We found that there was no change in the Ki67 status of *Lgr5*[+CBC] cells in the murine distal colon

**Fig. 4 | Cytokinetics of colonic *Lgr5*[+CBC] across species. a** Schematic representation of BrdU injections (green) in NMRs and time of colonic tissue collection (red). **b** Images of NMR colonic crypt stained with anti-BrdU antibody (brown) and haematoxylin (blue). CBC cells are marked by dotted circles. **c** Bar graphs showing the mean percentage (±SEM) of labelled CBC cells (BrdU[+]CBC) over time, derived from counting *n* = 30 crypts per animal (3 NMRs sacrificed at 0.33, 1, 2 days; 2 at 0.67, 3, 4, 5 days). **d** Confocal image showing *Lgr5* mRNA (red dots), BrdU (green) and DAPI (blue) in NMR colonic crypt cells after 3 days of BrdU injection. Asterisks (*) mark *Lgr5*[+] cells at the crypt base (*Lgr5*[+CBC]). White arrows indicate *Lgr5*[+CBC]BrdU[+] cells. **e** Scatter plot showing a linear increase in the BrdU labelling index (Number of *Lgr5*[+CBC]BrdU[+]/Total number of *Lgr5*[+CBC]) over time in NMRs. Each dot represents a mean of 40 crypts (±SEM) counted per animal (*n* = 3 animals at 0.33, 1 and 2 days; *n* = 2 animals at 0.67, 3, 4 and 5 days). **f** Top, Schema showing times when BrdU was injected (green) and colons harvested (red) in mice. Bottom, scatter plot showing a linear increase in the labelling index of murine *Lgr5*[+CBC] cells with time. Each dot denotes the mean (±SEM) of 30 crypts counted per animal (*n* = 3 animals/time point). **g** Bar graphs showing the duration of specific cell cycle phases of *Lgr5*[+CBC] cells in mice and NMRs. Exact *P*-values are indicated on the graphs. **h** Experimental design for measuring the total cell cycle time ($T_T$) of *Lgr5*[+] cells above the crypt base by injecting BrdU (green) in mice and NMRs and time of tissue collection (red). **i** Bar graphs showing $T_T$ of colonic *Lgr5*[+] cells above the crypt base (left) and *Lgr5*[+CBC] cells (right) in mice and NMRs. Exact *P*-values are indicated on the graphs. **j** Stacked bar graphs showing the time (days) spent by mouse, human and NMR colonic ASCs in each phase of the cell cycle. Student's *t*-tests using two-tailed, unpaired and unequal variance were used. Scale bars are indicated on the images (25 μm).

(normal or atypical crypts) or in the damage-resistant crypts of the duodenum (Fig. 7g). A similar analysis of NMR *Lgr5*[+CBC] cells that had not undergone apoptosis in the duodenum showed no change in the Ki67 status compared to controls (Fig. 7h). However, we observed statistically significant changes in the Ki67 status of cells above the crypt base in both DSS-treated mice and NMRs. There was a 24% reduction in the proportion of Ki67[+] cells in the distal colonic crypts (both normal and atypical) of DSS-treated mice compared to controls (Fig. 7i, left). We even observed a 13% decrease in Ki67[+] cells in the duodenal crypts of treated mice (Fig. 7i, right). In NMRs, DSS treatment led to a much more profound decrease in the Ki67 positivity, with 88% reduction in the distal colon and 62% in the duodenum, compared to controls (Fig. 7j). Supplementary Table 2 provides a summary of the proliferative differences in the crypt cells following DSS treatment between the two rodent species and how these compare to a previous study in mice which used 2.5% DSS for 5 days[56].

Finally, to assess the ability of NMR crypts to return to normal, we withdrew DSS and allowed animals to recover for 14 days and analysed intestinal tissues every 1 to 2 days for 7 days (Fig. 7k). *Lgr5*[+CBC] cells in NMRs returned to numbers seen at homoeostasis, 4 cells/crypt in the distal colon and 8 cells/crypt in the duodenum, after 5 and 3 days of the recovery period, respectively (Fig. 7l, Supplementary Fig. 11b). This re-emergence of the *Lgr5*[+] cells suggests a source of another reserve stem cell population or dedifferentiation of progenitor cells. In summary, our results show that the NMR intestine responds to low levels of chemical insults by triggering apoptosis in the majority of *Lgr5*[+] cells across the entire intestine and by shutting down cellular proliferation in the progenitor cells of the crypts.

## Discussion

In mice and human intestines, *Lgr5*[+] cells found at the base of the crypt of Lieberkühn (*Lgr5*[+CBC]) are the bona fide ASCs that bear the regenerative burden required for tissue maintenance[35,36]. We find that the majority of *Lgr5*[+] cells in NMRs are also present at the crypt base in the small intestine and colon. However, NMR crypts harbour a higher proportion of ASCs compared to both mice and humans. Interestingly, an expanded poor of ASCs is also seen in the NMR haematopoietic system[58]. Our study is the first to cumulatively label the NMR proliferative population of ASCs and progenitors in vivo, allowing accurate measurements of division rates[33]. Surprisingly, although NMRs are phylogenetically closer to mice, turnover rates of *Lgr5*[+CBC] cells in NMRs are much more similar to human *LGR5*[+CBC] cells[31]. The slower cellular kinetics of ASCs may be a general characteristic across tissue types in NMRs as our findings in the small intestine and colon mirror the slower proliferation of NMR neural[59] and haematopoietic[58] stem cells. Other cell types like NMR iPSCs[60] and fibroblasts[61] also divide slower than murine counterparts and while cell-autonomous growth suppressive mechanisms have been identified and linked to cancer resistance in these cell lineages[60,62–64], future studies need to be undertaken to unravel specific cell cycle modulators in NMR ASCs.

The in vivo division rates of mouse and NMR *Lgr5*[+CBC] and human *LGR5*[+CBC][31] cells in the small intestine and colon showed an inverse correlation with lifespan but, intriguingly, no relationship was seen when using division rates of *Lgr5*[+] or *LGR5*[+] cells above the crypt base, which most likely represent early progenitors[65] and divide faster in all three species. This suggests that evolutionary constraints are most likely to act on ASC populations. We also observed a positive correlation between our estimates of ASC division rates in the colon and reported somatic mutation rates[32]. This is in line with previous phylogenetic[24] and comparative X, Y chromosomes and autosomes studies[66–68] showing that the accrual of mutations tracks with cell divisions. Our study also provides a unique insight into the non-replicative origins of somatic mutagenesis. The longer G1 and/or G2 arrest of NMR ASCs compared to extended G0 of human ASCs suggests that higher damage due to increased metabolic rates at G1 and G2 phases would increase the non-replicative errors in NMRs, which may partially explain the 2-fold difference in the substitution rates in the NMR colonic crypt cells compared to human counterparts[32].

We have also characterised the differences in the differentiated zone of the NMR intestine compared to mice. The selection of significantly shorter intestinal length in NMR has necessitated the co-emergence of longer villi in the small bowel, presumably to maximize surface area for nutrient and water absorption. The proportion of specialised cells (e.g. goblet, enteroendocrine and enterocytes) in the intestine, that carry out the main function of digestion and absorption, is also much higher in the NMR intestine compared to mice. The increased goblet cells, elevated expression of *Muc2* and the presence of a thicker intestinal mucus layer in NMRs provide a more robust physical barrier between the epithelial cells and luminal contents of the intestine that is harder to penetrate compared to mice. Indeed, we failed to induce even mild mucosal injury or inflammation in NMRs after treatment with very high concentrations of DSS for 3 days. Resistance to stressors is emerging as a common trait of long-lived organisms[69–71], including in NMRs[72,73]. However, even in the absence of any overt histological damage of the intestine, DSS treatment in the NMRs resulted in a more robust response in the stem and progenitor cells compared to mice. In NMRs, the majority of *Lgr5*-expressing cells undergo apoptosis and there is a drastic reduction in the proliferation of cryptal cells from the proximal to the distal regions of the intestine. Our results imply that the shorter intestinal tract in NMRs has evolved highly efficient sensory mechanisms to detect oxidative stress[74] in the lumen and their first line of defence against such an insult is to eliminate or halt the proliferation of most of their cycling cells in the crypts, and as such minimise macromolecular damage in ASCs, thus ensuring no damage is passed on to daughter cells. Finally, we show the re-emergence of *Lgr5*[+] cells after the removal of DSS which suggests a source of another reserve stem cell population or dedifferentiation of progenitor cells that serve to replenish *Lgr5*[+] cells after injury, similar to that previously shown in mouse[75,76].

In summary, our study characterising the intestinal tract of NMRs adds to the growing line of evidence showing how these

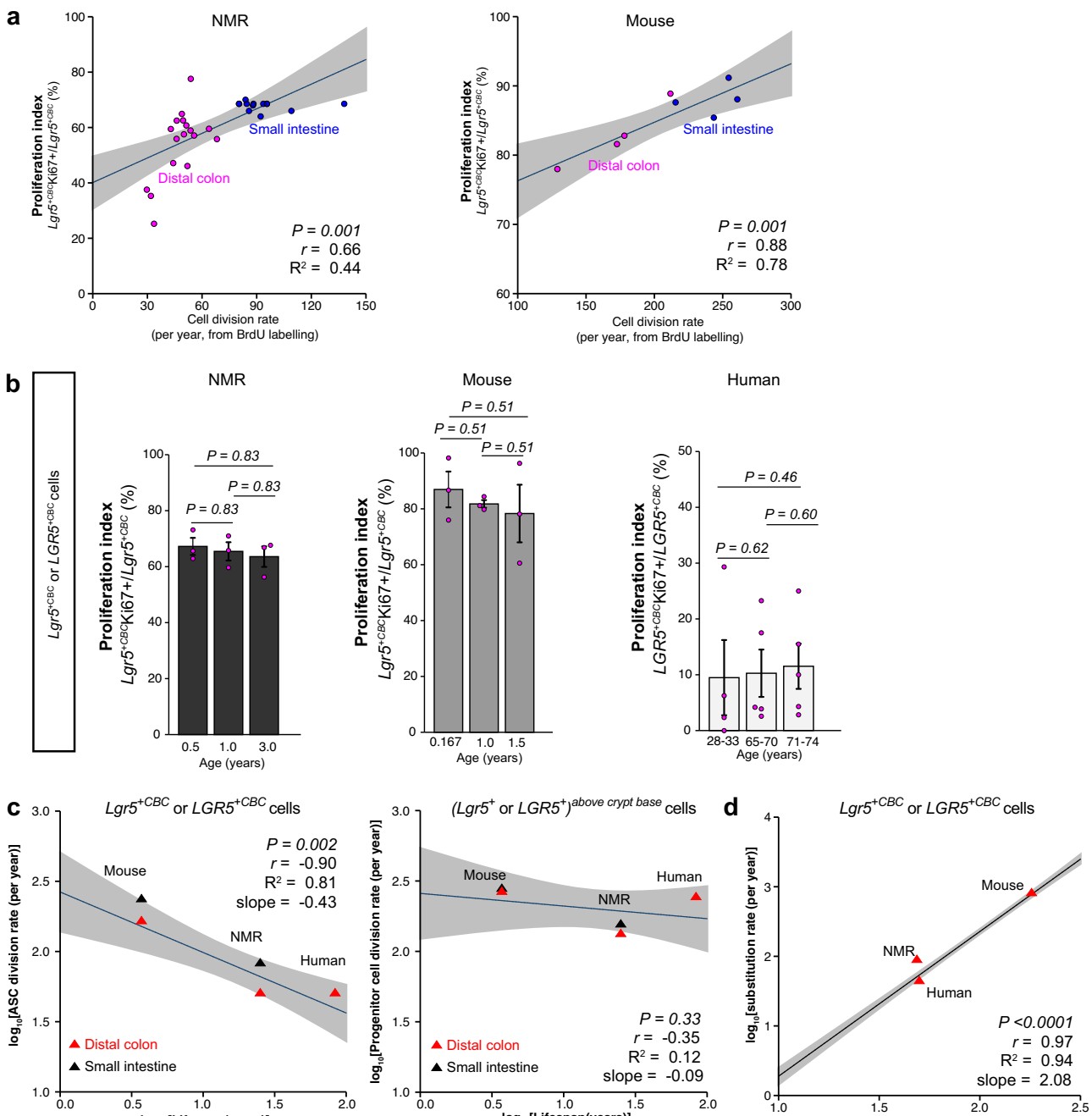

**Fig. 5 | ASC division rates are inversely proportional to lifespan and positively associated with somatic mutation rates. a** Regression analysis showing a positive correlation between *Lgr5*[+CBC] cell division rates per year determined by BrdU labelling and Ki67 positivity in NMR (left) and mouse (right) small intestine and colon. Each data point represents the mean value of 30 crypts counted per animal and the shaded area represents 95% confidence interval of the regression line. *P*-values and $R^2$ from two-tailed *F*-tests and correlation coefficients (*r*) are indicated on the graphs. **b** Bar graphs showing mean percentage (±SEM) of proliferation index of rodent *Lgr5*[+CBC] or human *LGR5*[+CBC] cells across different ages in NMRs (*n* = 3 animals/age group) (left), mice (*n* = 3 animals/age group) (middle) and humans (*n* = 4 individuals in 28 to 33-years-old age group, *n* = 5 in 65 to 70-years-old and *n* = 5 in 71 to 74-years-old group) (right). *P*-values from two-tailed Wilcoxon rank-sum tests are indicated on the graphs. **c** Left, allometric regression of division rates of colonic ASCs at the crypt base (*Lgr5*[+CBC] or *LGR5*[+CBC]) on lifespan. Mean division

rates of small intestinal *Lgr5*[+CBC] cells in mice and NMRs (black triangles) are also shown but have not been included in the least-square fit. Right, regression of division rates of colonic *Lgr5*[+] or *LGR5*[+] cells located above the crypt base on lifespan. Mean division rates of small intestinal *Lgr5*[+] above crypt base cells in mice and NMRs (black triangles) are also shown but have not been included in the least-square fit. Red triangles represent mean colonic *Lgr5*[+] cell division rates (*n* = 3 animals/species) or *LGR5*[+] cell division rates in humans (*n* = 2 individuals taken from Ishikawa et al.[31]).The shaded area represents the 95% confidence interval of the regression line. *P*-values and $R^2$ by two-tailed *F*-tests and correlation coefficients (*r*) are indicated on the graphs. **d** Allometric regression of substitution rates[32] on cell division rates of colonic ASCs (*Lgr5*[+CBC] or *LGR5*[+CBC]) cells. Red triangles indicate mean substitution rates taken from Cagan et al.[32] and the shaded area represents 95% confidence interval of the regression line. *P*-values and $R^2$ by two-tailed *F*-test and correlation coefficients (*r*) are indicated.

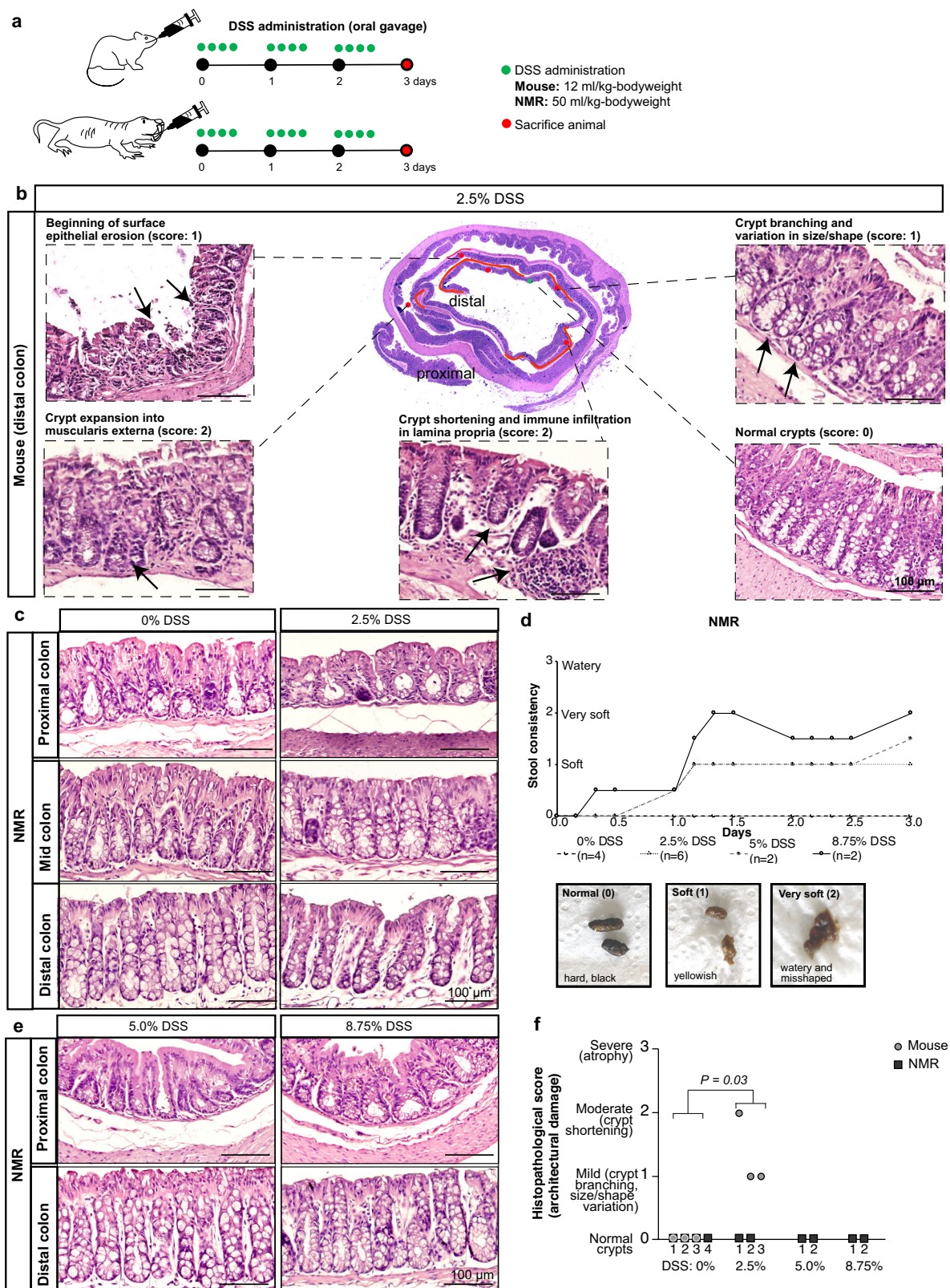

remarkable animals have evolved unique adaptations allowing for long-term maintenance of tissue homeostasis and, as a secondary consequence, lowering the incidence of ageing-related diseases such as cancers. The evolution of a larger reserve of ASCs across tissue types in NMRs would facilitate efficient tissue maintenance in an environment of high oxidative[77] and mechanical stress[78], and lower the probability of deleterious mutations becoming fixed due to

increased selection against deleterious variants[79–81] and slow down clonal expansion seen in ageing[82]. The slower ASC division rates in the NMR intestine, like in humans[31], likely prevent proliferative exhaustion of ASCs required for longer lifespans. Some tissues of the NMR produce higher amounts of reactive oxygen species from cytoplasmic and mitochondrial sources compared to mice[83]. The duration of the arrest in G1 and G2 has been shown to be linearly

**Fig. 6 | Histopathological assessment of NMR intestine after dextran sodium sulphate (DSS) treatment. a** Schema illustrating the experimental design for administering DSS via oral gavage at specific time points (green dots) and days of tissue harvest (red dots) in C57BL/6J mice (top) and wild-caught NMRs (bottom). **b** Centre, representative colonic gut roll of 2.5% DSS-treated mouse, with 30% of distal colon showing damage which is demarcated in red. Magnified images highlighting epithelial surface erosion, crypt damage and inflammation (black arrows) in DSS-treated mice. Undamaged crypts are also shown. **c** Haematoxylin and eosin stained images of the proximal, mid, and distal colons comparing control and 2.5% DSS treated NMRs. **d** Top, gradual change over 3 days in the stool consistency of NMRs subjected to DSS treatment at different concentrations [0% ($n = 4$ animals), 2.5% ($n = 6$ animals), 5.0% ($n = 2$ animals) and 8.75% ($n = 2$ animals)]. Bottom, photographs show three different stool consistencies observed in DSS-treated NMRs (hard, black: Score 0; soft and yellowish: Score 1; very soft, misshaped and slightly watery: Score 2). **e** Haematoxylin and eosin staining showing no damage in the proximal and distal colons of NMRs treated with 5.0% and 8.75% DSS. **f** Histopathological scores showing mild to moderate damage in DSS-treated mice ($n = 3$ animals per group, $P$-value = 0.03, two-tailed Wilcoxon rank-sum test) while no damage was seen in any of the DSS-treated NMRs (2.5%, 5% and 8.75%; $n = 2$ animals per treatment group). Scale bars are indicated on the images (100 μm).

correlated with the amount of DNA damage[22]. Therefore, the prolonged G1 and/or G2 phases of the cell cycle in NMR ASCs may reflect an adaptive mechanism that increases the time allocated for redox repair against a backdrop of high oxidative stress.

Altogether, our study provides a mechanistic insight into Kirkwood's disposal soma theory[84,85] which has long posited that species, like mice, with high extrinsic sources of mortality, invest energy in reproductive cell lineages rather than in optimising somatic cell maintenance by repair mechanisms like DNA repair and antioxidant systems. NMRs, like humans, have lower extrinsic threats and are able to reproduce over a longer period of time and, by evolutionary optimisation, have allocated more resources to long-term tissue maintenance, some of which as we show include an expanded pool of ASCs, slowing down of ASC division rates and highly efficient intracellular mechanisms to detect and respond to environmental change in the intestinal lumen.

## Methods

### Ethics
This study involved undertaking animal procedures in four different countries: U.K, USA, Austria, and the Republic of South Africa. Animal procedures were carried out in accordance with Home Office, UK regulations and the Animals (Scientific Procedures) Act, 1986 of UK, the Institutional Animal Care and Use Committee (IACUC) of USA, Act 7, 1991 of South Africa, and the Directive 2010/63/EU of the European Parliament.

Normal human colonoscopy samples were collected under the research tissue bank ethics 16/YH/0247 supported by NIHR Biomedical Research Centre, Oxford, U.K. and under the London Dulwich Research Ethics Committee (reference number 15/LO/1998). Written informed consent was obtained from all participants undergoing routine bowel cancer or IBD screening. All samples were anonymized.

### Mouse husbandry
Wild-caught mice (F1) were acquired from a founder population trapped in lower Austria and Vienna (2016) and housed at the Konrad Lorenz Institute of Ethology, University of Vienna, Austria. All C57BL/6J mice used in this study were purchased from Charles River (Kent, UK) or the Jackson Laboratory (USA) and housed at Biomedical Services Unit in John Radcliffe Hospital, Oxford, UK or at Rutgers University Animal Facility in Newark, New Jersey, USA. Mice were housed in individually ventilated cages under specific pathogen-free conditions and maintained at 19–23 °C temperature with 45-65% relative humidity, in an alternating 12-h light/12-h dark cycles and fed with food and water *ad libitum*.

### Naked mole rat husbandry
Naked mole rats (NMRs) were housed at the Animal Facility of the Department of Zoology and Entomology, University of Pretoria. The NMRs were kept in tunnel systems consisting of several Perspex chambers containing wood shavings as nestling material. The NMR room was maintained at temperatures ranging between 29–32 °C, with relative humidity around 40-60%. NMRs were fed chopped fresh fruits and vegetables (apple, sweet potato, cucumber, and capsicum)

daily *ad libitum* along with weekly supplement of ProNutro (Bokomo). Since NMRs obtain all their necessary water from food sources, no drinking water was provided to the animals. All scientific procedures on NMRs were conducted under ethics approval (NAS046-19 and NAS289-2020) by the Animal Ethics Committee, University of Pretoria. In addition, DAFF section 20 approval was granted (SDAH-Epi-20111909592).

For all analyses, both male and female mice, NMRs, and humans were included in the study.

### In vivo administration of BrdU and EdU
15 mg/mL solution of BrdU (5-bromo-2′-deoxyuridine, Abcam, ab142567) and 12.3 mg/mL solution of EdU (5-ethynyl-2-deoxyuridine, Merck, 900584) were prepared in sterile 1× PBS (Gibco, 10010023) and filtered through a 0.2 μm strainer. Using a 27-gauge needle and 1 mL syringe, 100 mg per kg bodyweight BrdU and 82.14 mg per kg bodyweight EdU were administered intraperitoneally. Animals were checked regularly for signs of discomfort (hunched back, shivering, low mobility) after the injection.

For cumulative labelling protocol using BrdU, the first injection in naked mole rats was administered between 14:00 to 15:00. Subsequent BrdU injections were given every 8 h for a duration of 5 days and intestinal tissues were collected every 8 h after the first injection. In C57BL/6J mice, the first BrdU injection was also given between 14:00 to 15:00, with further injections given every 6 h for a total of 2.25 days. Mouse intestinal tissues were collected 1 h after each injection. The rationale for the frequency and total number of injections in the two species is discussed in Supplementary Note 1.

### In vivo administration of dextran sulphate sodium
Dextran sulphate sodium (DSS) salt (Merck, 42867) was dissolved in sterile ddH$_2$O to prepare 0 to 8.75% (w/V) solution. Using a 2 mL syringe fitted with a plastic feeding tube (Prime Bioscience, FTP-20-38), 50 mL per kg bodyweight of DSS solution in NMRs or 12 mL per kg bodyweight in mice was administered orally at specific intervals for 3 days. Body mass was monitored daily and stool samples collected while animals were also checked for signs of discomfort (e.g. hunched back, shivering, low mobility) every 3 h.

### Intestinal tissue collection and processing
After sacrificing the animals by approved procedures, the intestine was immediately isolated from the abdominal cavity and fatty tissue was removed. The small intestine was then divided into three equal sections: SB1 (duodenum), SB2 (jejunum) and SB3 (ileum). All three parts of the small intestine and colon were then flushed with 1× PBS (Phosphate Buffered Saline) solution using a P1000 pipette to clean all the faecal material. Each tissue section was then cut open longitudinally using a gut cutting device[86] and the edges pinned down onto a 3MM filter paper such that the luminal side was facing upward. The tissue was then fixed in 10% neutral buffered formalin overnight at room temperature. The following day fixed intestinal tissues were rolled using the Swiss-rolling technique[87] and stored in 70% ethanol at 4 °C. Next, formalin-fixed Swiss-rolls were dehydrated through increasing concentrations of ethanol, cleared through xylene, and embedded in

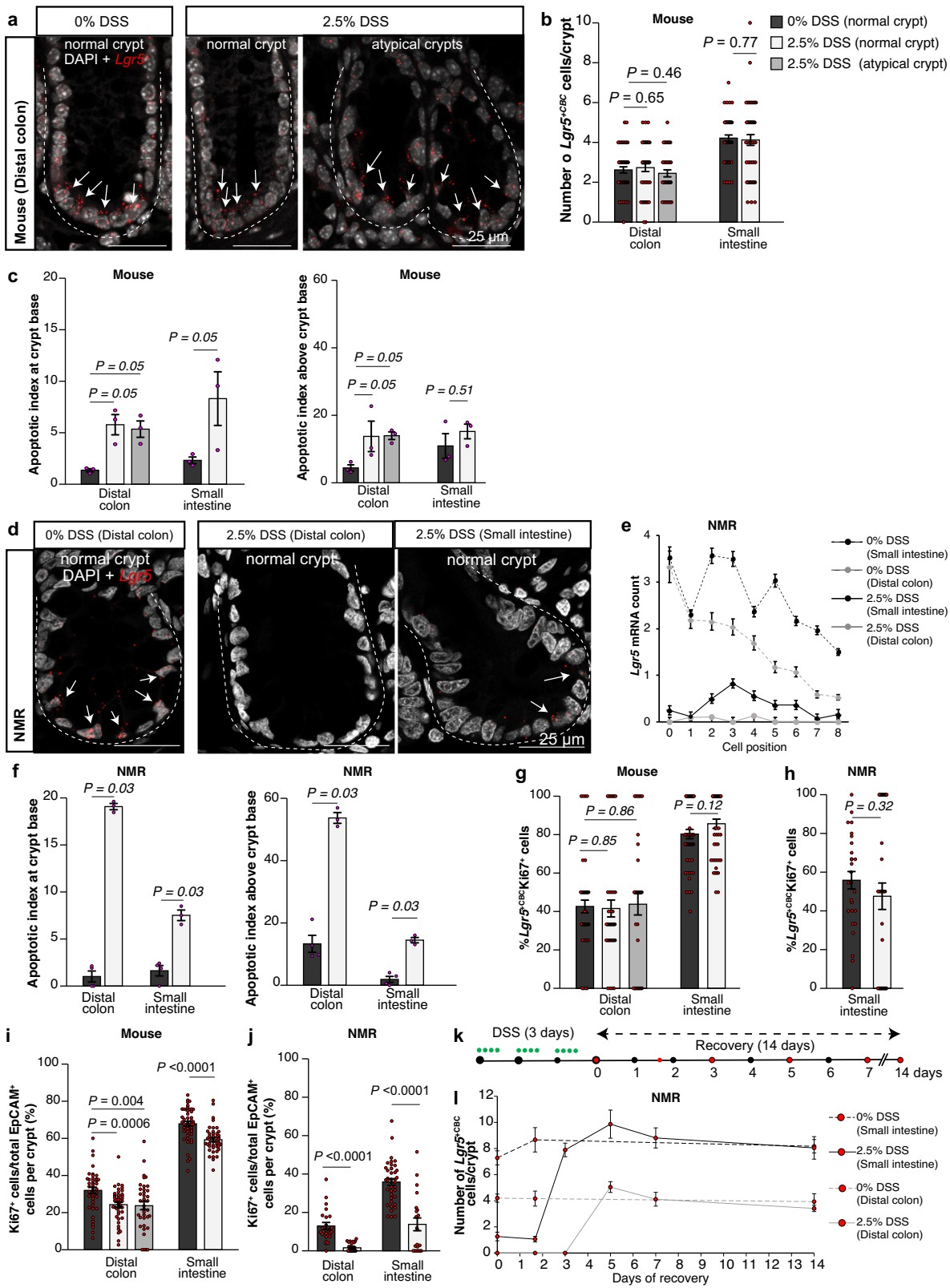

paraffin. The paraffin blocks were sectioned at 4 μm thickness using a microtome (Anglia Scientific).

## Haematoxylin and Eosin staining
Tissue sections on SuperFrost Plus slides (VWR, 6310108) were deparaffinized by submerging slides in xylene (2 times, 10 min each) and rehydrated in 100% ethanol (2 times, 5 min each), 95% ethanol

(2 min), 70% ethanol (2 min), 50% ethanol (2 min), and distilled water (5 min). Sections were then stained with Harris Haematoxylin (Merck, HHS32) for 2 min 45 s followed by washing in running tap water for 5 min. Next, slides were dipped in 95% ethanol ten times before sections were counter-stained with Eosin solution (Merck, 117081) for 3 min. This was followed by tissue sections being dehydrated in 95% ethanol (15 s) and 100% ethanol (2 times, 15 s each), dipped in xylene (2

**Fig. 7 | Response of NMR intestinal crypt cells to DSS. a** Confocal images showing *Lgr5* ISH (red) and DAPI (grey) staining in control (0%) and 2.5% DSS-treated murine colonic crypts. White arrows indicate *Lgr5*[+CBC] cells. **b** Bar graphs showing the mean (±SEM) of *Lgr5*[+CBC] cells per crypt between control and DSS-treated mice ($n = 40$ crypts counted from 3 animals/group). *P*-values from *t*-tests are indicated. **c** Bar graphs showing the mean (±SEM) TUNEL[+] cells per 100 crypts (apoptotic index) at the crypt base (left) and above crypt base (right) in mice ($n = 3$ animals/group). *P*-values from the two-tailed Wilcoxon rank-sum test are given. **d** Confocal images showing *Lgr5* ISH (red) and DAPI (grey) in NMR colonic and duodenal crypts. White arrows indicate *Lgr5*[+CBC] cells. **e** Scatter graph showing mean (±SEM) *Lgr5* mRNA expression levels at specific cell positions along the crypt axis in control and DSS-treated NMRs ($n = 50$ crypts counted from 3 animals/group). **f** The mean (±SEM) TUNEL[+] cells per 100 crypts at the crypt base (left) and above crypt base (right) in NMRs ($n = 4$ animals analysed in 0% and 3 animals in 2.5% DSS-treated group). *P*-

values generated from the two-tailed Wilcoxon rank-sum test are indicated. **g, h** Bar graphs showing mean percentage (±SEM) of Ki67-positive cells in *Lgr5*[+CBC] cells per crypt in (**g**), mouse colon and duodenum ($n = 40$ crypts from 3 animals/group) and in (**h**), NMR duodenum ($n = 30$ crypts from 3 animals/group). *P*-values from *t*-tests are indicated. **i, j** Bar graph showing mean percentage (±SEM) of Ki67[+] cells in all epithelial cells in the intestinal crypts of control and treated (**i**) mice ($n = 40$ crypts from 3 animals/group) and (**j**) NMRs ($n = 30$ crypts from 3 animals/group). *P*-values from *t*-tests are indicated. **k** Schema showing the recovery period in NMRs after 2.5% DSS treatment (green) and times of tissue collection (red). **l** Line graph showing the mean number (±SEM) of *Lgr5*[+CBC] cells in the duodenum (black) and colon (grey) of NMRs after DSS withdrawal ($n = 20$ crypts from 1 animal at each time-point). Where indicated, Student's *t*-test using two-tailed, unpaired and unequal variance was employed. Scale bars are indicated on the images (25 μm).

times, 5 min each), and finally coverslipped using DPX Mountant (Merck, 06522).

## Alcian blue staining

Tissue sections on SuperFrost Plus slides (VWR, 6310108) were first deparaffinized with xylene (2 times, 5 min each). They were rehydrated in 100%, 90%, 70% ethanol (5 min each) and tap water (2 min), dipped in 3% acetic acid solution (3 min) before staining with Alcian blue 8GX (Merck, A5268) solution (pH 2.5) for 30 min. Tissue sections were then washed (5 min) in running tap water and counterstained (5 min) with Nuclear Fast Red (Merck, N3020). After 1 min wash in running tap water again, tissue sections were dehydrated in ethanol, dipped in xylene and finally coverslipped using DPX Mountant (Merck, 06522).

## Measuring the thickness of the mucus layer in the colon

To preserve the mucus layer of the colonic epithelium, contact with any aqueous solution was avoided after the excision of the intestinal tissue. Without removing the faecal matter, several segments of the colon were cut using a scalpel and fixed overnight at room temperature in methacran/Carnoy's solution which was composed of 60% methanol, 30% chloroform, and 10% glacial acetic acid. On the second day, fixed tissues were processed in 100% methanol (2 times, 30 min each), 100% ethanol (3 times, 60 min each) and xylene (2 times, 60 min each). Processed tissues were embedded in paraffin and 4 μm thick sections cut and stained with Alcian blue as described above. Stained tissues were photomicrographed at 60× magnification on an Olympus BX51 brightfield microscope. For both NMRs and mice, 30 independent measurements of the mucus layer were taken from 3 animals using the 'measure' tool in Fiji package[88].

## Alkaline phosphatase staining

Tissue sections on SuperFrost Plus slides (VWR, 6310108) were deparaffinized in xylene (2 times, 5 min each) and rehydrated in 100%, 90%, 70% ethanol (5 min each) and distilled water (5 min). A hydrophobic barrier was drawn around the tissue sections using a PAP pen (Vector Lab, H-4000) before incubating in the AB solution (AP Staining kit, SystemBio, AP100B-1) for 20 min at room temperature in the dark. All sections were then washed in 1× PBS (5 min, on a shaker), counterstained with Nuclear Fast Red (5 min), washed in running tap water (1 min), dehydrated in ethanol, dipped in xylene and finally coverslipped with DPX Mountant (Merck, 06522).

## Immunohistochemistry

4 μm thick formalin-fixed paraffin-embedded (FFPE) sections were cut using a microtome and dried overnight on SuperFrost Plus slides (VWR, 6310108). Tissue sections were baked at 60 °C for 1 h the next day, deparaffinized in 3 rounds of xylene (5 min each) and rehydrated in 100%, 90%, 70% ethanol and distilled $H_2O$ (5 min each). Endogenous peroxidase activity was quenched by incubating sections in 3% $H_2O_2$ (Merck, 8222871000) for 20 min. A heat mediated antigen retrieval

was performed by boiling sections in 10 mM sodium citrate buffer (pH 6.0) for 10 min which was followed by 20 min of cooling down in the same solution. This was followed by incubating the tissue sections in 1× PBSTX (0.1% Triton X) for 10 min. All sections were then blocked for 1 h at room temperature using 5% serum which matched the species of the secondary antibody. Next, primary antibodies were diluted in antibody diluent (1% BSA dissolved in 1× PBS) which was applied to the tissue sections and incubated overnight at 4 °C. The primary antibodies used in this study were Chromogranin A (Abcam, ab15160) at 1:2000 and BrdU (Abcam, ab6326) at 1:500. It is noteworthy that in our BrdU staining, we did not use HCl-mediated DNA denaturation and only performed heat-mediated antigen retrieval (98-100 °C) which has been shown to produce a brighter signal than acid hydrolysis[89]. After 3 rounds of washes (5 min each) with 1× PBST (0.1% Tween20 in 1× PBS), tissue sections were then incubated for 1 h at room temperature with biotinylated secondary antibodies diluted at 1:300. For our study specifically, we used goat anti-rabbit IgG (Vector Laboratories, BA-1000) and goat anti-rat IgG (Abcam, ab207997). To detect the biotinylated target, we used the Avidin/Biotinylated enzyme Complex (ABC) kit (Vector Laboratories, PK-6101) and developed the signal using the DAB (3,3'-diaminobenzidine) solution (R&D systems, 4800-30-07). The tissue sections were then counterstained with Harris Haematoxylin (Merck, HHS32) for 5 s, dehydrated in 70%, 90% and 100% ethanol for 15 s each, dipped in xylene and coverslipped using DPX Mountant (Merck, 06522).

## mRNA ISH

Species-specific RNAscope probes from ACD Bio-techne were used to detect *Lgr5* mRNA expression in NMR (584631), mouse (312171) and human (311021) intestinal tissues. We used the RNAscope Multiplex Fluorescent Detection Kit v2 (ACD Bio-techne, 323110) and followed the instructions of the manufacturer (document number 323100-USM, ACD Bio-techne) to detect *Lgr5* mRNA targets at a single cell level in FFPE tissue sections mounted on SuperFrost Plus slides (VWR, 6310108).

## Multiplex mRNA ISH with immunofluorescence

To enable multiplexing of mRNA and proteins, we adapted the manufacturer's instructions (document number 323100-USM, ACD Bio-techne) for RNAscope Multiplex Fluorescent Detection Kit v2 (ACD Bio-techne, 323110) to exclude the step involving protease treatment. Once the mRNA signal was developed, we proceeded to detect proteins by first washing tissue sections (2 times, 2 min each) in 1× TBST (0.1% Tween20 in 1× Tris-buffered saline). This was followed by blocking for 1 h at room temperature with 10% serum which matched the species of the secondary antibodies. Multiple primary antibodies (diluted in 1% BSA in 1× TBS) were then applied to the tissue sections and incubated overnight at 4 °C. The dilutions of various primary antibodies used in our study were 1:500 for EpCAM (Abcam, ab71916), 1:500 for Ki67 (Cell Signaling, 12202), 1:200 for

p27$^{Kip1}$ (Cell Signaling, 3686 and 2552), 1:500 for BrdU (Abcam, ab6326) and 1:2000 for PHH3-S28 (Abcam, ab32388). Following primary antibody incubation, the next day we washed the sections thrice in 1× TBST (5 min each) before incubating them with fluorophore-linked secondary antibodies (at 1:500 dilution) for 1 h at room temperature. Fluorescent secondary antibodies used in our study included goat anti-rabbit Alexa 488 (Invitrogen, A11008), goat anti-rat Alexa 488 (Invitrogen, A11006), goat anti-rabbit Alexa 555 (Invitrogen, A21428) and goat anti-rabbit Alexa 633 (Invitrogen, A21070). Following the secondary antibody incubation, tissue sections were washed three times in 1× TBST (5 min each) and counterstained with DAPI (Invitrogen, D1306) for 15 min at room temperature before mounting with coverslips (VWR, 631-0138) using Diamond Antifade Mountant (Invitrogen, P36961).

### TUNEL assay
Click-iT™ Plus TUNEL Assay Kit (Invitrogen, C10617) was used following the manufacturer's instructions to detect apoptotic cells FFPE tissue sections.

### EdU detection
EdU-Click 488 kit (Base Click, BCK-EdU488-1) was used according to the instructions provided by the manufacturer to detect EdU-positive cells in FFPE tissue sections.

### Measuring plasma BrdU concentration
Plasma BrdU concentration was determined following the protocol described by Barker et al.[90]. In brief, 100 μL naked mole rat blood was collected by a tail vein puncture after 8 h and 16 h of BrdU injection. The blood was mixed with heparin to stop clotting and centrifuged at 13,000$g$ for 15 min to separate all blood cells. Plasma was collected from the top layer and stored at −80 °C.

HEK293T cells (ATCC, CRL-3216) were cultured in high-glucose DMEM (Merck, D6546) containing 10% FBS (Gibco, 10270), 1× Penicillin-Streptomycin (Merck, P4333-100ML), and 2 mM L-glutamine (Gibco, 25030-024) at 37 °C with 5% $CO_2$. Cells were plated on a 13 mm sterile glass coverslip precoated with poly L-lysine (VWR, 631-0149) in a 24-well plate (Starlab, CC7682-7524) and cultured overnight. The media was replaced with 500 μL fresh culture media containing 10 μL plasma or standard BrdU solution (3, 10, 20, 30, 40, 50 μg/ml) and incubated at 37 °C for 4 h. Cells were then washed with 1× PBS and fixed in 4% paraformaldehyde for 20 min at room temperature. Fixed cells were kept in 1× PBS at 4 °C before proceeding to immunocytochemical detection of BrdU.

Fixed cells on coverslips in 24 well plates were incubated with 3% $H_2O_2$ for 10 min at room temperature. After washing with 1× PBS, cells were incubated in 2 N HCl for 1 h at room temperature to denature DNA strands. Fixed cells were then incubated in 0.1 M Borate buffer (pH 8.5) for 30 min at room temperature and in 1× PBSTX (0.1% Triton X) for 10 min. Cells were blocked with 5% goat serum for 1 h at room temperature and incubated with rat anti-BrdU primary antibody (Abcam, ab6326, 1:2000) overnight at 4 °C. The next day, cells were washed three times in 1× PBST and incubated with goat anti-rat-biotin-linked secondary antibody (Abcam, ab207997, 1:400) for 1 h at room temperature. The biotinylated signal was developed using the ABC Kit (Vectastain, PK-6101) following the manufacturer's instructions and detected with DAB solution (R&D systems, 4800-30-07). Gill's No. 3 Haematoxylin (Merck, GHS316-500ML) was used for counterstaining and cells on the coverslips were mounted on glass slides using Aquatex mounting agent (Merck, 108562).

### Crypt-villous isolation and qRT-PCR
Intestinal tissue was washed with PBS, cut open longitudinally and laid flat on a glass slide with the luminal side facing upward. The small intestinal villi were scrapped off the flat tissue by a glass slide and collected in cold 1× PBS. The remaining tissue containing crypts was chopped into <2 mm pieces using a scalpel, washed three times with ice-cold 1× PBS and incubated in chelation medium (2 mM EDTA in 1× PBS without Ca$^{2+}$ and Mg$^{2+}$, Gibco 10010023) for 40 min with agitation at 4 °C. The digested tissue was shaken vigorously for 30 s in 1× PBS to release crypts and villi. To separate out crypts and villi of the small intestine, the solution was passed through a 100 μm cell strainer. The isolated crypts in the flow through were pelleted and transferred to RLT Buffer (Qiagen, 79216). RNeasy microkit (Qiagen, 74004) was used for RNA extraction. Extracted RNAs were incubated with DNase1 (ThermoFisher, EN0521) at 37 °C for 30 min, followed by a 10 min incubation with EDTA at 65 °C. High-Capacity cDNA Reverse Transcription Kit (Applied Biosystems, 4368814) was used to generate complementary DNA from total RNA. Quantitative real-time-PCR (qRT-PCR) was performed on LightCycler96 (Roche) with mouse and naked mole rat Gapdh used as an endogenous control. The IDs of Taqman Gene expression assays (Applied Biosystems) used in this study are *Gapdh* (Mm99999915_g1, Hg05064520_gH), *Muc2* (Mm01276681_m1, Hg05250665_g1), *Synaptophysin* (Mm00436850_m1, Hg05249763_m1), and *Aldolase B* (Mm00523293_m1, Hg05103981_m1). The 2$^{-\Delta\Delta Ct}$ method was used to calculate the relative gene expression levels.

### Brightfield microscopy
Brightfield images of tissue sections were captured using an Olympus BX51 microscope coupled with an Olympus DP70 camera system using DP controller software. Villi were imaged using 10× objective while crypts were imaged with 20× (for colon) or 60× (for small intestine) objective lens. Histopathological scoring in this study was performed based on the digital images obtained on Hamamatsu (Nanozoomer HT) scanner at 40× magnification.

### Histological quantification (brightfield)
To quantify cell numbers in crypt-villous structures from brightfield images, 'cell counter' plugin of Fiji software was used. The dimensions of crypt-villous structure were calculated using the 'measure' tool in Fiji.

### Fluorescent microscopy
Fluorescent images of intestinal crypts were acquired from 4 μm thick tissue sections with a Plan Apochromat 63× or 100× 1.4 oil objective on a Zeiss LSM 780 upright or inverted confocal microscope. Images were acquired in Zen SP7 FP3 (black) software using 405 nm, 488 nm, 561 nm, and 633 nm laser lines in sequential tracks. Z-stacks of 6-12 optical sections with 50% overlap between subsequent planes were captured within the span of a single cell at 0.3 μm z-distance, 0.087 μm pixel dimension, and 12-bit depth.

For generating the RGB images used in the figures (Figs. 1a, b, 2a–d, 3d, 4d, 7a, d, Supplementary Figs. 1–4, 5d, e, 11b), the original.czi raw files were imported into Fiji software package and a maximum intensity z-projection was created from the stacks. Using the "split channel" option of Fiji, the multicolour fluorescent images were separated into individual channels (DAPI, Alexa 488, Cy3, Alexa 633). The maximum and minimum displayed pixel values of individual channels were adjusted across the entire image set including in negative controls (i.e. linear adjustment) to correct for autofluorescence that had been introduced in the image stacks during acquisition. Then, using "merge channel" option in Fiji, two/more channels were combined to create a composite image (*Lgr5*/Ki67 or *LGR5*/KI67, *Lgr5*/EpCAM or *LGR5*/EPCAM, *Lgr5*/p27 or *LGR5*/P27, *Lgr5*/BrdU, *Lgr5*/pHH3 or *LGR5*/PHH3) while keeping the individual channels intact. Finally, all the individual and composite images were converted into 'RGB color type' and saved in TIFF format. These images (TIFF) were compiled in Adobe Illustrator 2020 software to produce the panels presented in the figures.

## Histological quantification (fluorescent)

Z-stack images were processed in batch mode of Fiji package. Firstly, a maximum intensity projection was created to generate a 2D image from the stacks. Next, each channel of the image was separated, and maximum and minimum displayed pixel values were adjusted across the entire image set including negative controls. To quantify the number of rodent *Lgr5* or human *LGR5* mRNA expressed in a single cell, all the ISH dots were manually counted within the cell periphery demarcated by EpCAM staining. As the *Lgr5* or *LGR5* signal was captured using confocal microscopy at a resolution of 237 nm, overlapping/merged *Lgr5* or *LGR5* mRNA signal dots were rarely observed. To calculate the distribution of *Lgr5*⁺ or *LGR5*⁺ cells relative to other cells along the crypt axis, the cell present at the crypt apex was assigned position 0 and we counted cells on each side of this cell to acquire datapoints in our quantifications. Any cell containing more than three *Lgr5* or *LGR5* mRNA puncta was considered positive for *Lgr5* or *LGR5* expression (*Lgr5*⁺ or *LGR5*⁺).

We observed significant variation in autofluorescence levels between mouse, human and NMR intestinal tissues, with mouse tissue emitting the most and naked mole rats the least. This variation necessitated adjusting the laser powers of the confocal microscope during image acquisition so that maximal image contrast was achieved while also reducing the autofluorescence signals. The maximum and minimum displayed pixel values of individual channels were adjusted across the entire image set (i.e. linear adjustment), including in negative controls, to correct for autofluorescence. These adjustments resulted in varying intensities for specific signals in the three species and, therefore, we took a binary approach for the quantification of the antibody-based signals. The presence of any specific signal in the target compartment inside a cell was considered positive regardless of the staining intensity.

## Estimating the length of the total cell cycle (T$_T$) and S-phase (T$_S$) by cumulative labelling with BrdU

We determined the length of the cell cycle (T$_T$) and S-phase (T$_S$) in CBC cells (*Lgr5*$^{+CBC}$) of naked mole rats by counting the fraction of BrdU-labelled *Lgr5*$^{+CBC}$ cells after successive pulsing over 5 days in NMRs and 2.25 days in mice. As the CBC cells (*Lgr5*$^{+CBC}$) cells are on average asynchronously and asymmetrically dividing[45], the labelling index (LI) which provides the ratio of labelled cells to the total population (LI = *Lgr5*$^{+CBC}$BrdU⁺/*Lgr5*$^{+CBC}$) at any given time (*t*) can be modelled by Eq. 1 below where T$_T$ is the total cell division time[33].

$$LI = (1/T_T)X\,t + (T_S/T_T), \text{for } t\,T_T - T_S$$
$$LI = 1 \text{ for } t > T_T - T_S \tag{1}$$

Equation 1 assumes that there are no or only very few stem cells (based on p27 negativity in NMR and mouse *Lgr5*$^{+CBC}$ cells) that remain quiescent for the duration of the BrdU experiment. The *lfit* tool in STATA was used to calculate the least square fit of the data by considering the time points before LI reached saturation. We derived T$_T$ from the slope of the regression (T$_T$ = 1/slope). When *t* = 0, LI$_0$ = T$_S$/T$_T$ which is the y-intercept of the graph. Thus, the duration of S-phase (T$_S$) was estimated from the y-intercept of the regression line.

## Estimating the duration of the specific cell cycle phases

For human *LGR5*$^{+CBC}$ cells, we assumed KI67 is undetectable at G1/S transition and detected in the S to M phases of the cell cycle[46]. We determined the fraction of *LGR5*$^{+CBC}$ cells that expressed KI67 and calculated the length of S, G2 and M-phase (T$_{(S, G2, M)}$) using Eq. 2:

$$T_{(S,G2,M)}\text{KI67}^+ = T_T^{(Ref\,31)}X\,LGR5^{+CBC}\text{KI67}^+/LGR5^{+CBC} \tag{2}$$

The time in mitosis (T$_M$) was calculated after quantifying the fraction of rodent (mouse or NMR) *Lgr5*$^{+CBC}$ or human *LGR5*$^{+CBC}$ cells

positive for phospho-histone H3 using Eq. 3:

$$TM^{\text{KI67}+} = T_T^{(linear\,regression)}X\,Lgr5^{+CBC}\text{pHH3}+(\text{Ser28})/Lgr5^{+CBC} \tag{3}$$

or

$$T_M^{\text{KI67}+} = T_T(\text{ref31})X\,LGR5^{+CBC}\text{PHH3}^+(\text{Ser28})/LGR5^{+CBC}$$

Using T$_S$ estimated by Ishikawa et al.[31] previously, the length of G2-phase (T$_{G2}$) was calculated using Eq. 4:

$$T_{G2}^{\text{KI67}+} = T_{(S,G2,M)}\text{KI67}^+ - \left(T_S^{\text{KI67}+} + T_M^{\text{KI67}+}\right) \tag{4}$$

After quantifying the fraction of *LGR5*$^{+CBC}$ cells expressing P27, we calculated the time spent in G0 and G1 (T$_{(G1, G0)}^{P27+}$) using Eq. 5:

$$T_{(G1,G0)}^{P27+} = T_T(\text{ref31})X\,LGR5^{+CBC}\text{P27}+/LGR5^{+CBC} \tag{5}$$

We took the fraction of LGR5⁺P27⁺ cells in G0 phase (QF) from Ishikawa et al.[31] to calculate the length of G0 in human *LGR5*$^{+CBC}$ cells using Eq. 6:

$$T_{G0}^{P27+} = QF(\text{ref31})X\,T_{(G1,G0)}^{P27+} \tag{6}$$

Finally, using Eq. 7, we quantified the time human colonic *LGR5*$^{+CBC}$ cells spend in G1 (T$_{G1}$):

$$T_T = T_{G0}^{P27+} + T_{G1}^{P27+} + T_S^{\text{KI67}+} + T_{G2}^{\text{KI67}+} + T_M^{\text{KI67}+} \tag{7}$$

In NMR and mouse, *Lgr5*$^{+CBC}$ cells are negative for p27 such that T$_{G0}$ = 0. For these species, we derived the combined length of time spent in G1 and G2 (T$_{G1+}$T$_{G2}$) from Eq. 7.

## Estimating cell division time of *Lgr5*$^{+\text{above crypt base}}$ cells from single time point analysis of BrdU labelling

Using the length of T$_S$ from cumulative BrdU labelling in *Lgr5*$^{+CBC}$ cells and assuming no change in T$_S$ in *Lgr5*⁺ cells located at different positions within the crypt[31], we measured the total cell division time (T$_T$) of *Lgr5*$^{+\text{above crypt base}}$ cells using Eq. 1 by measuring the labelling index (LI) at a single time point (*t*) after pulsing animals with BrdU in vivo. More specifically, in C57BL/6 mice (*n* = 3 animals, 4 months old), we administered BrdU once and analysed intestinal tissue at *t* = 0.5 h. In NMRs (*n* = 3 animals, 6-24 months-old), we pulsed the animals with BrdU every 8 h and analysed the intestine after *t* = 1 day.

## Statistical analysis

We used Microsoft Excel (v16.77.1) for inputting raw data after collection. All statistical tests and graphs displayed in this paper were generated using StataMP 14.1. Details of statistical tests performed are described in figure legends. P-values are generated by conducting two-tailed *t*-tests, F-test and Wilcoxon rank sum test as indicated in each figure legend. No blinding and randomization were performed during the analysis.

## Illustration

All the figures presented in this manuscript were prepared using Adobe Illustrator 2020 (version 24.1). Vector line arts shown in Figs. 1c, d, 3a, h, 4a, h, 6a, Supplementary Figs. 5d, e, and 9a, b were created using the curvature tool of Adobe Illustrator.

## Reporting summary

Further information on research design is available in the Nature Portfolio Reporting Summary linked to this article.

## Data availability

Source data are provided as a Source Data file. Source data are provided with this paper.

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

## Acknowledgements

Adult human colonoscopy samples were collected by research nurses led by Yukari Kimura and James Chivenga at John Radcliffe Hospital, Oxford and Lee Meng Choong at King's College Hospital, London. We thank the patients who consented for our study. We thank Wolfson imaging facility at the Weatherall Institute of Molecular Medicine for allowing access to confocal microscopes and also for providing advice. We also thank Sheeba Irshad's laboratory for providing scanning microscopy support. We thank Annika Posautz for helping with the tissue processing of wild-caught mice and Adrian Tordiff for providing veterinary supervision of naked mole rats. We also thank Anne Goriely and Daniel Crouch for their comments on the manuscript. This research was financially supported by Cancer Research UK (Grant code: C6199/A27327, I.T.) and the South African Research Chair of Mammal Behavioural Ecology and Physiology from the DST-NRF (GUN 64756, N.C.B.). J.E.E. and S.I. were funded by the National Institute for Health Research (NIHR) Oxford Biomedical Research Centre. The views expressed are those of the authors and not necessarily those of the National Health Service, the NIHR or the Department of Health.

## Author contributions

S.I. conceived and designed the project. Animal experiments were conducted by S.M., S.B., and D.W.H. Histological staining and imaging were performed by S.M. and data analyses were performed by S.M. and S.B. Naked mole rats and ethics for in vivo work were acquired by N.C.B. and D.W.H. Ethics for DSS and BrdU experiments in C57BL/6 mice were acquired by N.G. Wild-caught mice were supplied by D.J.P. and B.W. S.G.T, B.H. and J.E.E. provided human colonoscopy samples. B.J., M.B., P.M.A. and F.R. provided statistical and mathematical input. Histopathological scoring was performed by P.G. and P.Z. Y.L., J.H. and A.T. helped with tissue processing and animal husbandry. J.K. provided guidance on confocal microscopy. RT-PCR was conducted by S.I. Intellectual input was provided by I.T., N.C.B., W.F.B., and J.E.E. The paper was written by S.I. and commented on by all other authors.

## Competing interests

M.B. is an employee of Bristol Myers Squibb, San Diego 92121, California, USA. P.M.A. is a consultant for CRISPR Therapeutics (Cambridge, MA, USA) and received funding from Kite Pharma (San Diego, CA, USA) for research unrelated to the work reported here. A.T. is an employee of Novo Nordisk Research Centre Oxford, Oxford OX3 7FZ, UK. The remaining authors declare no competing interests.
