## [Peer Review File · Nature Communications]

Adult stem cell activity in naked mole rats for long-term tissue maintenanceREVIEWER COMMENTS

Reviewer #1 (Remarks to the Author):

Montazid et al. examined the cell proliferation behavior of intestinal epithelial stem cells in naked mole rat, the longest-living rodent, and implicate slow proliferation of stem cells and long-term survival of mammals. This is an interesting area of research because of the recent focus on the importance of dormant stem cells in regeneration and cancer development. This report provides supportive data that is consistent with recently reported somatic mutation ratios in many species and the high quiescence of human colon stem cells compared to mice. In this work, which is primarily a histological study, the reviewer has several concerns, particularly regarding tissue staining, mRNA detection, and the accuracy of the counts, as follows.

Major points

1. In this paper, mathematical validation is based on data from a histological evaluation. The reliability of the original data is very important, but unfortunately, the detection of Lgr5/LGR5 mRNA in situ hybridization is not convincing (especially in Fig. 2), probably due to insufficient technical problems. The expression of mRNA is vague, and the threshold that separates expression from non-expression is not clear. Who did the counting?
2. When looking at the staining images of naked mole rat in Extended data Fig. 1 and 5, the cells at the base of the crypts are almost unstained for p27. While the authors explained this data owing to the undetectable Ki67 protein due to degradation at G1, this may indicate that p27 antibodies are not at work. In fact, the expression pattern of p27 is also different between Figure 2 and Extended data Fig. 1, and the distinction from nonspecific staining is not clear. On the Cell Signaling Technology website, this antibody (#3686) is validated for human, rat, and monkey species, but not for mouse and naked mole rat. Were the same results obtained with the other reliable p27 antibodies which were used in other literatures? On what basis do the authors claim that this staining pattern is validated?
3. Most data remain descriptive with classical experimental methods. There should be support for data from transcriptome analysis, such as single-cell RNA-seq of naked mole rat epithelial cells.
4. Even though the study was performed in vivo in naked mole rats, the study in organoids was performed only in mice. Comparative studies in mouse, naked mole rat, and human intestinal organoids are required.
5. Cell proliferation is also related to the effects of aging. Even at the same age of 12 months, the lifetimes of mice and naked mole rats differ greatly. In Fig. 5, the authors made comparisons between 6-month-old and 3-year-old naked mole rats. However, naked mole rats have a life span of >20 years. Comparisons at this slight difference in age are meaningless even if there is no difference. What about the differences in cell cycle with age in the naked mole rats more than 20 years old?

6. In humans and mice, p27 is the main player with p16 taking a minor supporting role, but in naked mole rats, the two genes have become decoupled (Proc Natl Acad Sci U S A. 2009; 106(46): 19352-7). What about p16 participation in naked mole rats?

7. The association between somatic mutation rate and longevity has already been suggested in references 4 and 5, but where is the advantage or novelty of using naked mole rat to show it in this paper? What will be changed by what this paper reveals? Please emphasize the importance of its significance.

Minor points

1. Most Bar graphs should be overlaid with dot plots.

2. The concentrations of DSS that cause enteritis vary widely depending on the batch and the environment in which the animals are housed. Weight trend graphs should also be shown to firmly indicate that DSS enteritis has occurred.

3. In Fig. 4j, human and naked mole rat as well as mouse data should be included in the comparison.

4. The cell cycle varies with time of day. Consistency and description of sacrificed time is desirable for evaluation of proliferation.

5. Longevity of the gut is not directly related to survival. Is the proliferation of vital organs other than the intestine similarly slow?

Reviewer #2 (Remarks to the Author):

This well written paper with beautiful histological images examines intestinal stem cells of the extraordinarily long-lived and cancer resistant naked mole-rat (NMRs). The authors report a considerably larger adult stem cells (ASC) pool and that the ASC of NMRs do not enter quiescence but rather show slow proliferation through the active cell cycle, facilitating better translational fidelity, DNA repair, reduced mutation rates and less propensity for ASC exhaustion. In addition, the expanded pool of differentiated cells in the intestine is posited to afford greater protection of the stem cell zone and better maintenance of tissue homeostasis even when challenged with a well-established intestinal stressor. The methodology is sound and the experiments well planned. The results are clearly articulated. The discussion of the findings is very brief and could be expanded to more accurately place these exciting data in light of what is already known about some of the unusual features of the naked mole-rat.

This original study supports prior findings of sustained tissue homeostasis and stress resilience of NMRs and provides a mechanistic explanation for their extreme longevity and cancer resilience. Several seminal NMR papers should be cited e.g., Yamamura et al., 2022 (<https://doi.org/10.1111/cas.15570>) Miyawaki et al., Nature.com/articles/ncomms11471 Miura 2021 doi: 10.1007/978-3-030-65943-1_13,

and many others

The longer proliferative cycle of NMR ASCs when compared to that of mice is similar to that observed for species differences in skin fibroblasts in culture. We question if this simply reflects a difference in body temperature/ metabolic rate or if there is another explanation? No mention is made that NMRs maintain a markedly lower body temperature than seen in mice and if this could affect the ASC kinetic profile.

The authors examine ASC division rates in 3 age groups 6m, 1y and 3 years (out of a >37 year lifespan) whereas those of both mice and humans spanned a far larger proportion of their lifespan (2m to 18m out of an average 3 year lifespan and 28-74 years in humans) .

How were these age groups chosen? Given that no species showed an age-related difference in cell proliferation rates, was this finding expected? please explain the lack of age effects in humans and mice. Can one really determine the relationship between ASC division and lifespan for the NMR given that all age cohorts examined fall within their 1st 10% of their lifespan.

Where did you get the NMR lifespan record of 25 years? This is a gross underestimate. Lee et al., 2020 doi: 10.1007/s11357-019-00150-7. published a lifespan of 37 years others as did Yamamura; others have published >31 years (e.g., AnAge website) while Ruby et al., 2018 kaplan Meier demographic analysis (<https://doi.org/10.7554/eLife.31157>) showed that the median lifespan for subordinate mole-rats was 19 years while that of dominant breeders was significantly longer.

The response to DSS showed that mice showed typical injury whereas the NMRs were resilient against such stress but ASCs stopped proliferating. A similar response has been previously reported in skin fibroblasts in response to several stressors (Lewis KN et al., 2012 doi: 10.1159/000335966). This profound resistance to stress should be more comprehensively discussed.

The discussion appears rather superficial and makes mechanistic suggestions that are not fully substantiated. The authors suggest that the harsh underground habitat of NMRs has acted as a strong selective force for a larger reserve of ASCs; I wonder if the toxic nature of the plant secondary defense mechanisms in their diet (large pyrethrium tubers) may play a role in both their larger ASC and their resilience in the face of stress. Does their microbiome influence this in anyway?

I also question the link to the disposable soma theory, since the authors make no mention of their eusocial lifestyle whereby reproduction is restricted to only a few animals with high reproductive demands, given that all animals used in this study are very young subordinates that may have not even attained adult mass, I wonder if the breeding females that have high physiological demands associated with doubling of body mass during pregnancy show similar ASC kinetics and if they would support the disposable soma theory.

Reviewer #3 (Remarks to the Author):

Montazid et al. present an analysis of intestinal stem cell (ISC) proliferation and cell cycle activity in 3 species (mouse, naked mole rat = NMR, and human). The authors employ Lgr5 as a marker and study associations between cell cycle activity, aging, and lifespan. The authors provide compelling evidence on differences in cell cycle and quiescence control of ISCs. While short lived mice have highest proliferation rates, ISCs of long-lived humans show reduced rates of cell cycle activity associating with longer lifespan in these species compared to mice. The authors also find a similar kind of correlation for the association of cell cycle activity and mutation rates in ISCs (as extracted from other publications). Interestingly, the authors also find some evidence that different species employ different mechanism of ISC cell cycle control to ensure longevity. While NMRs exhibited prolonged G1 and G2 stages to reduce cycling activity, human ISCs exhibit an increase in quiescence (G0). Moreover, the authors identify a difference in the composition of differentiated cells in the intestinal epithelium and mucosal layer thickness as well as ISC sensitivity to chemical induced toxicity in NMRs compared to mice, indicating that in addition to cell cycle control, shifts in the production of differentiated cells can also contribute to species-specific mechanisms of longevity protection of the intestine.

Overall criticism: this is an important study that provides a detailed and compelling analysis of ISC cell cycle control. The study has important implications for our understanding of somatic stem cells in organism aging. I have a few specific comments that should be addressed:

1. For some of the figures the authors use antibody staining to determine cell cycle stages or proliferation activity. The authors should discuss the possibility that species specific differences in the sensitivity of the antibodies lead to differences in the staining of cells in different species, which could have an impact on the quantification of cells being positive/negative for certain cell cycle or proliferation markers if in one species the detection limit is lower or higher compared to the other species.
2. In Figure 2 the authors use 2-3 animals per species. I would recommend to leave out p-value calculations for this experiment (and any other with <3 n-number). It is understandable that studies on NMRs cannot be conducted with too many replicates. The data show clear differences and strong effect sizes, but I would recommend to leave out the p-value calculation in these cases.
3. In figure 4J and in the description, I did not completely understand how the authors discriminate G0 from G1 cells. In the J-panel both cell populations have the same markers. Can the authors explain this a bit better?
4. At the end of Figure 6 description the authors conclude: "In summary, our results show that even without tissue damage, NMR intestine responds to low levels of chemical insults more efficiently than mouse, shutting down cellular proliferation in the crypts and triggering apoptosis in the majority of Lgr5+ cells across the entire intestine." I don't know whether this difference is indeed pointing to a better "efficiency". What would be the advantage? It may even have some unwanted side effects (see comment below).
5. Figure 6e: the photographs on Stuhl consistency are not clear. What should be seen here?
6. Figure 6i: It looks like there are Lgr5-positive cells left in position 4/5 of intestinal crypt of DSS-treated NMRs. There is evidence in mice, showing that DNA damage induced depletion is stronger at crypt base and that position-4 LGR5-positive cells can survive and appear to repopulate the crypt base. The study

showed that different levels in Wnt signalling (reflected by Lgr5 levels) positively correlate with DNA damage sensitivity (Tao S et al. 2015). This could be relevant also for the response to DSS and could be discussed.

7. Discussion, 1st paragraph: the authors should consider to re-order the discussion: it would be better to first discuss the data and then move into the more speculative role of increases in stem cell number per crypts as an anticancer mechanisms, possibly selected in this harsh environment during evolution.

8. Discussion, the authors describe: "we observe between ASC division rates and lifespan is not seen when estimates of Lgr5+/LGR5+ cells outside the crypt base are used..." I think that it is possible that these cells outside the crypt might be progenitor or ISCs that are on the way to turn into progenitors. That would explain why there is no correlation of division rates of these cells to longevity because they are short-lived.

9. Discussion on mutation rate differences, the authors speculate: "The longer G1 and/or G2 arrest of NMR ASCs compared to truncated G1 and G2, but extended G0, of human ASCs, suggests that higher damage due to increased metabolic rates at gap phases of the active cell cycle would increase the non-replicative errors in NMRs, which may partially explain the 2-fold difference in the substitution rates in the NMR colonic crypt cells compared to human counterparts". While this may play role, an alternative explanation may be that the increased sensitivity of NMR ISCs to damage (DSS model) and the subsequent regeneration of ISCs from position 4/5 ISCs and/or progenitors may come at costs of increasing mutation burden in NMRs during lifetime.

Reviewer: Lenhard Rudolph

Reviewer #1

Montazid et al. examined the cell proliferation behavior of intestinal epithelial stem cells in naked mole rat, the longest-living rodent, and implicate slow proliferation of stem cells and long-term survival of mammals. This is an interesting area of research because of the recent focus on the importance of dormant stem cells in regeneration and cancer development. This report provides supportive data that is consistent with recently reported somatic mutation ratios in many species and the high quiescence of human colon stem cells compared to mice. In this work, which is primarily a histological study, the reviewer has several concerns, particularly regarding tissue staining, mRNA detection, and the accuracy of the counts, as follows.

Major points

1. In this paper, mathematical validation is based on data from a histological evaluation. The reliability of the original data is very important, but unfortunately, the detection of *Lgr5/LGR5* mRNA *in situ* hybridization is not convincing (especially in Fig. 2), probably due to insufficient technical problems. The expression of mRNA is vague, and the threshold that separates expression from non-expression is not clear. Who did the counting?

Response from authors:

We would first like to thank the reviewer for highlighting the issues related to the visualization of *Lgr5/LGR5* mRNA using the *in situ* hybridization (ISH) signal in the images provided in Fig. 1 and 2. Identical fixation, permeabilization and staining protocols were used for intestinal tissues from all three species. Moreover, the mRNA probes were specifically custom-made for all three species. We have used high-resolution microscopy for acquiring images and have updated the methods section to include more details on image acquisition and processing (lines 835 to 857, resubmitted manuscript). Briefly, all *Lgr5/LGR5 in situ* hybridization images have been acquired using a Plan Apochromat 63x or 100x 1.4 oil objective on a Zeiss LSM 780 upright or inverted confocal microscope with Zen SP7 FP3 (black) software. Z-stacks of 6-12 optical sections were captured within the span of a single cell for each crypt at 0.3 μm z-distance, 0.087 μm pixel dimension and 12 bit-depth.

The issue with the visualization of the ISH signal is not technical, but rather due to 2 main factors as discussed below:

- 1) We have aimed to summarise the findings across two tissue types (small intestine and colon) and across three species (mouse, human and naked mole rat) in Fig. 1 and 2. We have had to use an image size of 33X48mm in order to fit all the images on one page. This reduction in size has mostly impacted the visualization of *Lgr5*-mRNA in mouse crypt cells (Fig. 1a, Fig. 2a) as these cells are smaller/narrower in size and express fewer mRNA transcripts per cell compared to human and naked mole rats (see Note 1 on page 5 on this document).
- 2) In Fig. 1 and 2, we have used multichannel confocal images to show the presence of *Lgr5/LGR5* mRNA in specific cell types (EPCAM⁺) at the crypt bottom, with or without Ki67 staining. The use of all 2/3 channels with DAPI staining in some images makes it harder to easily visualize all the signals in some images. This is again mostly an issue with the mouse small intestine in Fig. 1a and Fig. 2a.

To address both these issues, we have made the following changes to the manuscript:

- 1) In addition to providing a 3-colour image in Fig. 1a-b, we also now show representative digitally zoomed single channels (Alexa 488, Cy3, and Alexa 633) of the RGB images that were used for the characterization/quantification of all *Lgr5*⁺/*LGR5*⁺ cells (**new Supplementary Fig. 1, 2, and 3**).
- 2) We have replaced **Fig. 1a** (mouse small intestine) and **Fig 1b** (NMR distal colon) with other representative images where the EpCAM staining (Alexa 633) is not spatially overlapping with the *Lgr5* mRNA puncta (Cy3) so that the two signals are easily discernible.

3) For Fig 2, which describes the Ki67 and p27 expression in mice, humans and NMRs in the *Lgr5/LGR5* cells specifically at the crypt base (*Lgr5^{CBC}/LGR5^{CBC}*), we have replaced the previous images that showed the entire crypt with those that highlight the crypt base region only. This has been done by digitally zooming (2X) the crypt base which makes individual fluorophores/markers easier to visualize in composite images (new **Fig. 2a-d** resubmitted manuscript).

Examples of these changes and how they have improved the visualization are provided here (**Supplementary Fig. 1 and Updated Fig 2a**) and please also refer to the main resubmitted manuscript for **Supplementary Fig. 2 and 3**:

Supplementary Fig. 1 | Representative composite and 2X digitally zoomed-in single-channel fluorescence images of mouse small intestinal crypts. Maximum-intensity z-projections for 4 µm-thick mouse tissue sections co-stained with **a**, *Lgr5* RNAscope probe (red), anti-EpCAM antibody (yellow) and DAPI (blue). White arrows indicate *Lgr5*-expressing (*Lgr5*⁺) cells. The periphery of *Lgr5*⁺ cells (containing >3 puncta of signal representing mRNA transcripts) is outlined in white lines on each

single channel image and is based on EpCAM staining. Asterisk (*) denotes Ki67 positive cells. Scale bars are shown as 10 μm and 20 μm .

Fig. 2a (Updated) | Representative confocal images of the crypt base from mouse and NMR small intestine (duodenum) that were co-stained with *Lgr5* RNAscope probe (red), Ki67 (green) antibody and DAPI (blue). Asterisk (*) indicate *Lgr5*-expressing cells at the crypt base (*Lgr5*^{CBC}) and white arrows mark *Lgr5*^{CBC}Ki67⁺ cells.

Note 1:

We provide a quantitative analysis of the distribution of the *Lgr5* mRNA transcripts visualized as puncta in the graph below (Graph 1). Roughly 47% of mouse cells in the small bowel that were found to be positive for *Lgr5* ISH had 4 mRNA puncta (Graph 1). In contrast, the majority of naked mole rat cells positive for *Lgr5* ISH had 6 to 10 puncta per cell which manifests as a much stronger signal. Hence, the difference in the number of mRNA puncta is not due to technical issues, but because of the differences in the underlying mRNA expression levels between species.

Graph 1: Histogram plots showing percentages of total *Lgr5*⁺ cells containing variable number of *Lgr5* mRNA puncta.

The other point that the reviewer raises:

“The expression of mRNA is vague, and the threshold that separates expression from non-expression is not clear. Who did the counting?”

In the methods section, we had previously mentioned the criteria for quantifying cells that were considered positive for *Lgr5* expression as: “Any cell containing more than three *Lgr5/LGR5* mRNA puncta was considered positive for *Lgr5/LGR5* expression (*Lgr5/LGR5*⁺).” (lines 870 to 872, resubmitted manuscript). The reason for selecting this threshold was based on an in-depth analysis done at the beginning of the study where we assessed the congruence between *Lgr5* mRNA detection and protein level expression of *Lgr5* in mouse intestinal crypt cells. Using protein expression of *Lgr5*, Baker *et al.* have previously reported the presence of 3 to 4 *Lgr5*-expressing cells in the murine small intestinal crypts in a *Lgr5-lacZ* mouse model¹. We also found 3 to 4 cells positive for *Lgr5* expression (*Lgr5*⁺) in our *in situ* hybridization assay when scoring cells with >3 puncta as positive (see Graph 2 below) and, therefore, used this threshold for scoring *Lgr5*⁺ cells in our study.

Graph 2: Bar graphs showing the threshold for scoring *Lgr5*⁺ cells with varying *Lgr5* mRNA puncta counted per cell in mouse small intestinal crypts. n=138 crypts from 3 C57BL/6J mice were counted for each threshold. In all cases, statistical significances were determined by performing a two-tailed, unpaired Student's t-tests using an unequal variance. Significant P-values are denoted as such: ***<0.001. Each bar represents the mean ± standard error of the mean.

2. When looking at the staining images of naked mole rats in Extended data Fig. 1 and 5, the cells at the base of the crypts are almost unstained for p27. While the authors explained this data owing to the undetectable Ki67 protein due to degradation at G1, this may indicate that p27 antibodies are not at work. In fact, the expression pattern of p27 is also different between Figure 2 and Extended data Fig. 1, and the distinction from nonspecific staining is not clear. On the Cell Signaling Technology website, this antibody (#3686) is validated for human, rat, and monkey species, but not for mouse and naked mole rat. Were the same results obtained with the other reliable p27 antibodies which were used in other literatures? On what basis do the authors claim that this staining pattern is validated?

Response from authors:

Extended Data Fig. 1 in the previously submitted manuscript did indeed show p27/P27 staining in the intestinal tissue of the three species being examined (mouse, human and naked mole rat). However, previous Extended Fig. 5 (now called Supplementary Fig. 8 in the resubmitted manuscript) shows the characterisation of the differentiated cell types within the mouse and naked mole rat intestines. There appears to be a typo in the reviewer's comment and we believe the figures being questioned/compared were Fig. 2c and Fig. 2d with former Extended data Fig. 1 (now called Supplementary Fig. 4).

To address the reviewer's comment, **we provide 4 supporting evidence** that the antibody (Cell Signalling Technology, #3686) that was originally reported by the manufacturer to cross-react with human, rat and monkey p27 also binds specifically to mouse and naked mole rat p27 protein in most cells of the anterior prostate (positive control) and a few cells in the intestinal tissue:

1. The antigenic sequence of the #3686 antibody is considered proprietary by Cell Signaling Technology, but the technical support team from the company BLASTED this sequence (reference) against the p27 Kip1 homologous protein (query) in the naked mole rat (UniProt ID: A0A0P6JT45). The results showed 94.74% homology between the reference and query sequence. This result is identical to that generated with the rat p27 Kip1 sequence (query coverage =100%

and percentage identity = 94.74%) and based on this sequence homology results, #3686 antibody was predicted to work in rats by the manufacturer (CST). Based on the new BLAST results generated with the NMR sequence, CST has confirmed that they would predict #3686 antibody to also work in NMR tissues (email correspondence attached).

CST also confirmed 100% sequence identity between mouse p27 protein (Uniprot ID: P46414) and the epitope of CST #3686 antibody. Additionally, they referred us to publications on their website showing positive immunohistochemistry with #3686 antibody in mouse tissues^{2,3}. The results of the BLAST search undertaken by Cell Signaling Technologies is summarised in Table 1 here:

Species	CST # 3686 epitope*	CST # 2552 epitope*
Human Uniprot ID: P46527 CDN1B_HUMAN	% Identity =100% Query Coverage = 100%	% Identity =92.86% Query Coverage = 100%
Mouse Uniprot ID: P46414 CDN1B_MOUSE	% Identity =100% Query Coverage = 100%	% Identity =100% Query Coverage = 100%
Rat Uniprot ID: F7EXK3 · F7EXK3_RAT	% Identity =94.74% Query Coverage = 100%	% Identity =100% Query Coverage = 100%
Naked mole rat UniProt ID: A0A0P6JT45 · A0A0P6JT45_HETGA	% Identity =94.74% Query Coverage = 100%	% Identity =92.86% Query Coverage = 100%
Monkey Uniprot ID: G7PJW2 · G7PJW2_MACFA	% Identity =100% Query Coverage, = 100%	% Identity =92.86% Query Coverage = 100%

Table 1: Summary of the BLAST results comparing p27/P27 protein sequence from five species with epitopes of two anti-p27 antibodies from Cell Signaling Technology
*The epitope sequences of the antibodies are proprietary. Therefore, all BLAST results were provided by the technical support team at Cell Signaling Technology. An email correspondence between the first author (S.M.) and CST has been included in the supplementary files.

2. We have now also validated #3686 antibody in the anterior prostate of both naked mole rats and mice which express very high levels of p27 protein in the majority of the cells⁴ and makes it very easy to assess the specificity of this antibody in these species. **These new positive control images have now been included in the updated Supplementary Fig. 4 (previously called Extended Data Fig. 1).** To exclude non-specific binding of the secondary antibody (goat anti-rabbit Alexa555, Invitrogen #A21428), negative controls, where staining was performed with the secondary antibody only, has also been included in Supplementary Fig. 4d.

3. A few cells in the mouse and NMR intestine that are present above the crypt base (*Lgr5*⁻) also express the p27 protein (Fig 2d and Supplementary 4a-b).

4. We **have also used a second p27 antibody** (CST #2552) on tissues of mice, NMRs and humans, and the staining pattern is similar to that achieved with CST #3686 antibody (**Supplementary Fig. 4a-d**).

Supplementary Fig. 4 (updated) | Expression of p27/P27 in mouse, human and NMR intestine. Representative confocal images showing intestinal crypts stained with anti-p27 antibodies (red) and counterstained with DAPI (grey) in **a**, wild-caught mouse (12 month-old) and wild-caught NMR (12 month-old) small intestine, and **b**, mouse (12

month-old), human (28-33-year-old) and NMR (12 month-old) colon. Dashed lines demarcate the outer periphery of crypts. White arrows indicate p27/P27⁺ cells outside the crypt base to the top of the crypts. **c**, Immunofluorescent images of the mouse (12-month-old) and NMR (12-month-old) anterior prostate stained with anti-p27 antibodies (red) and DAPI (grey). **d**, Confocal images of crypts from mouse, human, and NMR distal colon stained with only secondary antibody (Invitrogen, #A21428) and counterstained with DAPI (grey). In **a-d**, the top panel shows staining with CST #3686 and the bottom panel shows staining with CST #2552 antibody. All images in this figure were acquired using a 40X objective lens (1.1 W Corr) on a Zeiss 780 LSM upright confocal microscope. The scale bar is indicated on the images (20 μ m).

In summary, we are confident that the p27 antibody (CST #3683) is indeed staining NMR and mouse tissues (Supplementary Fig. 4a-c) and the lack of staining specifically in the majority of *Lgr5*⁺ cells at the crypt base (*Lgr5*^{CBC}) is a biological result and not a technical artefact (Fig 2c-d). Our interpretation is that *Lgr5*^{CBC} cells in the rodent species, unlike in humans, do not enter G0 (p27⁺), but remain in the active cell cycle (G1-M).

The reviewer states “*In fact, the expression pattern of p27 is also different between Figure 2 and Extended data Fig. 1, and the distinction from nonspecific staining is not clear.*”

The confusion may have arisen from the different magnifications at which the images were acquired for the two figures previously. In Fig. 2c and Fig. 2d images were taken at 63X magnification on a confocal microscope such that the crypt base region where *Lgr5*/*LGR5*-expressing cells reside is the focal point. The images in previous Extended Data Fig. 1 were taken on a widefield microscope using 20X magnification lens so that the whole crypt axis could be visualised and p27 positive cells further up the crypt could be seen. To make the visual comparison between Fig 2c, 2d and updated Supplementary Fig 4a-b (previously called Extended Data Fig. 1) easier, **we restained the tissues and took new images using a confocal microscope at 40X magnification** (Supplementary Fig 4a-b).

3. Most data remain descriptive with classical experimental methods. There should be support for data from transcriptome analysis, such as single-cell RNA-seq of naked mole rat epithelial cells.

Response from authors:

The aim of our study was to characterize the structure and known cell types of the intestine (histologically), assess the functionality of the intestinal mucosa against toxins (DSS treatment) and determine stem cell kinetics (cumulative BrdU labelling) in a novel organism, the naked mole rat. More specifically, we quantified the proportion of main secretory lineages (goblet, enteroendocrine cells), absorptive (enterocytes), and spatial distribution of *Lgr5*⁺ cells at a single cell level. The results obtained highlight some fascinating differences in the naked mole rat intestinal biology and help us to understand how these animals maintain optimal tissue biology that is required for a longer lifespan. Whilst sc-RNA analysis would have been interesting in potentially identifying novel epithelial, immune and stromal subsets, we believe it is beyond the scope of this study to have undertaken such an analysis, but should definitely be considered in follow-up studies.

4. Even though the study was performed *in vivo* in naked mole rats, the study in organoids was performed only in mice. Comparative studies in mouse, naked mole rat, and human intestinal organoids are required.

Response from authors:

Assessing cellular kinetics *in vivo* is the most accurate way to determine the turnover rate of adult stem cells in their niche, but this is a laborious method that involves sacrificing a significant number of animals. We did this for naked mole rat intestinal stem cells (Fig. 3d-e, 4d-e) which first required quantification of BrdU clearance time *in vivo*, establishing the appropriate dose of BrdU to use and determine the frequency of injections needed to label all cells in the S-phase (Supplementary Fig. 5a-d). In total, we used n=31 naked mole rats for this experiment. For the human *LGR5*⁺ cell turnover rate, we have relied on the high-quality data recently provided through an elegant orthotopic xenotransplantation methodology pioneered in Toshiro Sato's laboratory that allowed division rates of human *LGR5*⁺ to be estimated *in vivo*⁵.

The rationale for using mouse organoids (small intestine and colon) to derive *Lgr5*⁺ cell turnover rate was that there was already *in vivo* BrdU data available for mouse small intestine from high profile publications^{1,6}. We sought to minimize the use of mice in our study by first assessing the usefulness of organoids in stem cell turnover studies by expanding small intestinal organoids from mice and subjecting them to cumulative BrdU labelling. As presented in our previous draft (line 260-267, old manuscript), there was a very strong congruence between the published CBC turnover rate *in vivo*^{1,6} and *Lgr5*⁺ cell division time we found *in vitro* in organoids. This showed that mouse organoids recapitulated the *in vivo* stem cell kinetics and, therefore, for the mouse colonic *Lgr5*⁺ cell division rates, we grew mouse colonic organoids and established the turnover rate by administering BrdU *in vitro* (Fig. 4f in old manuscript, line 335 to 337).

However, we appreciate the reviewer's concern that all comparative analyses should be done using one methodology. Rather than rely on organoids (as suggested) for which we have no established protocol for NMRs, we have **now extensively performed BrdU labelling experiments *in vivo* in mice (updated Fig. 3f and 4f, Supplementary Fig. 5e)** to provide a side-by-side comparison to the NMR *in vivo*

experiment (Fig. 3e and 4e) and Ishiwaka et al's human *in vivo* data⁵. We have removed the small intestinal and colonic organoid data that was previously presented in the manuscript.

Briefly, n=30 C57BL/6J mice (2 to 3-month-old) were injected with BrdU every 6 h for 2.25 days (maximum 10 injections per mice in total) and small intestinal and colonic tissues were collected from n=3 animals at 1h, 7h, 13h, 19h, 25h, 31h, 37h, 43h, 49h and 55h after the first injection (updated Fig. 3f and 4f, top panel). We co-stained the tissue sections with anti-BrdU antibody and *Lgr5* mRNA probe and quantified the percentage of *Lgr5*^{+CBC} cells that were positive for BrdU (i.e. labelling index). We observed a linear increase in the labelling index with time which reached saturation at 25 h in the small intestine (**Updated Fig. 3f, bottom**) and 39 h in the colon (**Updated Fig. 4f, bottom**). Changes to the text have been made in the resubmitted manuscript (highlighted in red) describing the new mouse *in vivo* labelling data (lines 211 to 239 and lines 251 to 275 in the resubmitted manuscript).

Fig. 3f (updated) | Top, schema showing time points at which C57BL/6J mice (2-3 month old, n=30) were injected intraperitoneally with BrdU (green) and when intestinal tissue was collected (red). Bottom, scatter plot showing a linear increase in the labelling index ($Lgr5^{+CBC}BrdU^+ / Lgr5^{+CBC}$) with time in the mouse small intestine. Each dot denotes mean \pm SEM of 30 crypts counted per animal (n=3 animals/time point). The dotted line (red) on the graph represents a linear projection defined by the equation displayed on the plot. T_s was derived from the y-intercept ($T_s \times \text{slope}$).

Fig. 4f (updated) | Top, experimental strategy showing the time points at which C57BL/6J mice (2-3 month old, n=30) were injected intraperitoneally with BrdU (green) and intestinal tissues were harvested (red). Bottom, scatter plot showing a linear increase in the labelling index ($Lgr5^{+CBC}BrdU^+ / Lgr5^{+CBC}$) with time in mouse distal colon. Each red dot denotes the average labelling index calculated from 30 crypts per animal and n=3 mice were evaluated for each time point. The equation on the graph defines the linear projection shown as a red dotted line. T_s is derived from the y-intercept ($T_s \times \text{slope}$).

Furthermore, we have also assessed the impact of repeated BrdU injections on the cellular kinetics of mouse intestinal crypts by injecting EdU in mice that had received no BrdU injection (control) and mice that had received 10 rounds of BrdU injections (BrdU treated) (**Supplementary Fig. 5e, top**). After one hour of EdU administration, we detected similar percentage ($26.3\% \pm 1.3\%$) of EdU⁺ cells in the intestinal tissue of both the control and BrdU-treated mice (**Supplementary Fig. 5e, bottom**). This indicates that repeated administration of BrdU in mice did not impact the cellular kinetics in the intestinal crypts. All this new information is highlighted in red in the **Supplementary Note 1 of the updated manuscript** (lines 267 to 274, resubmitted Supplementary Information)

Supplementary Fig. 5e (updated) | Top, experimental strategy showing injection of EdU (purple dots) in mice that had either not received any BrdU injections (control) or had been given 10 consecutive BrdU injections previously (green dots). Tissues were analysed 1 hour after EdU injection (red dots at 56 h). Bottom, representative immunofluorescent images from mouse small intestinal crypts stained with EdU-Alexa488 dye (red), anti-BrdU (green) antibody and DAPI (grey). Bar graphs displaying no significant difference in the percentage of EdU⁺ cells in mouse duodenal crypts with no BrdU injections or after 10 successive BrdU injections (n=40 crypts from 2 animals per group, $P=0.40$). In all cases, statistical significances were determined by performing Student's t-tests using a two-tailed, unpaired, and unequal variance.

Each bar represents mean \pm standard error of mean. Scale bars are indicated on the images (25 μ m).

5. Cell proliferation is also related to the effects of aging. Even at the same age of 12 months, the lifetimes of mice and naked mole rats differ greatly. In Fig. 5, the authors made comparisons between 6-month-old and 3-year-old naked mole rats. However, naked mole rats have a life span of >20 years. Comparisons at this slight difference in age are meaningless even if there is no difference. What about the differences in cell cycle with age in the naked mole rats more than 20 years old?

Response from authors:

We agree with the reviewer that ideally we would want to show no change in the proliferation of *Lgr5*⁺ cells at the crypt base in > 20-year-old mole rats. Unfortunately, getting older naked mole rats is extremely difficult. However, we show that cell proliferative dynamics of *Lgr5/LGR5* expressing cells specifically at the crypt base in mice and humans do not change with increasing age and we believe it to be a common feature of ASCs shared across mammalian species. In humans, we comprehensively analysed samples spanning most of the adult lifespan (28 to 74 years) and found no change in Ki67 status in *LGR5*^{CBC} cells (Fig. 5d, right panel). Similarly, we assessed 2- to 18-month-old mice and also did not find any increase or decrease in the proliferative index of *LGR5*^{CBC} cells with age (Fig. 5d, middle panel). Indeed, for naked mole rats, we were limited in our analysis and could only use tissues from 0.5 to 3 years of age (Fig 5d, left panel) which corresponds to 2 to 12% of their total lifespan. However, extrapolating from the other two species in our study, it is reasonable to assume that there is also no change in the proliferation status of *Lgr5*⁺ cells at the crypt base with increasing age in naked mole rats. Moreover, given the exceptional longevity and disease resistance in NMRs and several studies showing a minimal decline of tissue function with age, out of the 3 species we compared, NMRs would be predicted to exhibit no or minimal change in ASC kinetics and functionality towards the end of life. We also request Reviewer 1 to refer to our response to Reviewer 2 (comment 3, page 28) and Graph 2 on page 30 of this document.

6. In humans and mice, p27 is the main player with p16 taking a minor supporting role, but in naked mole rats, the two genes have become decoupled (Proc Natl Acad Sci U S A. 2009; 106(46): 19352-7). What about p16 participation in naked mole rats?

Response from authors:

The reviewer has mentioned a very interesting study that has shown how cell cycle arrest in naked mole rat skin fibroblasts is controlled by a temporal expression of p16^{Ink4a} and p27^{Kip1}, with early arrest regulated by p16^{Ink4a} while regular contact inhibition controlled by p27^{Kip1}.

To address the reviewer's comment, we attempted to stain naked mole rat small intestine, colon and anterior prostate with anti-p16^{Ink4a} antibody (ab189034). We failed to detect any p16^{Ink4a} positive cells in any of these tissues (data not shown). We think this negative result is likely due to the anti-p16 antibody not cross-reacting with the naked mole rat protein. We will need to test a panel of different anti-p16 antibodies which is beyond the scope of this study. Speculatively, based on the skin fibroblast study that the reviewer has mentioned, we would expect that as epithelial cells of the intestine migrate up towards the differentiated zone of the crypt-villus axis and are in constant contact with adjacent cells, p27^{Kip1} would be the dominant regulator of cell cycle arrest.

7. The association between somatic mutation rate and longevity has already been suggested in references 4 and 5, but where is the advantage or novelty of using naked mole rat to show it in this paper? What will be changed by what this paper reveals? Please emphasize the importance of its significance.

Response from authors:

We agree with the reviewer that the association between somatic mutation rate and longevity has already been shown by ref 5. What we are proposing is one of the mechanisms underlying this observation which is that the accrual of somatic mutations tracks with stem cell divisions. We investigated whether differences in the number of cell divisions we have estimated among species could explain the observed differences in mutation burden reported in ref 5. Indeed, we show that it is specifically cell division rates of the intestinal stem cells at the crypt base ($Lgr5^{+CBC}/LGR5^{+CBC}$) that are associated with somatic mutation rate as well as longevity. We did not find this association in cell division rates of $Lgr5/LGR5$ -expressing cells outside the crypt base, which most likely represent early progenitor cells. Our ASC division rates explain 70% of the differences reported in the somatic mutation rates across species⁷. Under a model where mutation accumulation in the crypt is a result of mutations which occur at cell division in $Lgr5^{+CBC}/LGR5^{+CBC}$ cells, we found that in mice every cell division would lead to about 5 substitution mutations, compared to 2 in NMRs and 1 in humans. Normalising substitution mutation rate per species by cell division rates (165 divisions/year in mice and 50 divisions/year in humans) reduced the 17-fold difference between mouse and human mutation rates to 5, thus roughly 70% of the excess mutations in mice per year can be accounted for by ASC division rates alone. The equivalent comparison between mouse and NMR substitution rates per ASC division rate also showed 70% higher mutation rates in mice resulted from having a faster ASC division rate.

Furthermore, our detailed dissection of specific phases of the cell cycle in different species provides a unique insight into the contribution of different sources of somatic mutagenesis. We show that NMR $Lgr5^{+CBC}$ cells spent a longer time in G1 and/or G2 compared to human counterparts that prevented proliferation by remaining in G0 (Fig. 4j, resubmitted manuscript). As such, NMR ASCs experience higher metabolic rates and subsequently higher mutations in G1/G2 phases than human ASCs, and this can

partly explain the 1.95-fold difference in mutation rates between the two species. In summary, we show that ASC division rates play an important role in the accrual of somatic mutations, with fast-dividing mouse lineages accumulating much higher mutation rates while ASCs in longer-lived species turn over slowly. **This is discussed in depth in the manuscript lines 338 to 384.**

Minor points

1. Most Bar graphs should be overlaid with dot plots.

Response from authors:

Nature Communications formatting guidelines suggest including individual dot plots only in case of $n < 10$. We have counted $n = 30$ to 150 crypts in our analyses and plotting these datapoints as dot plots looks chaotic and messy when overlaid on bar graphs. Whenever we have used $n < 10$ (Fig. 5b, 5e, 5f, 6c, 6f, Supplementary Fig. 9e and 10b), we have shown individual dot plots.

2. The concentrations of DSS that cause enteritis vary widely depending on the batch and the environment in which the animals are housed. Weight trend graphs should also be shown to firmly indicate that DSS enteritis has occurred.

Response from authors:

We have performed all the experiments in mice and naked mole rats using the same batch of DSS (Cat: 42867-100g, lot: BCCB5021). It is not possible to house naked mole rats and mice in the same housing facility. The C57BL/6J mice were raised in a Specific Pathogen Free (SPF) facility whereas the naked mole rats were housed in a non-SPF animal facility. Indeed, it has been shown in mice that housing conditions influence the gut microbiota of mice which consequently impacts the development of colitis. For example, in some studies, colitis in germ-free mice has been shown to be largely absent whereas enteric bacteria present in conventional mice seem to be important for DSS-induced colitis⁸. Similarly, after transfer from SPF to non-SPF housing conditions, Rag1^{-/-} mice developed more severe colitis in non-SPF unit⁹. Extrapolating from these studies, we would have expected to have seen more damage/severity of colitis in naked mole rats that were not housed in sterile conditions and thus were exposed to potentially more pathogenic microorganisms. However, in our study, 2.5% DSS treatment induced intestinal damage in SPF-raised C57BL/6J mice (Fig. 6c, Supplementary Fig. 9e), but failed to cause any inflammation or epithelial damage in naked mole rats even at higher concentrations (5% and 8.75% DSS) (Fig. 6f and Supplementary Fig. 10b). Therefore, the difference in housing units cannot explain the resistance of naked mole rats to DSS induced colitis and, as discussed in the manuscript previously, is most likely due to the higher proportion of

goblet cells (Supplementary Fig. 8c) that express higher levels of mucins (Supplementary Fig. 8e) and secrete a thicker protective mucin layer in the intestinal lumen that is harder to penetrate (Supplementary Fig. 8d).

Two independent pathologists confirmed the extent of intestinal damage in our experimental mice using in-depth histopathological scoring of epithelial damage and extent of inflammation (Fig. 6c, Supplementary Fig. 9e). Furthermore, the status of animals was also monitored daily by general examination of posture and mobility. Animals were also weighed daily. We had not included this data in the previous manuscript, but **have now added the weight trend graphs in new Supplementary Fig. 11 and discussed the results in new Supplementary Note 4** in the resubmitted manuscript.

Briefly, in agreement with previous findings^{10,11}, we observed an 8.4% loss of initial body weight in mice receiving 2.5% DSS in drinking water by day 7, while the weight of control mice receiving unsupplemented water remained unchanged (Supplementary Fig. 11a). In our other experiments where we administered DSS by oral gavaging for 3 days only, weight loss was not a measure to assess the effect of DSS, but was more reflective of the stress induced on the animals by oral gavaging which further supports our reasoning in conducting the experiment for only 3 days and not extending it further. For mice treated with 0% or 2.5% DSS by 4 oral gavages per day (12 rounds of gavage in 3 days), we observed nearly 5% loss of initial body weight in both control and DSS-treated cohort by day 3 (Supplementary Fig. 11b). Similarly, we also observed nearly 5-9% loss of initial body weight in naked mole rats that received 0% or 2.5%, 5%, 8.5% DSS via oral gavage, again showing that body weight loss is due to the invasive method of delivering DSS (Supplementary Fig. 11c).

Supplementary Fig. 11 (new) | Change in bodyweight during dextran sodium sulphate (DSS) treatment in mice and naked mole rats. a, Change in the percentage of initial body weight with time in C57BL/6J mice receiving 0% (n=2) or 2.5% (n=4) DSS in water *ad libitum* for 7 days. **b,** Line graph showing gradual decline in the percentage of initial body weight over time in C57BL/6J mice treated with 0% (n=3) or 2.5% DSS (n=3) by oral gavaging at a frequency of four times per day for a total of 3 days. **c,** Line graph displaying the change in the percentage of initial body weight with time in NMRs treated with 0% (n=4), 2.5% (n=6), 5.0% (n=2), and 8.75% (n=2) DSS by oral gavage (four times per day) for 3 days. Each dot represents the mean \pm sem.

3. In Fig. 4j, human and naked mole rat as well as mouse data should be included in the comparison.

Response from authors:

We have now included the mouse data in the comparison in **updated Fig 4j**:

Fig. 4j (updated) | Stacked bar graph showing the time (days) spent in each cell cycle phase by mouse, human, and NMR colonic $Lgr5^{+CBC}/LGR5^{+CBC}$ cells.

4. The cell cycle varies with time of day. Consistency and description of sacrificed time is desirable for evaluation of proliferation.

Response from authors:

We have **now added these details to the methods section “In vivo administration of BrdU and EdU”** (lines 632 to 640, resubmitted manuscript) as shown here:

“For cumulative labelling protocol using BrdU, the first injection in naked mole rats was administered between 14:00 to 15:00. Subsequent BrdU injections were given every 8 hours for a duration of 5 days and intestinal tissues were collected every 8 hours after the first injection. In C57BL/6J mice, the first BrdU injection was also given

between 14:00 to 15:00, with further injections given every 6h for a total of 2.25 days. Mouse intestinal tissues were collected 1 hour after each injection. The rationale for the frequency and total number of injections in the two species is discussed in Supplementary Note 1. Tissue processing and staining with BrdU and EdU are described in subsequent method sections.”

5. Longevity of the gut is not directly related to survival. Is the proliferation of vital organs other than the intestine similarly slow?

Response from authors:

Yes, the slower proliferation of naked mole rat cells seems to be a shared feature across tissue types. For example, Emmrich et al¹² have recently shown that the cell cycle of hematopoietic cells is prolonged compared to mouse counterparts. Similarly, neural stem and progenitor cells from naked mole rats showed markedly slower proliferation rates than mouse cells and were found to be in prolonged G0/G1 phase¹³. Naked mole rats fibroblasts also proliferate very slowly in culture¹⁴. **These studies have now been mentioned in the discussion.**

Reviewer #2 (Remarks to the Author):

This well written paper with beautiful histological images examines intestinal stem cells of the extraordinarily long-lived and cancer resistant naked mole-rat (NMRs). The authors report a considerably larger adult stem cells (ASC) pool and that the ASC of NMRs do not enter quiescence but rather show slow proliferation through the active cell cycle, facilitating better translational fidelity, DNA repair, reduced mutation rates and less propensity for ASC exhaustion. In addition, the expanded pool of differentiated cells in the intestine is posited to afford greater protection of the stem cell zone and better maintenance of tissue homeostasis even when challenged with a well- established intestinal stressor. The methodology is sound and the experiments well planned. The results are clearly articulated. The discussion of the findings is very brief and could be expanded to more accurately place these exciting data in light of what is already known about some of the unusual features of the naked mole-rat.

1) This original study supports prior findings of sustained tissue homeostasis and stress resilience of NMRs and provides a mechanistic explanation for their extreme longevity and cancer resilience. Several seminal NMR papers should be cited e.g., Yamamura et al., 2022 (<https://doi.org/10.1111/cas.15570>) Miyawaki et al., [Nature.com/articles/ncomms11471](https://www.nature.com/articles/ncomms11471) Miura 2021 doi: 10.1007/978-3-030-65943-1_13, and many others

Response from authors:

We thank the reviewer for highlighting these references and have now included these and others on stress resistance in the discussion of the manuscript. We would also request the reviewer to refer to the response to point 5 later on in this section.

2) The longer proliferative cycle of NMR ASCs when compared to that of mice is similar to that observed for species differences in skin fibroblasts in culture. We question if this simply reflects a difference in body temperature/ metabolic rate or if there is another explanation? No mention is made that NMRs maintain a markedly lower body temperature than seen in mice and if this could affect the ASC kinetic profile.

Response from authors:

The reviewer is referring to Seluanov et al¹⁴ study that showed cells from small shorter-lived species displayed continuous rapid proliferation whereas cells from small long-lived species, including NMR lung and skin fibroblasts grown in culture, showed continuous slow proliferation. All fibroblast lines from 14 rodent species with diverse body sizes and lifespans except NMRs were maintained at 37°C while NMR fibroblasts were grown at 32°C. Figure 2b of this study¹⁴ itself addresses the reviewer’s query of whether lower body temperature/metabolic rate explains the slower cellular kinetics. NMR fibroblasts are not an outlier in this study, but the growth rate of fibroblasts from these animals clustered with the slow-dividing group that included cells from chipmunks, muskrats, chinchillas and squirrels. Long lifespan, rather than body temperature or metabolic rates, is the common trait shared by these species.

Secondly, there is also evidence in our study being reviewed here that differences in body temperature/metabolic rates cannot explain the differences observed in ASC kinetics between different species. As our study was conducted *in vivo*, the NMR body temperature was monitored at 32°C while in mice it was 37°C. Unlike previous studies where slower kinetics were characterised using growth curves and relative uptake of Brdu/EdU, we used cumulative BrdU labelling to determine the exact duration of the cell cycle in NMR and mouse intestinal epithelial cells. Moreover, we were able to assess the cell cycle turnover of different cell types in the intestine simultaneously. The cell cycle durations of ASCs (*Lgr5*⁺ cells at the crypt base) and early progenitors (*Lgr5*⁺ cells above the crypt base) in the mouse and NMR intestine are summarised in the table below (also refer to Fig 3h, 4h, resubmitted manuscript):

Cell division time (days)	Region	Mouse	NMR	Human ⁵
Lgr5 ^{+CBC}	Small intestine	1.54 ± 0.08	4.41 ± 0.07	
	Colon	2.21 ± 0.32	7.29 ± 0.62	7.3 ⁵
Lgr5 ^{+above crypt base}	Small intestine	1.30 ± 0.06	2.34 ± 0.17	
	Colon	1.38 ± 0.03	2.74 ± 0.18	1.5 ⁵

Of specific relevance to the point raised by the reviewer, we show that the different cell types within the NMR intestine exhibit different cell division rates, with ASCs

(*Lgr5^{CBC}*) showing longer turnover while early progenitors (*Lgr5^{above crypt base}*) exhibit much faster division rates. If cellular kinetics were affected by differences in body temperature and metabolic rates, we would expect to see *all* cells in the NMRs dividing slower than in mice. However, the estimates of *Lgr5^{above crypt base}* in the NMR small intestine are very similar to mouse *Lgr5^{CBC}* cells. We would like to emphasize that the most important conclusion from our comparative estimates of cellular kinetics is that the NMR ASCs divide slower than short-lived mouse counterparts, but have very similar rates to ACSc found in longer-lived humans⁵. Based on the reviewer's comment, we have updated the discussion as shown here (lines 499 to 506, resubmitted manuscript)

“The slower cellular kinetics of ASCs may be a general characteristic across tissue types in NMRs as our findings in the small intestine and colon mirror the slower proliferation of NMR neural¹³ and haematopoietic¹² stem cells. Even NMR iPSCs¹⁵ and fibroblasts^{15,16} in culture divide slower than murine counterparts and while cell-autonomous growth suppressive mechanisms have been identified and linked to cancer resistance in these cell types^{15,17-19}, future studies need to be undertaken to unravel specific cell cycle modulators in NMR ASCs.”

3) The authors examine ASC division rates in 3 age groups 6m, 1y and 3 years (out of a >37 year lifespan) whereas those of both mice and humans spanned a far larger proportion of their lifespan (2m to 18m out of an average 3 year lifespan and 28-74 years in humans). How were these age groups chosen? Given that no species showed an age-related difference in cell proliferation rates, was this finding expected? please explain the lack of age effects in humans and mice. Can one really determine the relationship between ASC division and lifespan for the NMR given that all age cohorts examined fall within their 1st 10% of their lifespan.

Response from authors

The reviewer highlights an important point regarding a seemingly unexpected finding where we do not see any age-related differences in the cell proliferation rate. More specifically, we have shown no change in the proliferation status (Ki67⁺) of *Lgr5⁺/LGR5⁺* cells at the crypt base cells with increasing age. Our unexpected finding is supported by a separate study published in Nature²⁰ which showed that the fraction

of ASCs (*Lgr5*⁺/*LGR5*⁺) that were EdU⁺ or Ki67⁺ was unchanged in old C57BL/6J mice (24 months) and old human (>75 years) samples compared to young (3 to 9 months in mice and < 25-year-old in humans) (please refer to Extended data Fig.1f-h in Ref 20).²⁰

The reviewer asks “how were these age groups chosen?”. At the beginning of the study, we compared the intestinal tissue of NMRs with wild-caught (not laboratory/inbred) mice that live on average 18-20 months due to external threats. These mice were provided by Dr Dustin Penn’s laboratory at the University of Veterinary Medicine, Vienna where wild-caught mice and successive generations were aged in non-sterile housing conditions, like the NMRs used throughout our study. We first compared the total proportion of one differentiated cell type, enteroendocrine (Chromogranin A⁺) and stem (*Lgr5*⁺) cells in the intestinal tissue of 2M, 12M and 18M wild-caught mice (refer to Graph 2 on page 30 of this document). We found that no age-associated change in the fraction of enteroendocrine cells in the murine intestine (Graph 2b, left). When quantifying the *Lgr5*⁺ population, there was also no difference in the proportion of these cells per crypt in 2M, 12M or 18M (Graph 2b, right), suggesting that stem cell equilibrium has been reached at 2M of age in adult mice and did not change with increasing age. Furthermore, our analysis of the proliferation status (Ki67⁺) of *Lgr5*⁺*CBC* cells in 2M, 12M and 18M wild-caught mice also showed no difference with increasing age (Fig 5d, middle panel, resubmitted manuscript). Using the C57BL/6J laboratory mouse strain, Nalapareddy et al²¹ also showed no age-associated change in the number of *Lgr5*⁺ cells per crypt between 2-3M-old and 20-22-month-old animals (refer to Fig 2a Ref 21). Therefore, *Lgr5*⁺ cell proportions and kinetics in both wild-caught and laboratory-strain^{20,21} mice do not increase or decrease with age in adult mice.

For humans, we most comprehensively analysed samples spanning most of the adult lifespan (28 to 74 years) and also found no change in the Ki67 status of *LGR5*⁺*CBC* cells with age (Fig. 5d, right panel, resubmitted manuscript). Therefore, both in mice and humans, our study and other²⁰ show that the proliferative dynamics of intestinal ASCs remain unaffected by ageing and we believe it to be a common feature shared across mammalian species.

For NMRs, we were restricted to intestinal tissues collected from 6M, 12M and 36M old animals as it is very difficult to get access to older NMRs. This cohort indeed only corresponds to 2 to 12% of NMR's total lifespan. When we compared the differentiated and stem cell populations in the intestine within these 3 age groups, we found no differences in the enteroendocrine or *Lgr5*⁺ cells (Graph 2c below). Analysis of the proliferative index of ASCs (*Lgr5*⁺) in both the small intestine and colon in 6M, 12M, 36M old NMRs also showed no change (Fig. 5d, left panel, resubmitted manuscript). While we agree that the inclusion of an older NMR group would be ideal, we think it is reasonable to extrapolate from mouse and human data that there would also be no change in the proliferation status of *Lgr5*⁺ cells at the crypt base with increasing age in these animals. Moreover, given the exceptional longevity and disease resistance in NMRs and several studies showing minimal decline of tissue function with age, out of the 3 species we compared, NMRs would be the one where we would expect no or minimal change in ASC kinetics and functionality towards the end of life.

Graph-2 | Comparison of small intestinal cell populations between different age cohorts of wild-caught mice and wild-caught naked mole rats (NMRs). **a**, Bar graphs comparing the percentages of ChrA⁺ cells/villi and *Lgr5*⁺ cells/crypt in the duodenum (SB1) of 2, 12 and 18-month-old mice (n=30 crypts or villi were counted from 3 animals per age group). **b**, Bar graphs showing the percentages of ChrA⁺ cells in the villi and *Lgr5*⁺ cells in the crypts of small intestines from 6, 12 and 36-month-old

naked mole rats (n=30 crypts or villi were counted from 3 animals for each arm of the comparison). In all cases, statistical significances were determined by performing a two-tailed, unpaired Student's *t*-tests using an unequal variance. Significant *P*-values are denoted as such: ***<0.001. Each bar represents the mean ± standard error of the mean.

4) Where did you get the NMR lifespan record of 25 years? This is a gross underestimate. Lee et al., 2020 doi: 10.1007/s11357-019-00150-7. published a lifespan of 37 years others as did Yamamura; others have published >31 years (e.g., AnAge website) while Ruby et al., 2018 kaplan Meier demographic analysis (<https://doi.org/10.7554/eLife.31157>) showed that the median lifespan for subordinate mole-rats was 19 years while that of dominant breeders was significantly longer.

Response from authors

The reviewer is referring to Maximum Lifespan Potential (MLSP) for NMR²² which is often longer for certain individuals in a species. We have taken the mean adult lifespan data from Cagan et al⁷ and Ruby et al²³. Cagan et al estimated mean adult lifespan using Kaplan-Meier survival analysis from data available in Species 360 database and human census⁷. They define lifespan “as the age at which 80% of individuals reaching adulthood have died, to reduce the effects of outliers and variable cohort sizes that affect maximum lifespan estimates²⁴ (Methods Ref 7).

Ruby et al did a survival analysis using over 3,000 data points including sub-groups of male/female and breeder/non-breeder NMRs²³. They have shown that in non-breeding females, nearly 80% of the population perishes by day 9,000 (i.e., 24.65 y) similar to the report by Cagan et al⁷. As we only used non-breeding animals in our study, we decided to use the lifespan given by Cagan et al⁷ and Ruby et al²³.

5) The response to DSS showed that mice showed typical injury whereas the NMRs were resilient against such stress but ASCs stopped proliferating. A similar response has been previously reported in skin fibroblasts in response to several stressors (Lewis KN et al., 2012 doi: 10.1159/000335966). This profound resistance to stress should be more comprehensively discussed.

Response from authors

We thank the reviewer for highlighting this and have added this reference and others on the association between stress resistance and longevity in the discussion (lines 536 to 537, resubmitted manuscript). However, we have refrained from overinterpreting our results and aligning our results with what has been shown in previous studies that have used NMR fibroblasts *in vitro*. The main difference with our study is that we have characterised the behaviour of epithelial cells (and not fibroblasts) to a specific chemical, DSS, *in vivo*. Moreover, we would like to emphasize that while we did not see overt damage in the intestinal tissue of DSS-treated NMRs and detected no change in the differentiated cells residing in upper regions of the crypts, we do observe a much more robust response in the stem/progenitor cells at the bottom of the crypt in these animals. This included apoptosis and shutting down of proliferation in NMR crypt cells at a much higher level than in mice. Therefore, different cell types within the same tissue behave differently to the same stress in NMRs, and while NMR ASCs are hypersensitive to DSS, differentiated cells of the intestinal epithelium are not. The most likely reason for the latter is the thicker mucus layer in the NMR intestinal lumen that is produced by a higher proportion of goblet cells (refer to Supplementary Fig 5c, 5d, and Discussion of resubmitted manuscript).

6)The discussion appears rather superficial and makes mechanistic suggestions that are not fully substantiated. The authors suggest that the harsh underground habitat of NMRs has acted as a strong selective force for a larger reserve of ASCs; I wonder if the toxic nature of the plant secondary defense mechanisms in their diet (large pyrethrium tubers) may play a role in both their larger ASC and their resilience in the face of stress. Does their microbiome influence this in anyway?

Response from authors

We thank the reviewer for this comment. A limited number of studies have characterized the components of the NMR diet and microbiome^{25,26} and the impact on ASC function is unknown and it would be largely speculative to link this to our findings. There is also a paucity of data on the role of the microbiome/diet on the *Lgr5⁺/LGR5⁺* in mice and humans. We have now rephrased the sentence and rather than discussing what potential selective pressures may have given rise to a higher proportion of ASC populations in NMRs, we have focused on what a large ASC reserve would mean for

the maintenance of tissue homeostasis required for a longer lifespan as shown here (lines 556 to 560, resubmitted manuscript)

“The evolution of a larger reserve of ASCs across tissue types in NMRs would facilitate efficient tissue maintenance in an environment of high oxidative²⁷ and mechanical stress²⁸, and lower the probability of deleterious mutations becoming fixed due to increased selection against deleterious variants²⁹⁻³¹ while slowing down clonal expansion seen in ageing³².”

7) I also question the link to the disposable soma theory, since the authors make no mention of their eusocial lifestyle whereby reproduction is restricted to only a few animals with high reproductive demands, given that all animals used in this study are very young subordinates that may have not even attained adult mass, I wonder if the breeding females that have high physiological demands associated with doubling of body mass during pregnancy show similar ASC kinetics and if they would support the disposable soma theory.

Response from authors

The reviewer raises an interesting point regarding potential differences in ASC kinetics during specific physiological conditions like pregnancy. The larger implication here is that intestinal ASC may be regulated by oestrogen/ER α signalling and that animals with differing reproductive potential may have different patterns of ASC kinetics. While the long-range effect of oestrogen has been demonstrated in haematopoietic stem cells (HSCs) whereby HSC cells divide slightly more frequently in females than in male mice³³, there is no study to our knowledge showing systemic oestrogen regulates mouse or human *Lgr5*⁺ cell proliferation in the intestinal crypt. Wnt, Notch, Hedgehog, BMP and Hippo/YAP pathways are the established modulators of ASCs in the intestine^{34,35}.

It is noteworthy that all NMRs used in our study had a breeding, pregnant female co-habiting with the subordinates. It has been shown that the levels of estradiol are elevated in the female subordinates through coprophagy during the queen's gestation period³⁶. Therefore, any potential effect of oestrogen on the kinetics of *Lgr5*^{+CBC} cells

would also be seen in the young female subordinates we have used in our study. Additionally, if intestinal ASCs were indeed regulated by oestrogen/ER α signalling, we would expect to see differences in the turnover rates of *Lgr5*^{CBC} cells in subordinate females compared to males. We show here (Graph 5 below) that we do not observe any difference in the proliferative index (Ki67⁺) of NMR *Lgr5*^{CBC} cells between male and female animals (Graph 5a). This is similar to that seen in mouse and human ASCs separated by sex (Graph 5b, 5c). Moreover, when we assessed BrdU labelling index of *Lgr5*^{CBC} cells in NMR male or female subordinates, we again observed no difference (Graph 5d). This result aligned with that seen in mice (Graph 5e). We therefore conclude that it is unlikely that intestinal ASCs would exhibit differences in the turnover rates based on reproductive status.

Graph 05 | Proliferation of intestinal *Lgr5*^{+CBC} cells is not influenced by sex. a-c, Bar graphs showing no significant difference in intestinal *Lgr5*/*LGR5*^{+CBC} cell proliferation index (Ki67+) between male and female individuals in a, naked mole rats (NMRs), b, mice and c, humans. P -values calculated from the two-tailed Wilcoxon ranksum test are shown on the

graph. **d-e**, Scatter plots comparing the cumulative increase in BrdU labelling ($Lgr5^{+CBC}BrdU/Lgr5^{+CBC}$) with time between male (blue) and female (green) individuals in **d**, NMRs and **e**, mice. The least-square fit of the data points before saturation (labelling index < 1.00) is shown as dashed lines defined by the equation displayed on the graph. 1/slope of the regression line denotes the rate of cell division. *P*-values from two-tailed *F*-tests comparing the slope of the regression lines (i.e., rate of cell division) between male and female individuals are labelled on each graph

Reviewer #3

Montazid et al. present an analysis of intestinal stem cell (ISC) proliferation and cell cycle activity in 3 species (mouse, naked mole rat = NMR, and human). The authors employ Lgr5 as a marker and study associations between cell cycle activity, aging, and lifespan. The authors provide compelling evidence on differences in cell cycle and quiescence control of ISCs. While short lived mice have highest proliferation rates, ISCs of long-lived humans show reduced rates of cell cycle activity associating with longer lifespan in these species compared to mice. The authors also find a similar kind of correlation for the association of cell cycle activity and mutation rates in ISCs (as extracted from other publications). Interestingly, the authors also find some evidence that different species employ different mechanism of ISC cell cycle control to ensure longevity. While NMRs exhibited prolonged G1 and G2 stages to reduce cycling activity, human ISCs exhibit an increase in quiescence (G0). Moreover, the authors identify a difference in the composition of differentiated cells in the intestinal epithelium and mucosal layer thickness as well as ISC sensitivity to chemical induced toxicity in NMRs compared to mice, indicating that in addition to cell cycle control, shifts in the production of differentiated cells can also contribute to species-specific mechanisms of longevity protection of the intestine.

Overall criticism: this is an important study that provides a detailed and compelling analysis of ISC cell cycle control. The study has important implications for our understanding of somatic stem cells in organism aging. I have a few specific comments that should be addressed:

1. For some of the figures, the authors use antibody staining to determine cell cycle stages or proliferation activity. The authors should discuss the possibility that species specific differences in the sensitivity of the antibodies lead to differences in the staining of cells in different species, which could have an impact on the quantification of cells being positive/negative for certain cell cycle or proliferation markers if in one species the detection limit is lower or higher compared to the other species.

Response from authors:

The reviewer highlights an important point that species-specific differences in the sensitivity of antibody binding could potentially impact downstream quantitative analyses. In the study presented here, we have used five antibodies (anti-EpCAM,

Ki67, p27, BrdU, and phospho-histone-H3) to stain and determine cell cycle stages or proliferative activity of epithelial cells of the intestinal tissues from mice, humans and naked mole rats (Fig. 1-4, Supplementary Fig. 6). We selected antibodies for downstream quantitative analysis after a careful initial screening with a number of antibodies from different manufacturers for each target protein. Besides the intestine, we also included other tissues and cell lines known to express each protein/target (e.g mouse/NMR anterior prostate, human cancer cell line) to assess the specificity and sensitivity of antibodies in all three species. Only those antibodies that gave a highly specific signal were used for quantification purposes in the intestine. The table on the next page provides a summary of the antibodies used in our study and should help in assessing the reliability of the sensitivity/specificity of the antibodies used.

The reviewer's concern may have stemmed from the differences in the staining intensity that are seen in some immunofluorescent images between the three species. We observed significant variation in the autofluorescence levels across these species, with mouse tissue emitting the most and naked mole rats the least. This variation necessitated adjusting the laser powers of the confocal microscope during image acquisition so that maximal image contrast was achieved while reducing the autofluorescence signals. The maximum and minimum displayed pixel values of individual channels were adjusted across the entire image set to correct for autofluorescence. These adjustments resulted in varying intensities for specific signals in the three species. We took a binary approach for the quantification of the antibody-based signals. The presence of any specific signal in the target compartment inside a cell was considered positive regardless of the staining intensity, mitigating the need to measure the signal intensity and introducing an arbitrary threshold for scoring a cell positive or negative which would inevitably ignore species-specific differences in the levels of protein expression. We have now added these details in the "Fluorescent Microscopy" (lines 835-857, resubmitted manuscript) and "Histological quantification (Fluorescent)" (lines 874-885, resubmitted manuscript) sections in Methods.

2. In Figure 2 the authors use 2-3 animals per species. I would recommend to leave out p-value calculations for this experiment (and any other with <3 n-number). It is understandable that studies on NMRs cannot be conducted with too many replicates. The data show clear differences and strong effect sizes, but I would recommend to leave out the p-value calculation in these cases.

Response from authors:

In Fig 2, data is obtained from n=3 animals per species. For humans, we have used samples from n=4 donors. The numbers are given in the Figure 2 legend. The Table below provides a summary of the datapoints used to generate Fig. 2a, 2b, 2d.

Figure 2		Mouse and NMR		Human	
Panel no	Description	n(animal)/ species	n(crypt)/ species*	n(individual)	n(crypt)*
Fig 2a	% Lgr5 ^{CBC} Ki67 ⁺ (small intestine)	3	126	---	---
Fig 2b	% Lgr5 ^{CBC} Ki67 ⁺ (distal colon)	3	80	4	65
Fig 2d	% Lgr5 / LGR5 ^{CBC} p27 ⁺ (distal colon)	3	50	4	30

*mean ± SEM was generated by averaging over the values from individual crypts

In most of the figures, we have used n=3 animals per rodent group. The experiments where we used n=2 NMRs were in the cumulative BrdU labelling experiment at each

time point but there were no p-values calculated for this and only SEM are given. Similarly in Supplementary Fig 5c (previously referred to as Extended Data Fig. 2c), we used n=2 NMRs per time point to calculate the concentration of BrdU in the blood. We do not show any p-values here either. In Supplementary Fig 5a-d, 5d-e (previously referred to as Extended Data Figure 2), we have also used n=2 NMRs, but have counted 40-60 crypts per arm and the two-tailed, unpaired Student's t-tests were run using total crypts counted.

3. In figure 4J and in the description, I did not completely understand how the authors discriminate G0 from G1 cells. In the J-panel both cell populations have the same markers. Can the authors explain this a bit better?

Response from authors:

We thank the reviewer for highlighting this and realise that the schema shown in Figure 4j obscures how human LGR5⁺ cells arrested in G0 are discriminated from those in G1. We have not used the KI67⁺P27⁺ status of cells to determine the duration of G0 and G1. As such we have now removed the schema (4j) and instead added a more detailed description in the results section (lines 296 to 308 in the resubmitted manuscript) as shown here:

“For estimating the time spent by LGR5⁺CBC cells in the G1 and G0 phases of the cell cycle, we first used P27⁺ status of these cells to find the proportion of LGR5⁺CBC cells in these two phases. We found that 84.8% ± 1.91 of LGR5⁺CBC cells were P27⁺ cells. Using Equation 5 ($T_{(G1, G0)}^{P27+} = T_T^{(Ref\ 4)} \times LGR5^{+}CBC^{P27+} / LGR5^{+}CBC$, Methods), we estimated G1 and G0 of LGR5⁺CBC cells to be 6.19 days (Supplementary Fig. 7a-b). To discriminate LGR5⁺CBC cells in G0 from those in G1, we used the data published by Ishikawa et al where double staining of Pyronin Y and Hoechst 33342 was used to identify LGR5⁺P27⁺ cells with DNA^{LOW}/RNA^{LOW} content in G0 and DNA^{LOW}/RNA^{HIGH} in G1⁴. Approximately 83% of LGR5⁺P27⁺ cells were found to be in the G0 phase⁴, referred to as the quiescence fraction (QF). Using Equation 6 ($T_{G0}^{P27+} = QF^{(Ref\ 4)} \times T_{(G1, G0)}^{P27+}$, Methods), we estimated the duration of G0 to be roughly 5.13 days (Supplementary Fig. 7b). Finally, the duration of T_{G1} (Equation 7 Methods) was found to be 1.35 days (Supplementary Fig. 7b).”

4. At the end of Figure 6 description the authors conclude: “In summary, our results show that even without tissue damage, NMR intestine responds to low levels of chemical insults more efficiently than mouse, shutting down cellular proliferation in the crypts and triggering apoptosis in the majority of *Lgr5*⁺ cells across the entire intestine.” I don’t know whether this difference is indeed pointing to a better “efficiency”. What would be the advantage? It may even have some unwanted side effects (see comment below).

Response from authors:

We thank the reviewer for highlighting this and have now removed the words “more efficiently” and changed the sentence (lines 483-486 in the resubmitted manuscript) as shown here:

“In summary, our results show that even without tissue damage, NMR intestine responds to low levels of chemical insults by shutting down cellular proliferation in the crypts and triggering apoptosis in the majority of *Lgr5*⁺ cells across the entire intestine.”

5. Figure 6e: the photographs on stool consistency are not clear. What should be seen here?

Response from authors:

We have now labelled the images in Fig. 6e to highlight the specific changes in stool consistency:

Fig. 6e | Gradual change in the stool consistency of NMRs subjected to DSS-treatment (2.5%, 5% and 8.75%) (2 to 6 animals per treatment group). Photographs show three different stool consistencies observed (hard, black: Score 0; soft and yellowish: Score 1; very soft and slightly watery: Score 2).

6. Figure 6i: It looks like there are *Lgr5*-positive cells left in position 4/5 of intestinal crypt of DSS-treated NMRs. There is evidence in mice, showing that DNA damage induced depletion is stronger at crypt base and that position-4 *LGR5*-positive cells can survive and appear to repopulate the crypt base. The study showed that different levels in Wnt signalling (reflected by *Lgr5* levels) positively correlate with DNA damage sensitivity (Tao S et al. 2015). This could be relevant also for the response to DSS and could be discussed.

Response from authors:

We thank the reviewer for this comment which led us to reanalysis the data in Fig 6i. The reviewer suggests that in DSS-treated NMRs there appear to be *Lgr5*-positive cells retained at position 4/5 in the small intestine (Fig. 6i). However, Fig. 6i shows the average mRNA expression of all epithelial cells at each cell position of the NMR crypt. In order to more thoroughly assess if cell position within the crypts impacts the sensitivity of *Lgr5*⁺ cells to DSS treatment in NMRs, we reanalysed the data and quantified the frequency distribution of *Lgr5*⁺ cells along the vertical axis of the crypt (Graph 6 below). In comparison with the controls, in DSS-treated NMRs, we observed approximately 80% decrease in the proportion of *Lgr5*⁺ cell numbers at the crypt base. A similar fraction of *Lgr5*⁺ cells were also lost at cell positions above the crypt base following 2.5% DSS treatment (Graph-6). Whilst differential response between these two positions has been elegantly shown in irradiated mice³⁷, we failed to observe this pattern in NMRs. A likely explanation may lie in the different damaging agents used between the two studies and will be very interesting to explore in future studies.

Graph 6: Frequency distribution graph of *Lgr5*⁺ cells at specific positions relative to the base in the NMR small intestinal crypts in control and 2.5% DSS-treated NMRs. n = 52 crypts were counted from 3 animals/group.

7. Discussion, 1st paragraph: the authors should consider to re-order the discussion: it would be better to first discuss the data and then move into the more speculative role of increases in stem cell number per crypts as an anticancer mechanisms, possibly selected in this harsh environment during evolution.

Response from authors:

We very much thank the reviewer for this excellent suggestion and have restructured the discussion accordingly.

8. Discussion, the authors describe:” we observe between ASC division rates and lifespan is not seen when estimates of *Lgr5*⁺/*LGR5*⁺ cells outside the crypt base are used...” I think that it is possible that these cells outside the crypt might be progenitor or ISCs that are on the way to turn into progenitors. That would explain why there is no correlation of division rates of these cells to longevity because they are short-lived.

Response from authors:

We had been cautious in categorising these cells as such previously without further characterisation, but have now taken the reviewer's suggestion and specified that *Lgr5*⁺ cells above the crypt base are "most likely represent early progenitors" in the discussion (lines 511, resubmitted manuscript)

9. Discussion on mutation rate differences, the authors speculate: "The longer G1 and/or G2 arrest of NMR ASCs compared to truncated G1 and G2, but extended G0, of human ASCs, suggests that higher damage due to increased metabolic rates at gap phases of the active cell cycle would increase the non-replicative errors in NMRs, which may partially explain the 2-fold difference in the substitution rates in the NMR colonic crypt cells compared to human counterparts". While this may play role, an alternative explanation may be that the increased sensitivity of NMR ISCs to damage (DSS model) and the subsequent regeneration of ISCs from position4/5 ISCs and/or progenitors may come at costs of increasing mutation burden in NMRs during lifetime.

Response from authors:

The reviewer provides a plausible alternative explanation for the 2-fold difference in mutation rates between NMRs and human crypts. However, the increased sensitivity of NMR *Lgr5*⁺ cells at the base and above was only compared to mice and we do not know if human *LGR5*⁺ cells are equally as sensitive as those in NMRs. At this stage, it would be too speculative to comment on the differences in mutation burden being due to possible differences in sensitivity in ISCs between NMR and humans.

- 1 Barker, N. *et al.* Identification of stem cells in small intestine and colon by marker gene *Lgr5*. *Nature* **449**, 1003-1007 (2007). <https://doi.org:10.1038/nature06196>
- 2 Gershon, T. R. *et al.* Hexokinase-2-mediated aerobic glycolysis is integral to cerebellar neurogenesis and pathogenesis of medulloblastoma. *Cancer Metab* **1**, 2 (2013). <https://doi.org:10.1186/2049-3002-1-2>
- 3 Hussain, R. & Macklin, W. B. Integrin-Linked Kinase (ILK) Deletion Disrupts Oligodendrocyte Development by Altering Cell Cycle. *J Neurosci* **37**, 397-412 (2017). <https://doi.org:10.1523/JNEUROSCI.2113-16.2016>
- 4 Cosi, I. *et al.* ETV4 promotes late development of prostatic intraepithelial neoplasia and cell proliferation through direct and p53-mediated downregulation of p21. *J Hematol Oncol* **13**, 112 (2020). <https://doi.org:10.1186/s13045-020-00943-w>

- 5 Ishikawa, K. *et al.* Identification of Quiescent LGR5(+) Stem Cells in the Human Colon. *Gastroenterology* (2022). <https://doi.org:10.1053/j.gastro.2022.07.081>
- 6 Escobar, M. *et al.* Intestinal epithelial stem cells do not protect their genome by asymmetric chromosome segregation. *Nat Commun* **2**, 258 (2011).
<https://doi.org:10.1038/ncomms1260>
- 7 Cagan, A. *et al.* Somatic mutation rates scale with lifespan across mammals. *Nature* **604**, 517-524 (2022). <https://doi.org:10.1038/s41586-022-04618-z>
- 8 Hernandez-Chirlaque, C. *et al.* Germ-free and Antibiotic-treated Mice are Highly Susceptible to Epithelial Injury in DSS Colitis. *J Crohns Colitis* **10**, 1324-1335 (2016). <https://doi.org:10.1093/ecco-jcc/jjw096>
- 9 Cazares-Olivera, M. *et al.* Animal unit hygienic conditions influence mouse intestinal microbiota and contribute to T-cell-mediated colitis. *Exp Biol Med (Maywood)* **247**, 1752-1763 (2022). <https://doi.org:10.1177/15353702221113826>
- 10 Bonfiglio, R. *et al.* Extensive Histopathological Characterization of Inflamed Bowel in the Dextran Sulfate Sodium Mouse Model with Emphasis on Clinically Relevant Biomarkers and Targets for Drug Development. *Int J Mol Sci* **22** (2021).
<https://doi.org:10.3390/ijms22042028>
- 11 Araki, Y., Mukaisyo, K., Sugihara, H., Fujiyama, Y. & Hattori, T. Increased apoptosis and decreased proliferation of colonic epithelium in dextran sulfate sodium-induced colitis in mice. *Oncol Rep* **24**, 869-874 (2010). <https://doi.org:10.3892/or.2010.869>
- 12 Emmrich, S. *et al.* Characterization of naked mole-rat hematopoiesis reveals unique stem and progenitor cell patterns and neotenic traits. *EMBO J* **41**, e109694 (2022).
<https://doi.org:10.15252/emboj.2021109694>
- 13 Yamamura, Y. *et al.* Isolation and characterization of neural stem/progenitor cells in the subventricular zone of the naked mole-rat brain. *Inflamm Regen* **41**, 31 (2021).
<https://doi.org:10.1186/s41232-021-00182-7>
- 14 Seluanov, A. *et al.* Distinct tumor suppressor mechanisms evolve in rodent species that differ in size and lifespan. *Aging Cell* **7**, 813-823 (2008).
<https://doi.org:10.1111/j.1474-9726.2008.00431.x>
- 15 Miyawaki, S. *et al.* Tumour resistance in induced pluripotent stem cells derived from naked mole-rats. *Nat Commun* **7**, 11471 (2016).
<https://doi.org:10.1038/ncomms11471>
- 16 Seluanov, A., Gladyshev, V. N., Vijg, J. & Gorbunova, V. Mechanisms of cancer resistance in long-lived mammals. *Nat Rev Cancer* **18**, 433-441 (2018).
<https://doi.org:10.1038/s41568-018-0004-9>
- 17 Seluanov, A. *et al.* Hypersensitivity to contact inhibition provides a clue to cancer resistance of naked mole-rat. *Proc Natl Acad Sci U S A* **106**, 19352-19357 (2009).
<https://doi.org:10.1073/pnas.0905252106>
- 18 Tian, X. *et al.* High-molecular-mass hyaluronan mediates the cancer resistance of the naked mole rat. *Nature* **499**, 346-349 (2013). <https://doi.org:10.1038/nature12234>
- 19 Tian, X. *et al.* INK4 locus of the tumor-resistant rodent, the naked mole rat, expresses a functional p15/p16 hybrid isoform. *Proc Natl Acad Sci U S A* **112**, 1053-1058 (2015). <https://doi.org:10.1073/pnas.1418203112>
- 20 Pentinmikko, N. *et al.* Notum produced by Paneth cells attenuates regeneration of aged intestinal epithelium. *Nature* **571**, 398-402 (2019).
<https://doi.org:10.1038/s41586-019-1383-0>
- 21 Nalapareddy, K. *et al.* Canonical Wnt Signaling Ameliorates Aging of Intestinal Stem Cells. *Cell Rep* **18**, 2608-2621 (2017). <https://doi.org:10.1016/j.celrep.2017.02.056>

- 22 Lee, B. P., Smith, M., Buffenstein, R. & Harries, L. W. Negligible senescence in
naked mole rats may be a consequence of well-maintained splicing regulation.
Geroscience **42**, 633-651 (2020). <https://doi.org:10.1007/s11357-019-00150-7>
- 23 Ruby, J. G., Smith, M. & Buffenstein, R. Naked Mole-Rat mortality rates defy
gompertzian laws by not increasing with age. *Elife* **7** (2018).
<https://doi.org:10.7554/eLife.31157>
- 24 Moorad, J. A., Promislow, D. E., Flesness, N. & Miller, R. A. A comparative
assessment of univariate longevity measures using zoological animal records. *Aging
Cell* **11**, 940-948 (2012). <https://doi.org:10.1111/j.1474-9726.2012.00861.x>
- 25 Debebe, T. *et al.* Analysis of cultivable microbiota and diet intake pattern of the long-
lived naked mole-rat. *Gut Pathog* **8**, 25 (2016). <https://doi.org:10.1186/s13099-016-0107-3>
- 26 Debebe, T. *et al.* Unraveling the gut microbiome of the long-lived naked mole-rat. *Sci
Rep* **7**, 9590 (2017). <https://doi.org:10.1038/s41598-017-10287-0>
- 27 Pamerter, M. E. Adaptations to a hypoxic lifestyle in naked mole-rats. *J Exp Biol* **225**
(2022). <https://doi.org:10.1242/jeb.196725>
- 28 Jarvis, J. U., O'Riain, M. J., Bennett, N. C. & Sherman, P. W. Mammalian eusociality:
a family affair. *Trends Ecol Evol* **9**, 47-51 (1994). [https://doi.org:10.1016/0169-5347\(94\)90267-4](https://doi.org:10.1016/0169-5347(94)90267-4)
- 29 Nowak, M. A., Michor, F. & Iwasa, Y. The linear process of somatic evolution. *Proc
Natl Acad Sci U S A* **100**, 14966-14969 (2003).
<https://doi.org:10.1073/pnas.2535419100>
- 30 Werner, B., Dingli, D., Lenaerts, T., Pacheco, J. M. & Traulsen, A. Dynamics of
mutant cells in hierarchical organized tissues. *Plos Comput Biol* **7**, e1002290 (2011).
<https://doi.org:10.1371/journal.pcbi.1002290>
- 31 Cannataro, V. L., McKinley, S. A. & St Mary, C. M. The evolutionary trade-off
between stem cell niche size, aging, and tumorigenesis. *Evol Appl* **10**, 590-602
(2017). <https://doi.org:10.1111/eva.12476>
- 32 Goodell, M. A. & Rando, T. A. Stem cells and healthy aging. *Science* **350**, 1199-1204
(2015). <https://doi.org:10.1126/science.aab3388>
- 33 Nakada, D. *et al.* Oestrogen increases haematopoietic stem-cell self-renewal in
females and during pregnancy. *Nature* **505**, 555-558 (2014).
<https://doi.org:10.1038/nature12932>
- 34 Clevers, H. The intestinal crypt, a prototype stem cell compartment. *Cell* **154**, 274-
284 (2013). <https://doi.org:10.1016/j.cell.2013.07.004>
- 35 Vanuytsel, T., Senger, S., Fasano, A. & Shea-Donohue, T. Major signaling pathways
in intestinal stem cells. *Biochim Biophys Acta* **1830**, 2410-2426 (2013).
<https://doi.org:10.1016/j.bbagen.2012.08.006>
- 36 Watarai, A. *et al.* Responses to pup vocalizations in subordinate naked mole-rats are
induced by estradiol ingested through coprophagy of queen's feces. *Proc Natl Acad
Sci U S A* **115**, 9264-9269 (2018). <https://doi.org:10.1073/pnas.1720530115>
- 37 Tao, S. *et al.* Wnt activity and basal niche position sensitize intestinal stem and
progenitor cells to DNA damage. *EMBO J* **34**, 624-640 (2015).
<https://doi.org:10.15252/emj.201490700>

REVIEWERS' COMMENTS

Reviewer #1 (Remarks to the Author):

Your response to comment 3 about scRNA-seq analysis is misunderstood. Because scRNA-seq provides a more objective confirmation of the expression levels of proliferation markers in the Lgr5 stem cell subpopulation, you can compare the cycling status of LGR5+/Lgr5+ cells between human, mouse and naked mole rat. We suggested scRNA-seq of naked mole rat intestinal epithelium and did not ask for analysis of novel epithelial, immune and stromal subsets. However, the reviewer's reservations about the histological analysis of the original manuscript that led to these suggestions were mostly resolved by the presentation of a markedly improved images in the revised manuscript. In the revised manuscript, I will not request additional experiments because I do not think that scRNA-seq data is essential. The authors responded to other comments as carefully as possible, within time and availability constraints. No further additional comments.

Reviewer #3 (Remarks to the Author):

The authors have thoroughly addressed my comments. In looking at these changes, I have 2 points that the authors should consider:

1. The authors clarified that they did not distinguish G0 and G1 cell cycle stages in human Lgr5+ ASCs but they assumed that the same ratio of G0 to G1 cells was present in their study as in a previous publication from Ishikawa et al where double staining of Pyronin Y and Hoechst 33342 was used to identify G0 ASCs (based on low DNA, low RNA expression). The authors use this former study on human intestine, but they do not employ the same technique on the NMR. It might be better to tone down that conclusion that NMR ASCs in comparison to human ASC extend their cell cycle by spending 93% of their time in the gap phases without entering G0. In my view this conclusion would require the usage of the same technique on both human and NMR intestine.
2. The BrdU concentration in blood serum: The authors incubate HEK293 cell with defined the blood serum of naked mole rats and compare the measurement with results of HEK293 cell exposed to defined concentration of HEK293-cells (standard curve). This standard is depicted as a straight, dotted line in Suppl. Fig. 5c. However, looking at the standard the measurement points do not follow a straight line more a logarithmic line. Accordingly, the determined concentration 15µg/ml BrdU in blood serum at 8h after injection seems to be an overestimation

REVIEWERS' COMMENTS

Reviewer #1 (Remarks to the Author):

Your response to comment 3 about scRNA-seq analysis is misunderstood. Because scRNA-seq provides a more objective confirmation of the expression levels of proliferation markers in the Lgr5 stem cell subpopulation, you can compare the cycling status of LGR5+/Lgr5+ cells between human, mouse and naked mole rat. We suggested scRNA-seq of naked mole rat intestinal epithelium and did not ask for analysis of novel epithelial, immune and stromal subsets. However, the reviewer's reservations about the histological analysis of the original manuscript that led to these suggestions were mostly resolved by the presentation of a markedly improved images in the revised manuscript. In the revised manuscript, I will not request additional experiments because I do not think that scRNA-seq data is essential. The authors responded to other comments as carefully as possible, within time and availability constraints. No further additional comments.

Response from authors:

We thank the reviewer for his careful consideration of our manuscript.

Reviewer #3 (Remarks to the Author):

The authors have thoroughly addressed my comments. In looking at these changes, I have 2 points that I the authors should consider:

1. The authors clarified that they did not distinguish G0 and G1 cell cycle stages in human *Lgr5*⁺ ASCs but they assumed that the same ratio of G0 to G1 cells was present in their study as in a previous publication from Ishikawa et al where double staining of Pyronin Y and Hoechst 33342 was used to identify G0 ASCs (based on low DNA, low RNA expression). The authors use this former study on human intestine, but they do not employ the same technique on the NMR. It might be better to tone down that conclusion that NMR ASCs in comparison to human ASC extend their cell cycle by spending 93% of their time in the gap phases without entering G0. In my view this conclusion would require the usage of the same technique on both human and NMR intestine.

Response from authors:

We have now updated the manuscript according to the reviewer's suggestion as such:

Previous manuscript: "In summary, this analysis showed that mouse colonic ASCs (*Lgr5*^{+CBC}) divided nearly 70% faster than human and NMR ASCs (Fig. 4j). Human *LGR5*^{+CBC} cells slowed down their division rates by spending 70% of their time in quiescence while NMR *Lgr5*^{+CBC} cells extended their cell cycle by spending 93% of their time in the gap phases (Fig. 4j)."

Updated manuscript: "In summary, our cross-species comparison shows that human and NMR colonic ASCs turnover slower than mouse ASCs (*Lgr5*^{+CBC}) and while human *LGR5*^{+CBC} cells slowed down their division rates by entering quiescence, NMR *Lgr5*^{+CBC} cells extended their cell cycle by spending the majority of their time in the gap phases (Fig. 4j)."

2. The BrdU concentration in blood serum: The authors incubate HEK293 cell with defined the blood serum of naked mole rats and compare the measurement with results of HEK293 cell exposed to defined concentration of HEK293-cells (standard

curve). This standard is depicted as a straight, dotted line in Suppl. Fig. 5c. However, looking at the standard the measurement points do not follow a straight line more a logarithmic line. Accordingly, the determined concentration 15µg/ml BrdU in blood serum at 8h after injection seems to be an overestimation.

Response from authors:

We thank the reviewer for pointing this out. After fitting a logarithmic line to our data in Supplementary Fig. 5c (right), we observed a higher R^2 value than a linear fit, indicating the true trend of the data to be logarithmic. Therefore, we have now used the logarithmic regression equation presented on our updated graph in Supplementary Fig. 5c (right) to determine the BrdU concentration in NMR blood 8 h and 16 h post injection. In agreement with the reviewer’s prediction, the concentration of BrdU in NMR blood was found to be 7.22 µg/mL (± 3) 8 hours after a single BrdU injection and was undetectable at 16 hours post-injection (Supplementary Fig. 5c, right).

Supplementary Fig. 5c (right) | Right, A standard curve showing the mean percentage (\pm SEM) ($n=3$ wells) of BrdU+HEK293T cells with increasing concentrations of BrdU. The concentration of BrdU in NMR sera after 8 h of BrdU injection (18% labelled cells) was derived from the graph (7.22 µg/ml ± 3).